# Distributed Distributionally Robust Optimization with Non-Convex Objectives

**Yang Jiao**
Tongji University
yangjiao@tongji.edu.cn

**Kai Yang**[*]
Tongji University
kaiyang@tongji.edu.cn

**Dongjin Song**
University of Connecticut
dongjin.song@uconn.edu

## Abstract

Distributionally Robust Optimization (DRO), which aims to find an optimal decision that minimizes the worst case cost over the ambiguity set of probability distribution, has been widely applied in diverse applications, *e.g.*, network behavior analysis, risk management, *etc*. However, existing DRO techniques face three key challenges: 1) how to deal with the asynchronous updating in a distributed environment; 2) how to leverage the prior distribution effectively; 3) how to properly adjust the degree of robustness according to different scenarios. To this end, we propose an asynchronous distributed algorithm, named **A**synchronous **S**ingle-loo**P** alternat**I**ve g**R**adient proj**E**ction (ASPIRE) algorithm with the it**E**rative **A**ctive **SE**t method (EASE) to tackle the distributed distributionally robust optimization (DDRO) problem. Furthermore, a new uncertainty set, *i.e.*, constrained $D$-norm uncertainty set, is developed to effectively leverage the prior distribution and flexibly control the degree of robustness. Finally, our theoretical analysis elucidates that the proposed algorithm is guaranteed to converge and the iteration complexity is also analyzed. Extensive empirical studies on real-world datasets demonstrate that the proposed method can not only achieve fast convergence, and remain robust against data heterogeneity as well as malicious attacks, but also tradeoff robustness with performance.

## 1 Introduction

The past decade has witnessed the proliferation of smartphones and Internet of Things (IoT) devices, which generate a plethora of data everyday. Centralized machine learning requires gathering the data to a particular server to train models which incurs high communication overhead [46] and suffers privacy risks [43]. As a remedy, distributed machine learning methods have been proposed. Considering a distributed system composed of $N$ workers (devices), we denote the dataset of these workers as $\{D_1, \cdots, D_N\}$. For the $j^{\text{th}}$ ($1 \le j \le N$) worker, the labeled dataset is given as $D_j = \{\mathbf{x}_j^i, y_j^i\}$, where $\mathbf{x}_j^i \in \mathbb{R}^d$ and $y_j^i \in \{1, \cdots, c\}$ denote the $i^{\text{th}}$ data sample and the corresponding label, respectively. The distributed learning tasks can be formulated as the following optimization problem,

$$\min_{\boldsymbol{w} \in \mathcal{W}} \ F(\boldsymbol{w}) \quad \text{with} \quad F(\boldsymbol{w}) := \sum_j f_j(\boldsymbol{w}), \tag{1}$$

where $\boldsymbol{w} \in \mathbb{R}^p$ is the model parameter to be learned and $\mathcal{W} \subseteq \mathbb{R}^p$ is a nonempty closed convex set, $f_j(\cdot)$ is the empirical risk over the $j^{\text{th}}$ worker involving only the local data:

$$f_j(\boldsymbol{w}) = \sum_{i:\mathbf{x}_j^i \in D_j} \frac{1}{|D_j|} \mathcal{L}_j(\mathbf{x}_j^i, y_j^i; \boldsymbol{w}), \tag{2}$$

---

[*]Corresponding author.

36th Conference on Neural Information Processing Systems (NeurIPS 2022).

where $\mathcal{L}_j$ is the local objective function over the $j^{\text{th}}$ worker. Problem in Eq. (1) arises in numerous areas, such as distributed signal processing [19], multi-agent optimization [36], *etc.* However, such problem does not consider the data heterogeneity [57, 40, 39, 30] among different workers (*i.e.*, data distribution of workers could be substantially different from each other [44]). Indeed, it has been shown that traditional federated approaches, such as FedAvg [33], built for independent and identically distributed (IID) data may perform poorly when applied to Non-IID data [27]. This issue can be mitigated via learning a robust model that aims to achieve uniformly good performance over all workers by solving the following distributionally robust optimization (DRO) problem in a distributed manner:

$$\min_{\boldsymbol{w} \in \mathcal{W}} \max_{\mathbf{p} \in \boldsymbol{\Omega} \subseteq \Delta_N} F(\boldsymbol{w}, \mathbf{p}) := \sum_j p_j f_j(\boldsymbol{w}), \tag{3}$$

where $\mathbf{p} = [p_1, \cdots, p_N] \in \mathbb{R}^N$ is the adversarial distribution in $N$ workers, the $j^{\text{th}}$ entry in this vector, *i.e.*, $p_j$ represents the adversarial distribution value for the $j^{\text{th}}$ worker. $\Delta_N = \{\mathbf{p} \in \mathbb{R}^N_+ : \mathbf{1}^\top \mathbf{p} = 1\}$ and $\boldsymbol{\Omega}$ is a subset of $\Delta_N$. Agnostic federated learning (AFL) [35] firstly introduces the distributionally robust (agnostic) loss in federated learning and provides the convergence rate for (strongly) convex functions. However, AFL does not discuss the setting of $\boldsymbol{\Omega}$. DRFA-Prox [16] considers $\boldsymbol{\Omega} = \Delta_N$ and imposes a regularizer on adversarial distribution to leverage the prior distribution. Nevertheless, three key challenges have not yet been addressed by prior works. First, whether it is possible to construct an uncertainty framework that can not only flexibly maintain the trade-off between the model robustness and performance but also effectively leverage the prior distribution? Second, how to design asynchronous algorithms with guaranteed convergence? Compared to synchronous algorithms, the master in asynchronous algorithms can update its parameters after receiving updates from only a small subset of workers [58, 10]. Asynchronous algorithms are particularly desirable in practice since they can relax strict data dependencies and ensure convergence even in the presence of device failures [58]. Finally, whether it is possible to flexibly adjust the degree of robustness? Moreover, it is necessary to provide convergence guarantee when the objectives (*i.e.*, $f_j(\boldsymbol{w}_j), \forall j$) are non-convex.

To this end, we propose ASPIRE-EASE to effectively address the aforementioned challenges. Firstly, different from existing works, the prior distribution is incorporated within the constraint in our formulation, which can not only leverage the prior distribution more effectively but also achieve guaranteed feasibility for any adversarial distribution within the uncertainty set. The prior distribution can be obtained from side information or uniform distribution [41], which is necessary to construct the uncertainty (ambiguity) set and obtain a more robust model [16]. Specifically, we formulate the prior distribution informed distributionally robust optimization (PD-DRO) problem as:

$$\min_{\boldsymbol{z} \in \mathcal{Z}, \{\boldsymbol{w}_j \in \mathcal{W}\}} \max_{\mathbf{p} \in \mathcal{P}} \sum_j p_j f_j(\boldsymbol{w}_j) \tag{4}$$
$$\text{s.t.} \quad \boldsymbol{z} = \boldsymbol{w}_j, \ j = 1, \cdots, N,$$
$$\text{var.} \quad \boldsymbol{z}, \boldsymbol{w}_1, \boldsymbol{w}_2, \cdots, \boldsymbol{w}_N,$$

where $\boldsymbol{z} \in \mathbb{R}^p$ is the global consensus variable, $\boldsymbol{w}_j \in \mathbb{R}^p$ is the local variable (local model parameter) of $j^{\text{th}}$ worker and $\mathcal{Z} \subseteq \mathbb{R}^p$ is a nonempty closed convex set. $\mathcal{P} \subseteq \mathbb{R}^N_+$ is the uncertainty (ambiguity) set of adversarial distribution $\mathbf{p}$, which is set based on the prior distribution. To solve the PD-DRO problem in an asynchronous distributed manner, we first propose **A**synchronous **S**ingle-loo**P** alternat**I**ve g**R**adient proj**E**ction (ASPIRE), which employs *simple* gradient projection steps for the update of primal and dual variables at every iteration, thus is computationally *efficient*. Next, the it**E**rative **A**ctive **SE**t method (EASE) is employed to replace the traditional cutting plane method to improve the computational efficiency and speed up the convergence. We further provide the convergence guarantee for the proposed algorithm. Furthermore, a new uncertainty set, *i.e.*, constrained $D$-norm ($CD$-norm), is proposed in this paper and its advantages include: 1) it can flexibly control the degree of robustness; 2) the resulting subproblem is computationally simple; 3) it can effectively leverage the prior distribution and flexibly set the bounds for every $p_j$.

**Contributions.** Our contributions can be summarized as follows:

**1.** We formulate a PD-DRO problem with $CD$-norm uncertainty set. PD-DRO incorporates the prior distribution as constraints which can leverage prior distribution more effectively and guarantee robustness. In addition, $CD$-norm is developed to model the ambiguity set around the prior distribution and it provides a flexible way to control the trade-off between model robustness and performance.

**2.** We develop a *single-loop asynchronous* algorithm, namely ASPIRE-EASE, to optimize PD-DRO in an asynchronous distributed manner. ASPIRE employs simple gradient projection steps to

update the variables at every iteration, which is computationally efficient. And EASE is proposed to replace cutting plane method to enhance the computational efficiency and speed up the convergence. We demonstrate that even if the objectives $f_j(\boldsymbol{w}_j), \forall j$ are non-convex, the proposed algorithm is guaranteed to converge. We also theoretically derive the iteration complexity of ASPIRE-EASE.

**3.** Extensive empirical studies on four different real world datasets demonstrate the superior performance of the proposed algorithm. It is seen that ASPIRE-EASE can not only ensure the model's robustness against data heterogeneity but also mitigate malicious attacks.

## 2 Preliminaries

### 2.1 Distributionally Robust Optimization

Optimization problems often contain uncertain parameters. A small perturbation of the parameters could render the optimal solution of the original optimization problem infeasible or completely meaningless [5]. Distributionally robust optimization (DRO) [28, 17, 7] assumes that the probability distributions of uncertain parameters are unknown but remain in an ambiguity (uncertainty) set and aims to find a decision that minimizes the worst case expected cost over the ambiguity set, whose general form can be expressed as,

$$\min_{\boldsymbol{x} \in \mathcal{X}} \max_{P \in \mathbf{P}} \mathbb{E}_P[r(\boldsymbol{x}, \boldsymbol{\xi})], \tag{5}$$

where $\boldsymbol{x} \in \mathcal{X}$ represents the decision variable, $\mathbf{P}$ is the ambiguity set of probability distributions $P$ of uncertain parameters $\boldsymbol{\xi}$. Existing methods for solving DRO can be broadly grouped into two widely-used categories [42]: 1) Dual methods [15, 50, 18] reformulate the primal DRO problems as deterministic optimization problems through duality theory. Ben-Tal et al. [2] reformulate the robust linear optimization (RLO) problem with an ellipsoidal uncertainty set as a second-order cone optimization problem (SOCP). 2) Cutting plane methods [34, 6] (also called adversarial approaches [21]) continuously solve an approximate problem with a finite number of constraints of the primal DRO problem, and subsequently check whether new constraints are needed to refine the feasible set. Recently, several new methods [41, 29, 23] have been developed to solve DRO, which need to solve the inner maximization problem at every iteration.

### 2.2 Cutting Plane Method for PD-DRO

In this section, we introduce the cutting plane method for PD-DRO in Eq. (4). We first reformulate PD-DRO by introducing an additional variable $h \in \mathcal{H}$ ($\mathcal{H} \subseteq \mathbb{R}^1$ is a nonempty closed convex set) and protection function $g(\{\boldsymbol{w}_j\})$ [55]. Introducing additional variable $h$ is an epigraph reformulation [3, 56]. In this case, Eq. (4) can be reformulated as the form with uncertainty in the constraints:

$$\min_{\boldsymbol{z} \in \mathcal{Z}, \{\boldsymbol{w}_j \in \mathcal{W}\}, h \in \mathcal{H}} h$$
$$\text{s.t.} \ \sum_j \overline{p} f_j(\boldsymbol{w}_j) + g(\{\boldsymbol{w}_j\}) - h \leq 0, \tag{6}$$
$$\boldsymbol{z} = \boldsymbol{w}_j, \ j = 1, \cdots, N,$$
$$\text{var.} \ \ \boldsymbol{z}, \boldsymbol{w}_1, \boldsymbol{w}_2, \cdots, \boldsymbol{w}_N, h,$$

where $\overline{p}$ is the nominal value of the adversarial distribution for every worker and $g(\{\boldsymbol{w}_j\}) = \max_{\mathbf{p} \in \mathcal{P}} \sum_j (p_j - \overline{p}) f_j(\boldsymbol{w}_j)$ is the protection function. Eq. (6) is a semi-infinite program (SIP) which contains infinite constraints and cannot be solved directly [42]. Denoting the set of cutting plane parameters in $(t+1)^{\text{th}}$ iteration as $\mathbf{A}^t \subseteq \mathbb{R}^N$, the following function is used to approximate $g(\{\boldsymbol{w}_j\})$:

$$\overline{g}(\{\boldsymbol{w}_j\}) = \max_{\boldsymbol{a}_l \in \mathbf{A}^t} \boldsymbol{a}_l^\top \mathbf{f}(\boldsymbol{w}) = \max_{\boldsymbol{a}_l \in \mathbf{A}^t} \sum_j a_{l,j} f_j(\boldsymbol{w}_j), \tag{7}$$

where $\boldsymbol{a}_l = [a_{l,1}, \cdots, a_{l,N}] \in \mathbb{R}^N$ denotes the parameters of $l^{\text{th}}$ cutting plane in $\mathbf{A}^t$ and $\mathbf{f}(\boldsymbol{w}) = [f_1(\boldsymbol{w}_1), \cdots, f_N(\boldsymbol{w}_N)] \in \mathbb{R}^N$. Substituting the protection function $g(\{\boldsymbol{w}_j\})$ with $\overline{g}(\{\boldsymbol{w}_j\})$, we can obtain the following approximate problem:

$$\min_{\boldsymbol{z} \in \mathcal{Z}, \{\boldsymbol{w}_j \in \mathcal{W}\}, h \in \mathcal{H}} h$$
$$\text{s.t.} \ \sum_j (\overline{p} + a_{l,j}) f_j(\boldsymbol{w}_j) - h \leq 0, \forall \boldsymbol{a}_l \in \mathbf{A}^t, \tag{8}$$
$$\boldsymbol{z} = \boldsymbol{w}_j, \ j = 1, \cdots, N,$$
$$\text{var.} \ \ \boldsymbol{z}, \boldsymbol{w}_1, \boldsymbol{w}_2, \cdots, \boldsymbol{w}_N, h.$$

# 3 ASPIRE

Distributed optimization is an attractive approach for large-scale learning tasks [54, 8] since it does not require data aggregation, which protects data privacy while also reducing bandwidth requirements [45]. When the neural network models (*i.e.*, $f_j(\boldsymbol{w}_j), \forall j$ are non-convex functions) are used, solving problem in Eq. (8) in a distributed manner facing two challenges: 1) Computing the optimal solution to a non-convex subproblem requires a large number of iterations and therefore is highly computationally intensive if not impossible. Thus, the traditional Alternating Direction Method of Multipliers (ADMM) is ineffective. 2) The communication delays of workers may differ significantly [11], thus, asynchronous algorithms are strongly preferred.

To this end, we propose the **A**synchronous **S**ingle-loo**P** alternat**I**ve g**R**adient proj**E**ction (ASPIRE). The advantages of the proposed algorithm include: 1) ASPIRE uses simple gradient projection steps to update variables in each iteration and therefore it is computationally more efficient than the traditional ADMM method, which seeks to find the optimal solution in non-convex (for $\boldsymbol{w}_j, \forall j$) and convex (for $\boldsymbol{z}$ and $h$) optimization subproblems every iteration, 2) the proposed asynchronous algorithm does not need strict synchronization among different workers. Therefore, ASPIRE remains resilient against communication delays and potential hardware failures from workers. Details of the algorithm are given below. Firstly, we define the node as master which is responsible for updating the global variable $\boldsymbol{z}$, and we define the node which is responsible for updating the local variable $\boldsymbol{w}_j$ as worker $j$. In each iteration, the master updates its variables once it receives updates from at least $S$ workers, *i.e.*, active workers, satisfying $1 \leq S \leq N$. $\mathbf{Q}^{t+1}$ denotes the index subset of workers from which the master receives updates during $(t+1)^{\text{th}}$ iteration. We also assume the master will receive updated variables from every worker at least once for each $\tau$ iterations. The augmented Lagrangian function of Eq. (8) can be written as:

$$L_p = h + \sum_l \lambda_l \left( \sum_j (\overline{p} + a_{l,j}) f_j(\boldsymbol{w}_j) - h \right) + \sum_j \boldsymbol{\phi}_j^\top (\boldsymbol{z} - \boldsymbol{w}_j) + \sum_j \frac{\kappa_1}{2} ||\boldsymbol{z} - \boldsymbol{w}_j||^2, \quad (9)$$

where $L_p = L_p(\{\boldsymbol{w}_j\}, \boldsymbol{z}, h, \{\lambda_l\}, \{\boldsymbol{\phi}_j\})$, $\lambda_l \in \boldsymbol{\Lambda}, \forall l$ and $\boldsymbol{\phi}_j \in \boldsymbol{\Phi}, \forall j$ represent the dual variables of inequality and equality constraints in Eq. (8), respectively. $\boldsymbol{\Lambda} \subseteq \mathbb{R}^1$ and $\boldsymbol{\Phi} \subseteq \mathbb{R}^p$ are nonempty closed convex sets, constant $\kappa_1 > 0$ is a penalty parameter. Note that Eq. (9) does not consider the second-order penalty term for inequality constraint since it will invalidate the distributed optimization. Following [52], the regularized version of Eq. (9) is employed to update all variables as follows,

$$\widetilde{L}_p(\{\boldsymbol{w}_j\}, \boldsymbol{z}, h, \{\lambda_l\}, \{\boldsymbol{\phi}_j\}) = L_p - \sum_l \frac{c_1^t}{2} ||\lambda_l||^2 - \sum_j \frac{c_2^t}{2} ||\boldsymbol{\phi}_j||^2, \quad (10)$$

where $c_1^t$ and $c_2^t$ denote the regularization terms in $(t+1)^{\text{th}}$ iteration. To avoid enumerating the whole dataset, the mini-batch loss could be used. A batch of instances with size $m$ can be randomly sampled from each worker during each iteration. The loss function of these instances from $j^{\text{th}}$ worker is given by $\hat{f}_j(\boldsymbol{w}_j) = \sum_{i=1}^m \frac{1}{m} \mathcal{L}_j(\mathbf{x}_j^i, y_j^i; \boldsymbol{w}_j)$. It is evident that $\mathbb{E}[\hat{f}_j(\boldsymbol{w}_j)] = f_j(\boldsymbol{w}_j)$ and $\mathbb{E}[\nabla \hat{f}_j(\boldsymbol{w}_j)] = \nabla f_j(\boldsymbol{w}_j)$. In $(t+1)^{\text{th}}$ master iteration, the proposed algorithm proceeds as follows.

**1)** *Active workers* update the local variables $\boldsymbol{w}_j$ as follows,

$$\boldsymbol{w}_j^{t+1} = \begin{cases} \mathcal{P}_{\boldsymbol{\mathcal{W}}}(\boldsymbol{w}_j^t - \alpha_{\boldsymbol{w}}^{\widetilde{t}_j} \nabla_{\boldsymbol{w}_j} \widetilde{L}_p(\{\boldsymbol{w}_j^{\widetilde{t}_j}\}, \boldsymbol{z}^{\widetilde{t}_j}, h^{\widetilde{t}_j}, \{\lambda_l^{\widetilde{t}_j}\}, \{\boldsymbol{\phi}_j^{\widetilde{t}_j}\})), \forall j \in \mathbf{Q}^{t+1}, \\ \boldsymbol{w}_j^t, \forall j \notin \mathbf{Q}^{t+1}, \end{cases} \quad (11)$$

where $\widetilde{t}_j$ is the last iteration during which worker $j$ was active. It is seen that $\forall j \in \mathbf{Q}^{t+1}, \boldsymbol{w}_j^t = \boldsymbol{w}_j^{\widetilde{t}_j}$ and $\boldsymbol{\phi}_j^t = \boldsymbol{\phi}_j^{\widetilde{t}_j}$. $\alpha_{\boldsymbol{w}}^{\widetilde{t}_j}$ represents the step-size and let $\alpha_{\boldsymbol{w}}^t = \eta_{\boldsymbol{w}}^t$ when $t < T_1$ and $\alpha_{\boldsymbol{w}}^t = \underline{\eta_{\boldsymbol{w}}}$ when $t \geq T_1$, where $\eta_{\boldsymbol{w}}^t$ and constant $\underline{\eta_{\boldsymbol{w}}}$ will be introduced below. $\mathcal{P}_{\boldsymbol{\mathcal{W}}}$ represents the projection onto the closed convex set $\boldsymbol{\mathcal{W}}$ and we set $\boldsymbol{\mathcal{W}} = \{\boldsymbol{w}_j | \ ||\boldsymbol{w}_j||_\infty \leq \alpha_1\}$, $\alpha_1$ is a positive constant. And then, the active workers ($j \in \mathbf{Q}^{t+1}$) transmit their local model parameters $\boldsymbol{w}_j^{t+1}$ and loss $f_j(\boldsymbol{w}_j)$ to the master.

**2)** After receiving the updates from active workers, the *master* updates the global consensus variable $\boldsymbol{z}$, additional variable $h$ and dual variables $\lambda_l$ as follows,

$$\boldsymbol{z}^{t+1} = \mathcal{P}_{\boldsymbol{\mathcal{Z}}}(\boldsymbol{z}^t - \eta_{\boldsymbol{z}}^t \nabla_{\boldsymbol{z}} \widetilde{L}_p(\{\boldsymbol{w}_j^{t+1}\}, \boldsymbol{z}^t, h^t, \{\lambda_l^t\}, \{\boldsymbol{\phi}_j^t\})), \quad (12)$$

$$h^{t+1} = \mathcal{P}_{\boldsymbol{\mathcal{H}}}(h^t - \eta_h^t \nabla_h \widetilde{L}_p(\{\boldsymbol{w}_j^{t+1}\}, \boldsymbol{z}^{t+1}, h^t, \{\lambda_l^t\}, \{\boldsymbol{\phi}_j^t\})), \quad (13)$$

$$\lambda_l^{t+1} = \mathcal{P}_{\boldsymbol{\Lambda}}(\lambda_l^t + \rho_1 \nabla_{\lambda_l} \widetilde{L}_p(\{\boldsymbol{w}_j^{t+1}\}, \boldsymbol{z}^{t+1}, h^{t+1}, \{\lambda_l^t\}, \{\phi_j^t\})),\ l=1,\cdots,|\mathbf{A}^t|, \tag{14}$$

where $\eta_{\boldsymbol{z}}^t$, $\eta_h^t$ and $\rho_1$ represent the step-sizes. $\mathcal{P}_{\boldsymbol{\mathcal{Z}}}$, $\mathcal{P}_{\boldsymbol{\mathcal{H}}}$ and $\mathcal{P}_{\boldsymbol{\Lambda}}$ respectively represent the projection onto the closed convex sets $\boldsymbol{\mathcal{Z}}$, $\boldsymbol{\mathcal{H}}$ and $\boldsymbol{\Lambda}$. We set $\boldsymbol{\mathcal{Z}} = \{\boldsymbol{z}|\ ||\boldsymbol{z}||_\infty \leq \alpha_1\}$, $\boldsymbol{\mathcal{H}} = \{h|\ 0 \leq h \leq \alpha_2\}$ and $\boldsymbol{\Lambda} = \{\lambda_l|\ 0 \leq \lambda_l \leq \alpha_3\}$, where $\alpha_2$ and $\alpha_3$ are positive constants. $|\mathbf{A}^t|$ denotes the number of cutting planes. Then, master broadcasts $\boldsymbol{z}^{t+1}$, $h^{t+1}$, $\{\lambda_l^{t+1}\}$ to the active workers.

**3)** *Active workers* update the local dual variables $\phi_j$ as follows,

$$\phi_j^{t+1} = \begin{cases} \mathcal{P}_{\boldsymbol{\Phi}}(\phi_j^t + \rho_2 \nabla_{\phi_j} \widetilde{L}_p(\{\boldsymbol{w}_j^{t+1}\}, \boldsymbol{z}^{t+1}, h^{t+1}, \{\lambda_l^{t+1}\}, \{\phi_j^t\})), \forall j \in \mathbf{Q}^{t+1}, \\ \phi_j^t, \forall j \notin \mathbf{Q}^{t+1}, \end{cases} \tag{15}$$

where $\rho_2$ represents the step-size and $\mathcal{P}_{\boldsymbol{\Phi}}$ represents the projection onto the closed convex set $\boldsymbol{\Phi}$ and we set $\boldsymbol{\Phi} = \{\phi_j|\ ||\phi_j||_\infty \leq \alpha_4\}$, $\alpha_4$ is a positive constant. And master can also obtain $\{\phi_j^{t+1}\}$ according to Eq. (15). It is seen that the projection operation in each step is computationally simple since the closed convex sets have simple structures [4].

## 4 Iterative Active Set Method

Cutting plane methods may give rise to numerous linear constraints and lots of extra message passing [55]. Moreover, more iterations are required to obtain the $\varepsilon$-stationary point when the size of a set containing cutting planes increases (which corresponds to a larger $M$), which can be seen in Theorem 1. To improve the computational efficiency and speed up the convergence, we consider removing the inactive cutting planes. The proposed it**E**rative **A**ctive **SE**t method (EASE) can be divided into the two steps: during $T_1$ iterations, 1) solving the cutting plane generation subproblem to generate cutting plane, and 2) removing the inactive cutting plane every $k$ iterations, where $k > 0$ is a pre-set constant and can be controlled flexibly.

The cutting planes are generated according to the uncertainty set. For example, if we employ ellipsoid uncertainty set, the cutting plane is generated via solving a SOCP. In this paper, we propose $CD$-norm uncertainty set, which can be expressed as follows,

$$\mathcal{P} = \{\mathbf{p}\colon -\widetilde{p}_j \leq p_j - q_j \leq \widetilde{p}_j, \sum_j |\frac{p_j - q_j}{\widetilde{p}_j}| \leq \Gamma, \mathbf{1}^\top \mathbf{p} = 1\}, \tag{16}$$

where $\Gamma \in \mathbb{R}^1$ can flexibly control the level of robustness, $\mathbf{q} = [q_1, \cdots, q_N] \in \mathbb{R}^N$ represents the prior distribution, $-\widetilde{p}_j$ and $\widetilde{p}_j$ ($\widetilde{p}_j \geq 0$) represent the lower and upper bounds for $p_j - q_j$, respectively. The setting of $\mathbf{q}$ and $\widetilde{p}_j, \forall j$ are based on the prior knowledge. $D$-norm is a classical uncertainty set (which is also called as budget uncertainty set) [5]. We call Eq. (16) $CD$-norm uncertainty set since $\mathbf{p}$ is a probability vector so all the entries of this vector are non-negative and add up to exactly one, *i.e.*, $\mathbf{1}^\top \mathbf{p} = 1$. Due to the special structure of $CD$-norm, the cutting plane generation subproblem is easy to solve and the level of robustness in terms of the outage probability, *i.e.*, probabilistic bounds of the violations of constraints can be flexibly adjusted via a single parameter $\Gamma$. We claim that $l_1$-norm (or twice total variation distance) uncertainty set is closely related to $CD$-norm uncertainty set. Nevertheless, there are two differences: 1) $CD$-norm uncertainty set could be regarded as a weighted $l_1$-norm with additional constraints. 2) $CD$-norm uncertainty set can flexibly set the lower and upper bounds for every $p_j$ (*i.e.*, $q_j - \widetilde{p}_j \leq p_j \leq p_j + \widetilde{p}_j$), while $0 \leq p_j \leq 1, \forall j$ in $l_1$-norm uncertainty set. Based on the $CD$-norm uncertainty set, the cutting plane can be derived as follows,

1) Solve the following problem,

$$\mathbf{p}^{t+1} = \arg\max_{p_1, \cdots, p_N} \sum_j (p_j - \overline{p}) f_j(\boldsymbol{w}_j)$$

$$\text{s.t. } \sum_j |\frac{p_j - q_j}{\widetilde{p}_j}| \leq \Gamma,\ -\widetilde{p}_j \leq p_j - q_j \leq \widetilde{p}_j, \forall j,\ \sum_j p_j = 1 \tag{17}$$

$$\text{var. } \quad p_1, \cdots, p_N,$$

where $\mathbf{p}^{t+1} = [p_1^{t+1}, \cdots, p_N^{t+1}] \in \mathbb{R}^N$. Let $\widetilde{\mathbf{a}}^{t+1} = \mathbf{p}^{t+1} - \overline{\mathbf{p}}$, where $\overline{\mathbf{p}} = [\overline{p}, \cdots, \overline{p}] \in \mathbb{R}^N$. This first step aims to obtain the distribution $\widetilde{\mathbf{a}}^{t+1}$ by solving problem in Eq. (17). This problem can be effectively solved through combining merge sort [13] (for sorting $\widetilde{p}_j f_j(\boldsymbol{w}_j), j=1, \cdots, N$) with few basic arithmetic operations (for obtaining $p_j^{t+1}, j=1, \cdots, N$). Since $N$ is relatively large in

---

**Algorithm 1** ASPIRE-EASE

---

**Initialization:** iteration $t = 0$, variables $\{\boldsymbol{w}_j^0\}$, $\boldsymbol{z}^0$, $h^0$, $\{\lambda_l^0\}$, $\{\boldsymbol{\phi}_j^0\}$ and set $\mathbf{A}^0$.

**repeat**
  **for** *active worker* **do**
    updates local $\boldsymbol{w}_j^{t+1}$ according to Eq. (11);
  **end for**
  *active workers* transmit local model parameters and loss to *master*;
  *master* receives updates from *active workers* **do**
    updates $\boldsymbol{z}^{t+1}$, $h^{t+1}$, $\{\lambda_l^{t+1}\}$, $\{\boldsymbol{\phi}_j^{t+1}\}$ in master according to Eq. (12), (13), (14), (15);
  *master* broadcasts $\boldsymbol{z}^{t+1}$, $h^{t+1}$, $\{\lambda_l^{t+1}\}$ to *active workers*;
  **for** *active worker* **do**
    updates local $\boldsymbol{\phi}_j^{t+1}$ according to Eq. (15);
  **end for**
  **if** $(t + 1)$ mod $k == 0$ and $t < T_1$ **then**
    *master* updates $\mathbf{A}^{t+1}$ according to Eq. (19) and (20), and broadcast parameters to all workers;
  **end if**
  $t = t + 1$;
**until** convergence

---

distributed system, the arithmetic complexity of solving problem in Eq. (17) is dominated by merge sort, which can be regarded as $\mathcal{O}(N \log(N))$.

2) Let $\mathbf{f}(\boldsymbol{w}) = [f_1(\boldsymbol{w}_1), \cdots, f_N(\boldsymbol{w}_N)] \in \mathbb{R}^N$, check the feasibility of the following constraints:

$$\widetilde{\mathbf{a}}^{t+1\top}\mathbf{f}(\boldsymbol{w}) \leq \max_{\boldsymbol{a}_l \in \mathbf{A}^t} \boldsymbol{a}_l^{\top}\mathbf{f}(\boldsymbol{w}). \tag{18}$$

3) If Eq. (18) is violated, $\widetilde{\mathbf{a}}^{t+1}$ will be added into $\mathbf{A}^t$:

$$\mathbf{A}^{t+1} = \begin{cases} \mathbf{A}^t \cup \{\widetilde{\mathbf{a}}^{t+1}\}, \text{if Eq.(18) is violated,} \\ \mathbf{A}^t, \text{otherwise,} \end{cases} \tag{19}$$

when a new cutting plane is added, its corresponding dual variable $\lambda_{|\mathbf{A}^t|+1}^{t+1} = 0$ will be generated. After the cutting plane subproblem is solved, the inactive cutting plane will be removed, that is:

$$\mathbf{A}^{t+1} = \begin{cases} \complement_{\mathbf{A}^{t+1}}\{\boldsymbol{a}_l\}, \text{if } \lambda_l^{t+1} = 0 \text{ and } \lambda_l^t = 0, 1 \leq l \leq |\mathbf{A}^t|, \\ \mathbf{A}^{t+1}, \text{otherwise,} \end{cases} \tag{20}$$

where $\complement_{\mathbf{A}^{t+1}}\{\boldsymbol{a}_l\}$ is the complement of $\{\boldsymbol{a}_l\}$ in $\mathbf{A}^{t+1}$, and the dual variable will be removed. Then master broadcasts $\mathbf{A}^{t+1}$, $\{\lambda_l^{t+1}\}$ to all workers. Details of algorithm are summarized in Algorithm 1.

## 5 Convergence Analysis

**Definition 1** (Stationarity gap) *Following [52, 32, 53], the stationarity gap of our problem at $t^{th}$ iteration is defined as:*

$$\nabla G^t = \begin{bmatrix} \{\frac{1}{\alpha_{\boldsymbol{w}}^t}(\boldsymbol{w}_j^t - \mathcal{P}_{\boldsymbol{\mathcal{W}}}(\boldsymbol{w}_j^t - \alpha_{\boldsymbol{w}}^t \nabla_{\boldsymbol{w}_j} L_p(\{\boldsymbol{w}_j^t\}, \boldsymbol{z}^t, h^t, \{\lambda_l^t\}, \{\boldsymbol{\phi}_j^t\}))))\} \\ \frac{1}{\eta_{\boldsymbol{z}}^t}(\boldsymbol{z}^t - \mathcal{P}_{\boldsymbol{\mathcal{Z}}}(\boldsymbol{z}^t - \eta_{\boldsymbol{z}}^t \nabla_{\boldsymbol{z}} L_p(\{\boldsymbol{w}_j^t\}, \boldsymbol{z}^t, h^t, \{\lambda_l^t\}, \{\boldsymbol{\phi}_j^t\}))) \\ \frac{1}{\eta_h^t}(h^t - \mathcal{P}_{\boldsymbol{\mathcal{H}}}(h^t - \eta_h^t \nabla_h L_p(\{\boldsymbol{w}_j^t\}, \boldsymbol{z}^t, h^t, \{\lambda_l^t\}, \{\boldsymbol{\phi}_j^t\}))) \\ \{\frac{1}{\rho_1}(\lambda_l^t - \mathcal{P}_{\boldsymbol{\Lambda}}(\lambda_l^t + \rho_1 \nabla_{\lambda_l} L_p(\{\boldsymbol{w}_j^t\}, \boldsymbol{z}^t, h^t, \{\lambda_l^t\}, \{\boldsymbol{\phi}_j^t\})))\} \\ \{\frac{1}{\rho_2}(\boldsymbol{\phi}_j^t - \mathcal{P}_{\boldsymbol{\Phi}}(\boldsymbol{\phi}_j^t + \rho_2 \nabla_{\boldsymbol{\phi}_j} L_p(\{\boldsymbol{w}_j^t\}, \boldsymbol{z}^t, h^t, \{\lambda_l^t\}, \{\boldsymbol{\phi}_j^t\})))\} \end{bmatrix}, \tag{21}$$

*where $\nabla G^t$ is the simplified form of $\nabla G(\{\boldsymbol{w}_j^t\}, \boldsymbol{z}^t, h^t, \{\lambda_l^t\}, \{\boldsymbol{\phi}_j^t\})$.*

**Definition 2** ($\varepsilon$-stationary point) $(\{\boldsymbol{w}_j^t\}, \boldsymbol{z}^t, h^t, \{\lambda_l^t\}, \{\boldsymbol{\phi}_j^t\})$ *is an $\varepsilon$-stationary point ($\varepsilon \geq 0$) of a differentiable function $L_p$, if $||\nabla G^t|| \leq \varepsilon$. $T(\varepsilon)$ is the first iteration index such that $||\nabla G^t|| \leq \varepsilon$, i.e., $T(\varepsilon) = \min\{t \mid ||\nabla G^t|| \leq \varepsilon\}$.*

**Assumption 1** (Smoothness/Gradient Lipschitz) *$L_p$ has Lipschitz continuous gradients. We assume that there exists $L > 0$ satisfying*

$$||\nabla_\theta L_p(\{\boldsymbol{w}_j\}, \boldsymbol{z}, h, \{\lambda_l\}, \{\boldsymbol{\phi}_j\}) - \nabla_\theta L_p(\{\hat{\boldsymbol{w}}_j\}, \hat{\boldsymbol{z}}, \hat{h}, \{\hat{\lambda}_l\}, \{\hat{\boldsymbol{\phi}}_j\})||$$
$$\leq L||[\boldsymbol{w}_{\text{cat}} - \hat{\boldsymbol{w}}_{\text{cat}}; \boldsymbol{z} - \hat{\boldsymbol{z}}; h - \hat{h}; \boldsymbol{\lambda}_{\text{cat}} - \hat{\boldsymbol{\lambda}}_{\text{cat}}; \boldsymbol{\phi}_{\text{cat}} - \hat{\boldsymbol{\phi}}_{\text{cat}}]||,$$

*where* $\theta \in \{\{\boldsymbol{w}_j\}, \boldsymbol{z}, h, \{\lambda_l\}, \{\boldsymbol{\phi}_j\}\}$ *and* $[;]$ *represents the concatenation.* $\boldsymbol{w}_{\text{cat}} - \hat{\boldsymbol{w}}_{\text{cat}} = [\boldsymbol{w}_1 - \hat{\boldsymbol{w}}_1; \cdots; \boldsymbol{w}_N - \hat{\boldsymbol{w}}_N] \in \mathbb{R}^{pN}$, $\boldsymbol{\lambda}_{\text{cat}} - \hat{\boldsymbol{\lambda}}_{\text{cat}} = [\lambda_1 - \hat{\lambda}_1; \cdots; \lambda_{|\mathbf{A}^t|} - \hat{\lambda}_{|\mathbf{A}^t|}] \in \mathbb{R}^{|\mathbf{A}^t|}$, $\boldsymbol{\phi}_{\text{cat}} - \hat{\boldsymbol{\phi}}_{\text{cat}} = [\boldsymbol{\phi}_1 - \hat{\boldsymbol{\phi}}_1; \cdots; \boldsymbol{\phi}_N - \hat{\boldsymbol{\phi}}_N] \in \mathbb{R}^{pN}$.

**Assumption 2** (Boundedness) *Before obtaining the $\varepsilon$-stationary point (i.e., $t \leq T(\varepsilon) - 1$), we assume variables in master satisfy that* $||\boldsymbol{z}^{t+1} - \boldsymbol{z}^t||^2 + ||h^{t+1} - h^t||^2 + \sum_l ||\lambda_l^{t+1} - \lambda_l^t||^2 \geq \vartheta$, *where* $\vartheta > 0$ *is a relative small constant. The change of the variables in master is upper bounded within $\tau$ iterations:*
$$||\boldsymbol{z}^t - \boldsymbol{z}^{t-k}||^2 \leq \tau k_1 \vartheta, \quad ||h^t - h^{t-k}||^2 \leq \tau k_1 \vartheta, \quad \sum_l ||\lambda_l^t - \lambda_l^{t-k}||^2 \leq \tau k_1 \vartheta, \forall 1 \leq k \leq \tau,$$
*where $k_1 > 0$ is a constant.*

**Setting 1** (Bounded $|\mathbf{A}^t|$) $|\mathbf{A}^t| \leq M, \forall t$, *i.e., an upper bound is set for the number of cutting planes.*

**Setting 2** (Setting of $c_1^t$, $c_2^t$) $c_1^t = \frac{1}{\rho_1(t+1)^{\frac{1}{6}}} \geq \underline{c}_1$ *and* $c_2^t = \frac{1}{\rho_2(t+1)^{\frac{1}{6}}} \geq \underline{c}_2$ *are nonnegative nonincreasing sequences, where $\underline{c}_1$ and $\underline{c}_2$ are positive constants and meet* $M\underline{c}_1^2 + N\underline{c}_2^2 \leq \frac{\varepsilon^2}{4}$.

**Theorem 1** (Iteration complexity) *Suppose Assumption 1 and 2 hold. We set* $\eta_{\boldsymbol{w}}^t = \eta_{\boldsymbol{z}}^t = \eta_h^t = \frac{2}{L + \rho_1 |\mathbf{A}^t| L^2 + \rho_2 N L^2 + 8(\frac{|\mathbf{A}^t| \gamma L^2}{\rho_1 (c_1^t)^2} + \frac{N \gamma L^2}{\rho_2 (c_2^t)^2})}$ *and* $\underline{\eta}_{\boldsymbol{w}} = \frac{2}{L + \rho_1 M L^2 + \rho_2 N L^2 + 8(\frac{M \gamma L^2}{\rho_1 \underline{c}_1^2} + \frac{N \gamma L^2}{\rho_2 \underline{c}_2^2})}$. *And we set constants* $\rho_1 < \min\{\frac{2}{L + 2c_1^0}, \frac{1}{15\tau k_1 N L^2}\}$ *and* $\rho_2 \leq \frac{2}{L + 2c_2^0}$, *respectively. For a given $\varepsilon$, we have:*

$$T(\varepsilon) \sim \mathcal{O}\left(\max\left\{\left(\frac{4M\sigma_1^2}{\rho_1^2} + \frac{4N\sigma_2^2}{\rho_2^2}\right)^3 \frac{1}{\varepsilon^6}, \left(\frac{4(d_6 + \frac{\rho_2(N-S)L^2}{2})^2(\bar{d} + k_d(\tau - 1))d_5}{\varepsilon^2} + (T_1 + \tau)^{\frac{1}{3}}\right)^3\right\}\right), \tag{22}$$

*where $\sigma_1$, $\sigma_2$, $\gamma$, $\tau$, $k_d$, $\bar{d}$, $d_5$, $d_6$ and $T_1$ are constants. The detailed proof is given in Appendix A.*

There exists a wide array of works regarding the convergence analysis of various algorithms for nonconvex/convex optimization problems involved in machine learning [25, 53]. Our analysis, however, differs from existing works in two aspects. First, we solve the non-convex PD-DRO in an *asynchronous distributed manner*. To our best knowledge, there are few works focusing on solving the DRO in a distributed manner. Compared to solving the non-convex PD-DRO in a centralized manner, solving it in an *asynchronous distributed manner* poses significant challenges in algorithm design and convergence analysis. Secondly, we do not assume the inner problem can be solved nearly optimally for each outer iteration, which is numerically difficult to achieve in practice [4]. Instead, ASPIRE-EASE is *single loop* and involves simple gradient projection operation at each step.

## 6 Experiment

In this section, we conduct experiments on four real-world datasets to assess the performance of the proposed method. Specifically, we evaluate the robustness against data heterogeneity, robustness against malicious attacks and efficiency of the proposed method. Ablation study is also carried out to demonstrate the excellent performance of ASPIRE-EASE.

### 6.1 Datasets and Baseline Methods

We compare the proposed ASPIRE-EASE with baseline methods based on SHL [20], Person Activity [26], Single Chest-Mounted Accelerometer (SM-AC) [9] and Fashion MNIST [51] datasets. The baseline methods include $\text{Ind}_j$ (learning the model from an individual worker $j$), $\text{Mix}_{\text{Even}}$ (learning the model from all workers with even weights using ASPIRE), FedAvg [33], AFL [35] and DRFA-Prox [16]. The detailed descriptions of datasets and baselines are given in Appendix C.

In our empirical studies, since the downstream tasks are multi-class classification, the cross entropy loss is used on each worker (*i.e.*, $\mathcal{L}_j(\cdot), \forall j$). For SHL, Person Activity, and SM-AC datasets, we adopt the deep multilayer perceptron [49] as the base model. And we use the same logistic regression model as in [35, 16] for Fashion MNIST dataset. The base models are trained with SGD. More details are given in Appendix C. Following related works in this direction [41, 35, 16], worst case performance are reported for the comparison of robustness. Specifically, we use $\mathbf{Acc}_w$ and $\mathbf{Loss}_w$ to represent the worst case test accuracy and training loss (*i.e.*, the test accuracy and training loss on the worker with worst performance), respectively. We also report the standard deviation $\mathbf{Std}$ of

Table 1: Performance comparisons based on $\mathbf{Acc}_w$ (%) ↑, $\mathbf{Loss}_w$ ↓ and $\mathbf{Std}$ ↓ (↑ and ↓ respectively denote higher scores represent better performance and lower scores represent better performance). The boldfaced digits represent the best results, "$-$" represents not available.

| Model | SHL | | | Person Activity | | | SC-MA | | | Fashion MNIST | | |
|---|---|---|---|---|---|---|---|---|---|---|---|---|
| | $\mathbf{Acc}_w$↑ | $\mathbf{Loss}_w$↓ | $\mathbf{Std}$↓ | $\mathbf{Acc}_w$↑ | $\mathbf{Loss}_w$↓ | $\mathbf{Std}$↓ | $\mathbf{Acc}_w$↑ | $\mathbf{Loss}_w$↓ | $\mathbf{Std}$↓ | $\mathbf{Acc}_w$↑ | $\mathbf{Loss}_w$↓ | $\mathbf{Std}$↓ |
| max{Ind$_j$} | 19.06±0.65 | – | 29.1 | 49.38±0.08 | – | 8.32 | 22.56±0.78 | – | 17.5 | – | – | – |
| Mix$_{Even}$ | 69.87±3.10 | 0.806±0.018 | 4.81 | 56.31±0.69 | 1.165±0.017 | 3.00 | 49.81±0.21 | 1.424±0.024 | 6.99 | 66.80±0.18 | 0.784±0.003 | 10.1 |
| FedAvg [33] | 69.96±3.07 | 0.802±0.023 | 5.21 | 56.28±0.63 | 1.154±0.019 | 3.13 | 49.53±0.96 | 1.441±0.015 | 7.17 | 66.58±0.39 | 0.781±0.002 | 10.2 |
| AFL [35] | 78.11±1.99 | 0.582±0.021 | 1.87 | 58.39±0.37 | 1.081±0.014 | 0.99 | 54.56±0.79 | 1.172±0.018 | 3.50 | 77.32±0.15 | 0.703±0.001 | 1.86 |
| DRFA-Prox [16] | 78.34±1.46 | 0.532±0.034 | 1.85 | 58.62±0.16 | 1.096±0.037 | 1.26 | 54.61±0.76 | 1.151±0.039 | 4.69 | 77.95±0.51 | 0.702±0.007 | 1.34 |
| ASPIRE-EASE | **79.16±1.13** | **0.515±0.019** | **1.02** | 59.43±0.44 | 1.053±0.010 | 0.82 | 56.31±0.29 | 1.127±0.021 | **3.16** | **78.82±0.07** | **0.696±0.004** | **1.01** |
| ASPIRE-EASE$_{per}$ | 78.94±1.27 | 0.521±0.023 | 1.36 | **59.54±0.21** | **1.051±0.016** | **0.79** | **56.71±0.16** | **1.119±0.028** | 3.48 | 78.73±0.06 | 0.698±0.006 | 1.09 |

$[\text{Acc}_1, \cdots, \text{Acc}_N]$ (the test accuracy on every worker). In the experiment, $S$ is set as 1, that means the master will make an update once it receives a message. Each experiment is repeated 10 times, both mean and standard deviations are reported. We implement our model with PyTorch and conduct all the experiments on a server with two TITAN V GPUs.

## 6.2  Results

**Robustness against Data Heterogeneity**.   We first assess the robustness of the proposed ASPIRE-EASE by comparing it with baseline methods when data are heterogeneously distributed across different workers. Specifically, we compare the $\mathbf{Acc}_w$, $\mathbf{Loss}_w$ and $\mathbf{Std}$ of different methods on all datasets. The performance comparison results are shown in Table 1. In this table, we can observe that max{Ind$_j$}, which represents the best performance of individual training over all workers, exhibits the worst robustness on SHL, Person Activity, and SC-MA. This is because individual training (max{Ind$_j$}) only learns from the data in its local worker and cannot generalize to other workers due to different data distributions. Note that max{Ind$_j$} is unavailable for Fashion MNIST since each worker only contains one class of data and cross entropy loss cannot be used in this case. max{Ind$_j$} also does not have $\mathbf{Loss}_w$, since Ind$_j$ is trained only on individual worker $j$. The FedAvg and Mix$_{Even}$ exhibit better performance than max{Ind$_j$} since they consider the data from all workers. Nevertheless, FedAvg and Mix$_{Even}$ only assign the fixed weight for each worker. AFL is more robust than FedAvg and Mix$_{Even}$ since it not only utilizes the data from all workers but also considers optimizing the weight of each worker. DRFA-Prox outperforms AFL since it also considers the prior distribution and regards it as a regularizer in the objective function. Finally, we can observe that the proposed ASPIRE-EASE shows excellent robustness, which can be attributed to two factors: 1) ASPIRE-EASE considers data from all workers and can optimize the weight of each worker; 2) compared with DRFA-Prox which uses prior distribution as a regularizer, the prior distribution is incorporated within the constraint in our formulation (Eq. 4), which can be leveraged more effectively. And it is seen that ASPIRE-EASE can perform periodic communication since ASPIRE-EASE$_{per}$, which represents ASPIRE-EASE with periodic communication, also has excellent performance.

Within ASPIRE-EASE, the level of robustness can be controlled by adjusting $\Gamma$. Specially, when $\Gamma = 0$, we obtain a nominal optimization problem in which no adversarial distribution is considered. The size of the uncertainty set will increase with $\Gamma$ (when $\Gamma \leq N$), which enhances the adversarial robustness of the model. As shown in Figure 1, the robustness of ASPIRE-EASE can be gradually enhanced when $\Gamma$ increases. More results are available in Figure C2 of Appendix C.

**Robustness against Malicious Attacks**.   To assess the model robustness against malicious attacks, malicious workers with backdoor attacks [1, 48], which attempt to mislead the model training process, are added to the distributed system. Following [14], we report the success attack rate of backdoor attacks for comparison.  It can be calculated by checking how many instances in the backdoor dataset can be misled and categorized into the target labels. Lower success attack rates indicate more robustness against backdoor attacks. The comparison results are summarized in Table 2 and more detailed settings of backdoor attacks are available in Appendix C. In Table 2, we observe that AFL can be attacked easily since it could assign higher weights to malicious workers. Compared to AFL, FedAvg and Mix$_{Even}$ achieve relatively lower success attack rates since they assign equal weights to the malicious workers and other workers. DRFA-Prox can achieve even lower success attack rates since it can leverage the prior distribution to assign lower weights for malicious workers. The proposed ASPIRE-EASE achieves the lowest success attack rates since it can leverage the prior distribution more effectively. Specifically, it will assign lower weights to malicious workers with tight theoretical guarantees.

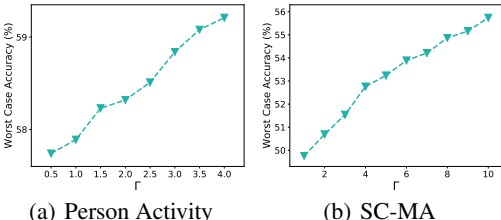

(a) Person Activity  (b) SC-MA

Figure 1: $\Gamma$ control the degree of robustness (worst case performance in the problem) on (a) Person Activity, (b) SC-MA datasets.

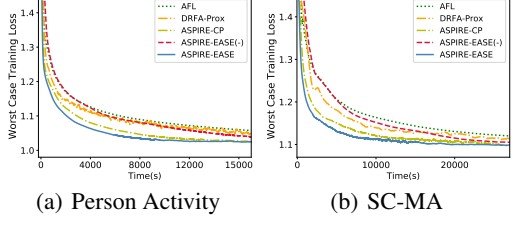

(a) Person Activity  (b) SC-MA

Figure 2: Comparison of the convergence time on worst case worker on (a) Person Activity, (b) SC-MA datasets.

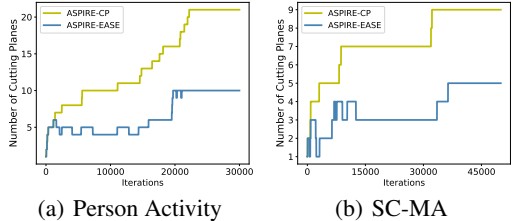

(a) Person Activity  (b) SC-MA

Figure 3: Comparison of ASPIRE-CP and ASPIRE-EASE regarding the number of cutting planes on (a) Person Activity, (b) SC-MA datasets.

Table 2: Performance comparisons about the success attack rate ($\%$) $\downarrow$. The boldfaced digits represent the best results.

| Model | SHL | Person Activity | SC-MA | Fashion MNIST |
|---|---|---|---|---|
| $\mathrm{Mix_{Even}}$ | 36.21±2.23 | 34.32±2.18 | 52.14±2.89 | 83.18±2.07 |
| FedAvg [33] | 38.15±3.02 | 33.25±2.49 | 55.39±3.13 | 82.04±1.84 |
| AFL [35] | 68.63±4.24 | 43.66±3.87 | 75.81±4.03 | 90.04±2.52 |
| DRFA-Prox [16] | 21.23±3.63 | 27.27±3.31 | 30.79±3.65 | 63.24±2.47 |
| ASPIRE-EASE | **9.17±1.65** | **22.36±2.33** | **14.51±3.21** | **45.10±1.64** |

**Efficiency**. In Figure 2, we compare the convergence speed of the proposed ASPIRE-EASE with AFL and DRFA-Prox by considering different communication and computation delays for each worker. The proposed ASPIRE-EASE has two variants, ASPIRE-CP (ASPIRE with cutting plane method), ASPIRE-EASE(-)(ASPIRE-EASE without asynchronous setting). More results are available in Figure C3 of Appendix C. Based on the comparison, we can observe that the proposed ASPIRE-EASE generally converges faster than baseline methods and its two variants. This is because 1) compared with AFL, DRFA-Prox, and ASPIRE-EASE(-), ASPIRE-EASE is an asynchronous algorithm in which the master updates its parameters only after receiving the updates from active workers instead of all workers; 2) unlike DRFA-Prox, the master in ASPIRE-EASE only needs to communicate with active workers once per iteration; 3) compared with ASPIRE-CP, ASPIRE-EASE utilizes active set method instead of cutting plane method, which is more efficient. It is seen from Figure 2 that, the convergence speed of ASPIRE-EASE mainly benefits from the asynchronous setting.

**Ablation Study**. For ASPIRE, compared with cutting plane method, EASE is more efficient since it considers removing the inactive cutting planes. To demonstrate the efficiency of EASE, we firstly compare ASPIRE-EASE with ASPIRE-CP concerning the number of cutting planes used during the training. In Figure 3, we can observe that ASPIRE-EASE uses fewer cutting planes than ASPIRE-CP, thus is more efficient. The convergence speed of ASPIRE-EASE and ASPIRE-CP in Figure 2 also suggests that ASPIRE-EASE converges much faster than ASPIRE-CP. More results are available in Figure C3 and C4, Appendix C.

## 7  Conclusion

In this paper, we present ASPIRE-EASE method to effectively solve the distributed distributionally robust optimization problem with non-convex objectives. In addition, $CD$-norm uncertainty set has been proposed to effectively incorporate the prior distribution into the problem formulation, which allows for flexible adjustment of the degree of robustness of DRO. Theoretical analysis has also been conducted to analyze the convergence properties and the iteration complexity of ASPIRE-EASE. ASPIRE-EASE exhibits strong empirical performance on multiple real-world datasets and is effective in tackling DRO problems in a fully distributed and asynchronous manner. In the future work, more uncertainty sets could be designed for our framework and more update rule for variables in ASPIRE could be considered.

## Acknowledgments and Disclosure of Funding

The work of Yang Jiao and Kai Yang was supported in part by the Fundamental Research Funds for the Central Universities of China, in part by the Shenzhen Institute of Artificial Intelligence and Robotics for Society (AIRS), in part by the National Natural Science Foundation of China under Grant 61771013, and in part by the Fundamental Research Funds of Shanghai Jiading District.

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
