# Appendix

## A  Proof of Theorem 1

Before proceeding to the detailed proofs, we provide some notations for the clarity in presentation. We use notation $< \cdot, \cdot >$ to denote the inner product and we use $|| \cdot ||$ to denote the $l_2$-norm. $|\mathbf{A}^t|$ and $|\mathbf{Q}^{t+1}|$ respectively denote the number of cutting planes and active workers in $(t+1)^{\text{th}}$ iteration.

Then, we cover some Lemmas which are useful for the deduction of Theorem 1.

**Lemma 1** *Suppose Assumption 1 and 2 hold, $\forall t \geq T_1 + \tau$, we have,*

$$L_p(\{\boldsymbol{w}_j^{t+1}\}, \boldsymbol{z}^t, h^t, \{\lambda_l^t\}, \{\boldsymbol{\phi}_j^t\}) - L_p(\{\boldsymbol{w}_j^t\}, \boldsymbol{z}^t, h^t, \{\lambda_l^t\}, \{\boldsymbol{\phi}_j^t\})$$

$$\leq \sum_{j=1}^{N} (\tfrac{L+1}{2} - \tfrac{1}{\eta_{\boldsymbol{w}}^t}) ||\boldsymbol{w}_j^{t+1} - \boldsymbol{w}_j^t||^2 + \tfrac{3\tau k_1 N L^2}{2} (||\boldsymbol{z}^{t+1} - \boldsymbol{z}^t||^2 + ||h^{t+1} - h^t||^2 + \sum_{l=1}^{|\mathbf{A}^t|} ||\lambda_l^{t+1} - \lambda_l^t||^2),$$
(A.1)

$$L_p(\{\boldsymbol{w}_j^{t+1}\}, \boldsymbol{z}^{t+1}, h^t, \{\lambda_l^t\}, \{\boldsymbol{\phi}_j^t\}) - L_p(\{\boldsymbol{w}_j^{t+1}\}, \boldsymbol{z}^t, h^t, \{\lambda_l^t\}, \{\boldsymbol{\phi}_j^t\}) \leq (\tfrac{L}{2} - \tfrac{1}{\eta_{\boldsymbol{z}}^t}) ||\boldsymbol{z}^{t+1} - \boldsymbol{z}^t||^2,$$
(A.2)

$$L_p(\{\boldsymbol{w}_j^{t+1}\}, \boldsymbol{z}^{t+1}, h^{t+1}, \{\lambda_l^t\}, \{\boldsymbol{\phi}_j^t\}) - L_p(\{\boldsymbol{w}_j^{t+1}\}, \boldsymbol{z}^{t+1}, h^t, \{\lambda_l^t\}, \{\boldsymbol{\phi}_j^t\}) \leq (\tfrac{L}{2} - \tfrac{1}{\eta_h^t}) ||h^{t+1} - h^t||^2.$$
(A.3)

*Proof of Lemma 1:*

According to Assumption 1, we have,

$$L_p(\{\boldsymbol{w}_1^{t+1}, \boldsymbol{w}_2^t, \cdots, \boldsymbol{w}_N^t\}, \boldsymbol{z}^t, h^t, \{\lambda_l^t\}, \{\boldsymbol{\phi}_j^t\}) - L_p(\{\boldsymbol{w}_j^t\}, \boldsymbol{z}^t, h^t, \{\lambda_l^t\}, \{\boldsymbol{\phi}_j^t\})$$

$$\leq \langle \nabla_{\boldsymbol{w}_1} L_p(\{\boldsymbol{w}_j^t\}, \boldsymbol{z}^t, h^t, \{\lambda_l^t\}, \{\boldsymbol{\phi}_j^t\}), \boldsymbol{w}_1^{t+1} - \boldsymbol{w}_1^t \rangle + \tfrac{L}{2} ||\boldsymbol{w}_1^{t+1} - \boldsymbol{w}_1^t||^2,$$

$$L_p(\{\boldsymbol{w}_1^{t+1}, \boldsymbol{w}_2^{t+1}, \boldsymbol{w}_3^t, \cdots, \boldsymbol{w}_N^t\}, \boldsymbol{z}^t, h^t, \{\lambda_l^t\}, \{\boldsymbol{\phi}_j^t\}) - L_p(\{\boldsymbol{w}_1^{t+1}, \boldsymbol{w}_2^t, \cdots, \boldsymbol{w}_N^t\}, \boldsymbol{z}^t, h^t, \{\lambda_l^t\}, \{\boldsymbol{\phi}_j^t\})$$

$$\leq \langle \nabla_{\boldsymbol{w}_2} L_p(\{\boldsymbol{w}_j^t\}, \boldsymbol{z}^t, h^t, \{\lambda_l^t\}, \{\boldsymbol{\phi}_j^t\}), \boldsymbol{w}_2^{t+1} - \boldsymbol{w}_2^t \rangle + \tfrac{L}{2} ||\boldsymbol{w}_2^{t+1} - \boldsymbol{w}_2^t||^2,$$

$$\vdots$$

$$L_p(\{\boldsymbol{w}_j^{t+1}\}, \boldsymbol{z}^t, h^t, \{\lambda_l^t\}, \{\boldsymbol{\phi}_j^t\}) - L_p(\{\boldsymbol{w}_1^{t+1}, \cdots, \boldsymbol{w}_{N-1}^{t+1}, \boldsymbol{w}_N^t\}, \boldsymbol{z}^t, h^t, \{\lambda_l^t\}, \{\boldsymbol{\phi}_j^t\})$$

$$\leq \langle \nabla_{\boldsymbol{w}_N} L_p(\{\boldsymbol{w}_j^t\}, \boldsymbol{z}^t, h^t, \{\lambda_l^t\}, \{\boldsymbol{\phi}_j^t\}), \boldsymbol{w}_N^{t+1} - \boldsymbol{w}_N^t \rangle + \tfrac{L}{2} ||\boldsymbol{w}_N^{t+1} - \boldsymbol{w}_N^t||^2.$$
(A.4)

Summing up the above inequalities in Eq. (A.4), we have,

$$L_p(\{\boldsymbol{w}_j^{t+1}\}, \boldsymbol{z}^t, h^t, \{\lambda_l^t\}, \{\boldsymbol{\phi}_j^t\}) - L_p(\{\boldsymbol{w}_j^t\}, \boldsymbol{z}^t, h^t, \{\lambda_l^t\}, \{\boldsymbol{\phi}_j^t\})$$

$$\leq \sum_{j=1}^{N} (\langle \nabla_{\boldsymbol{w}_j} L_p(\{\boldsymbol{w}_j^t\}, \boldsymbol{z}^t, h^t, \{\lambda_l^t\}, \{\boldsymbol{\phi}_j^t\}), \boldsymbol{w}_j^{t+1} - \boldsymbol{w}_j^t \rangle + \tfrac{L}{2} ||\boldsymbol{w}_j^{t+1} - \boldsymbol{w}_j^t||^2).$$
(A.5)

According to $\nabla_{\boldsymbol{w}_j} L_p(\{\boldsymbol{w}_j^t\}, \boldsymbol{z}^t, h^t, \{\lambda_l^t\}, \{\boldsymbol{\phi}_j^t\}) = \nabla_{\boldsymbol{w}_j} \widetilde{L}_p(\{\boldsymbol{w}_j^t\}, \boldsymbol{z}^t, h^t, \{\lambda_l^t\}, \{\boldsymbol{\phi}_j^t\})$ and the optimal condition for Eq. (11), for active nodes, *i.e.*, $\forall j \in \mathbf{Q}^{t+1}, \forall t \geq T_1 + \tau$, we have,

$$\left\langle \boldsymbol{w}_j^t - \boldsymbol{w}_j^{t+1}, \boldsymbol{w}_j^{t+1} - \boldsymbol{w}_j^t + \underline{\eta_{\boldsymbol{w}}} \nabla_{\boldsymbol{w}_j} L_p(\{\boldsymbol{w}_j^{\widetilde{t}_j}\}, \boldsymbol{z}^{\widetilde{t}_j}, h^{\widetilde{t}_j}, \{\lambda_l^{\widetilde{t}_j}\}, \{\boldsymbol{\phi}_j^{\widetilde{t}_j}\}) \right\rangle \geq 0.$$
(A.6)

According to Eq. (A.6), $\forall t \geq T_1 + \tau$, we have,

$$\left\langle \boldsymbol{w}_j^{t+1} - \boldsymbol{w}_j^t, \nabla_{\boldsymbol{w}_j} L_p(\{\boldsymbol{w}_j^{\widetilde{t}_j}\}, \boldsymbol{z}^{\widetilde{t}_j}, h^{\widetilde{t}_j}, \{\lambda_l^{\widetilde{t}_j}\}, \{\boldsymbol{\phi}_j^{\widetilde{t}_j}\}) \right\rangle \leq -\frac{1}{\underline{\eta_{\boldsymbol{w}}}} ||\boldsymbol{w}_j^{t+1} - \boldsymbol{w}_j^t||^2 \leq -\frac{1}{\eta_{\boldsymbol{w}}^t} ||\boldsymbol{w}_j^{t+1} - \boldsymbol{w}_j^t||^2.$$
(A.7)

And according to the Cauchy-Schwarz inequality, Assumption 1 and 2, we can get,

$$\left\langle \boldsymbol{w}_j^{t+1}-\boldsymbol{w}_j^t, \nabla_{\boldsymbol{w}_j} L_p(\{\boldsymbol{w}_j^t\}, \boldsymbol{z}^t, h^t, \{\lambda_l^t\}, \{\boldsymbol{\phi}_j^t\}) - \nabla_{\boldsymbol{w}_j} L_p(\{\boldsymbol{w}_j^{\widetilde{t}_j}\}, \boldsymbol{z}^{\widetilde{t}_j}, h^{\widetilde{t}_j}, \{\lambda_l^{\widetilde{t}_j}\}, \{\boldsymbol{\phi}_j^{\widetilde{t}_j}\}) \right\rangle$$

$$\leq \tfrac{1}{2}||\boldsymbol{w}_j^{t+1}-\boldsymbol{w}_j^t||^2 + \tfrac{L^2}{2}(||\boldsymbol{z}^t-\boldsymbol{z}^{\widetilde{t}_j}||^2 + ||h^t-h^{\widetilde{t}_j}||^2 + \sum_{l=1}^{|\mathbf{A}^t|}||\lambda_l^t-\lambda_l^{\widetilde{t}_j}||^2)$$

$$\leq \tfrac{1}{2}||\boldsymbol{w}_j^{t+1}-\boldsymbol{w}_j^t||^2 + \tfrac{3\tau k_1 L^2}{2}(||\boldsymbol{z}^{t+1}-\boldsymbol{z}^t||^2 + ||h^{t+1}-h^t||^2 + \sum_{l=1}^{|\mathbf{A}^t|}||\lambda_l^{t+1}-\lambda_l^t||^2).$$

(A.8)

Combining the above Eq. (A.5), (A.7) with Eq. (A.8), we can obtain Eq. (A.1), that is,

$$L_p(\{\boldsymbol{w}_j^{t+1}\}, \boldsymbol{z}^t, h^t, \{\lambda_l^t\}, \{\boldsymbol{\phi}_j^t\}) - L_p(\{\boldsymbol{w}_j^t\}, \boldsymbol{z}^t, h^t, \{\lambda_l^t\}, \{\boldsymbol{\phi}_j^t\})$$

$$\leq \sum_{j=1}^N (\tfrac{L+1}{2} - \tfrac{1}{\eta_{\boldsymbol{w}}^t})||\boldsymbol{w}_j^{t+1}-\boldsymbol{w}_j^t||^2 + \tfrac{3\tau k_1 N L^2}{2}(||\boldsymbol{z}^{t+1}-\boldsymbol{z}^t||^2 + ||h^{t+1}-h^t||^2 + \sum_{l=1}^{|\mathbf{A}^t|}||\lambda_l^{t+1}-\lambda_l^t||^2).$$

Following Assumption 1, we have,

$$L_p(\{\boldsymbol{w}_j^{t+1}\}, \boldsymbol{z}^{t+1}, h^t, \{\lambda_l^t\}, \{\boldsymbol{\phi}_j^t\}) - L_p(\{\boldsymbol{w}_j^{t+1}\}, \boldsymbol{z}^t, h^t, \{\lambda_l^t\}, \{\boldsymbol{\phi}_j^t\})$$

$$\leq \left\langle \nabla_{\boldsymbol{z}} L_p(\{\boldsymbol{w}_j^{t+1}\}, \boldsymbol{z}^t, h^t, \{\lambda_l^t\}, \{\boldsymbol{\phi}_j^t\}), \boldsymbol{z}^{t+1}-\boldsymbol{z}^t \right\rangle + \tfrac{L}{2}||\boldsymbol{z}^{t+1}-\boldsymbol{z}^t||^2.$$

(A.9)

According to $\nabla_{\boldsymbol{z}} L_p(\{\boldsymbol{w}_j^{t+1}\}, \boldsymbol{z}^t, h^t, \{\lambda_l^t\}, \{\boldsymbol{\phi}_j^t\}) = \nabla_{\boldsymbol{z}} \widetilde{L}_p(\{\boldsymbol{w}_j^{t+1}\}, \boldsymbol{z}^t, h^t, \{\lambda_l^t\}, \{\boldsymbol{\phi}_j^t\})$ and the optimal condition for Eq. (12), we have,

$$\left\langle \boldsymbol{z}^t-\boldsymbol{z}^{t+1}, \boldsymbol{z}^{t+1}-\boldsymbol{z}^t + \eta_{\boldsymbol{z}}^t \nabla_{\boldsymbol{z}} L_p(\{\boldsymbol{w}_j^{t+1}\}, \boldsymbol{z}^t, h^t, \{\lambda_l^t\}, \{\boldsymbol{\phi}_j^t\}) \right\rangle \geq 0. \qquad (A.10)$$

Combining Eq. (A.9) with Eq. (A.10), we can obtain the Eq. (A.2), that is,

$$L_p(\{\boldsymbol{w}_j^{t+1}\}, \boldsymbol{z}^{t+1}, h^t, \{\lambda_l^t\}, \{\boldsymbol{\phi}_j^t\}) - L_p(\{\boldsymbol{w}_j^{t+1}\}, \boldsymbol{z}^t, h^t, \{\lambda_l^t\}, \{\boldsymbol{\phi}_j^t\}) \leq (\tfrac{L}{2} - \tfrac{1}{\eta_{\boldsymbol{z}}^t})||\boldsymbol{z}^{t+1}-\boldsymbol{z}^t||^2.$$

According to Assumption 1, we have:

$$L_p(\{\boldsymbol{w}_j^{t+1}\}, \boldsymbol{z}^{t+1}, h^{t+1}, \{\lambda_l^t\}, \{\boldsymbol{\phi}_j^t\}) - L_p(\{\boldsymbol{w}_j^{t+1}\}, \boldsymbol{z}^{t+1}, h^t, \{\lambda_l^t\}, \{\boldsymbol{\phi}_j^t\})$$

$$\leq \left\langle \nabla_h L_p(\{\boldsymbol{w}_j^{t+1}\}, \boldsymbol{z}^{t+1}, h^t, \{\lambda_l^t\}, \{\boldsymbol{\phi}_j^t\}), h^{t+1}-h^t \right\rangle + \tfrac{L}{2}||h^{t+1}-h^t||^2.$$

(A.11)

According to $\nabla_h L_p(\{\boldsymbol{w}_j^{t+1}\}, \boldsymbol{z}^{t+1}, h^t, \{\lambda_l^t\}, \{\boldsymbol{\phi}_j^t\}) = \nabla_h \widetilde{L}_p(\{\boldsymbol{w}_j^{t+1}\}, \boldsymbol{z}^{t+1}, h^t, \{\lambda_l^t\}, \{\boldsymbol{\phi}_j^t\})$ and the optimal condition for Eq. (13), we have:

$$\left\langle h^t-h^{t+1}, h^{t+1}-h^t + \eta_h^t \nabla_h L_p(\{\boldsymbol{w}_j^{t+1}\}, \boldsymbol{z}^{t+1}, h^t, \{\lambda_l^t\}, \{\boldsymbol{\phi}_j^t\}) \right\rangle \geq 0. \qquad (A.12)$$

Combining Eq. (A.11) with Eq. (A.12), we can show that,

$$L_p(\{\boldsymbol{w}_j^{t+1}\}, \boldsymbol{z}^{t+1}, h^{t+1}, \{\lambda_l^t\}, \{\boldsymbol{\phi}_j^t\}) - L_p(\{\boldsymbol{w}_j^{t+1}\}, \boldsymbol{z}^{t+1}, h^t, \{\lambda_l^t\}, \{\boldsymbol{\phi}_j^t\}) \leq (\tfrac{L}{2} - \tfrac{1}{\eta_h^t})||h^{t+1}-h^t||^2.$$

**Lemma 2** *Suppose Assumption 1 and 2 hold, $\forall t \geq T_1+\tau$, we have:*

$$L_p(\{\boldsymbol{w}_j^{t+1}\}, \boldsymbol{z}^{t+1}, h^{t+1}, \{\lambda_l^{t+1}\}, \{\boldsymbol{\phi}_j^{t+1}\}) - L_p(\{\boldsymbol{w}_j^t\}, \boldsymbol{z}^t, h^t, \{\lambda_l^t\}, \{\boldsymbol{\phi}_j^t\})$$

$$\leq (\tfrac{L+1}{2} - \tfrac{1}{\eta_{\boldsymbol{w}}^t} + \tfrac{|\mathbf{A}^t|L^2}{2a_1} + \tfrac{|\mathbf{Q}^{t+1}|L^2}{2a_3}) \sum_{j=1}^N ||\boldsymbol{w}_j^{t+1}-\boldsymbol{w}_j^t||^2 + (\tfrac{L+3\tau k_1 N L^2}{2} - \tfrac{1}{\eta_{\boldsymbol{z}}^t} + \tfrac{|\mathbf{A}^t|L^2}{2a_1} + \tfrac{|\mathbf{Q}^{t+1}|L^2}{2a_3})||\boldsymbol{z}^{t+1}-\boldsymbol{z}^t||^2$$

$$+ (\tfrac{L+3\tau k_1 N L^2}{2} - \tfrac{1}{\eta_h^t} + \tfrac{|\mathbf{A}^t|L^2}{2a_1} + \tfrac{|\mathbf{Q}^{t+1}|L^2}{2a_3})||h^{t+1}-h^t||^2 + (\tfrac{a_1+3\tau k_1 N L^2}{2} - \tfrac{c_1^{t-1}-c_1^t}{2} + \tfrac{1}{2\rho_1})\sum_{l=1}^{|\mathbf{A}^t|}||\lambda_l^{t+1}-\lambda_l^t||^2$$

$$+ \tfrac{c_1^{t-1}}{2}\sum_{l=1}^{|\mathbf{A}^t|}(||\lambda_l^{t+1}||^2 - ||\lambda_l^t||^2) + \tfrac{1}{2\rho_1}\sum_{l=1}^{|\mathbf{A}^t|}||\lambda_l^t-\lambda_l^{t-1}||^2 + (\tfrac{a_3}{2} - \tfrac{c_2^{t-1}-c_2^t}{2} + \tfrac{1}{2\rho_2})\sum_{j=1}^N ||\boldsymbol{\phi}_j^{t+1}-\boldsymbol{\phi}_j^t||^2$$

$$+ \tfrac{c_2^{t-1}}{2}\sum_{j=1}^N (||\boldsymbol{\phi}_j^{t+1}||^2 - ||\boldsymbol{\phi}_j^t||^2) + \tfrac{1}{2\rho_2}\sum_{j=1}^N ||\boldsymbol{\phi}_j^t-\boldsymbol{\phi}_j^{t-1}||^2,$$

(A.13)

*where $a_1 > 0$ and $a_3 > 0$ are constants.*

### *Proof of Lemma 2*:

First of all, at $(t+1)^{\text{th}}$ iteration, the following equations hold and will be used in the derivation:

$$\sum_{j=1}^{N}||\phi_j^{t+1}-\phi_j^t||^2 = \sum_{j\in\mathbf{Q}^{t+1}}||\phi_j^{t+1}-\phi_j^t||^2, \ \sum_{j=1}^{N}(||\phi_j^{t+1}||^2-||\phi_j^t||^2) = \sum_{j\in\mathbf{Q}^{t+1}}(||\phi_j^{t+1}||^2-||\phi_j^t||^2).$$

According to Eq. (14), in $(t+1)^{\text{th}}$ iteration, $\forall\lambda\in\mathbf{\Lambda}$, it follows that:

$$\left\langle \lambda_l^{t+1}-\lambda_l^t-\rho_1\nabla_{\lambda_l}\widetilde{L}_p(\{w_j^{t+1}\},z^{t+1},h^{t+1},\{\lambda_l^t\},\{\phi_j^t\}), \lambda-\lambda_l^{t+1}\right\rangle \geq 0. \qquad \text{(A.14)}$$

Let $\lambda = \lambda_l^t$, we can obtain:

$$\left\langle \nabla_{\lambda_l}\widetilde{L}_p(\{w_j^{t+1}\},z^{t+1},h^{t+1},\{\lambda_l^t\},\{\phi_j^t\})-\frac{1}{\rho_1}(\lambda_l^{t+1}-\lambda_l^t), \lambda_l^t-\lambda_l^{t+1}\right\rangle \leq 0. \qquad \text{(A.15)}$$

Likewise, in $t^{\text{th}}$ iteration, we can obtain:

$$\left\langle \nabla_{\lambda_l}\widetilde{L}_p(\{w_j^t\},z^t,h^t,\{\lambda_l^{t-1}\},\{\phi_j^{t-1}\})-\frac{1}{\rho_1}(\lambda_l^t-\lambda_l^{t-1}), \lambda_l^{t+1}-\lambda_l^t\right\rangle \leq 0. \qquad \text{(A.16)}$$

$\forall t \geq T_1$, since $\widetilde{L}_p(\{w_j\},z,h,\{\lambda_l\},\{\phi_j\})$ is concave with respect to $\lambda_l$, we have,

$$\widetilde{L}_p(\{w_j^{t+1}\},z^{t+1},h^{t+1},\{\lambda_l^{t+1}\},\{\phi_j^t\}) - \widetilde{L}_p(\{w_j^{t+1}\},z^{t+1},h^{t+1},\{\lambda_l^t\},\{\phi_j^t\})$$

$$\leq \sum_{l=1}^{|\mathbf{A}^t|}\left\langle \nabla_{\lambda_l}\widetilde{L}_p(\{w_j^{t+1}\},z^{t+1},h^{t+1},\{\lambda_l^t\},\{\phi_j^t\}), \lambda_l^{t+1}-\lambda_l^t\right\rangle$$

$$\leq \sum_{l=1}^{|\mathbf{A}^t|}\left(\left\langle \nabla_{\lambda_l}\widetilde{L}_p(\{w_j^{t+1}\},z^{t+1},h^{t+1},\{\lambda_l^t\},\{\phi_j^t\})-\nabla_{\lambda_l}\widetilde{L}_p(\{w_j^t\},z^t,h^t,\{\lambda_l^{t-1}\},\{\phi_j^{t-1}\}), \lambda_l^{t+1}-\lambda_l^t\right\rangle\right.$$

$$\left.+\frac{1}{\rho_1}\left\langle \lambda_l^t-\lambda_l^{t-1}, \lambda_l^{t+1}-\lambda_l^t\right\rangle\right). \qquad \text{(A.17)}$$

Denoting $v_{1,l}^{t+1} = \lambda_l^{t+1}-\lambda_l^t-(\lambda_l^t-\lambda_l^{t-1})$, we have,

$$\sum_{l=1}^{|\mathbf{A}^t|}\left\langle \nabla_{\lambda_l}\widetilde{L}_p(\{w_j^{t+1}\},z^{t+1},h^{t+1},\{\lambda_l^t\},\{\phi_j^t\})-\nabla_{\lambda_l}\widetilde{L}_p(\{w_j^t\},z^t,h^t,\{\lambda_l^{t-1}\},\{\phi_j^{t-1}\}), \lambda_l^{t+1}-\lambda_l^t\right\rangle$$

$$=\sum_{l=1}^{|\mathbf{A}^t|}\left\langle \nabla_{\lambda_l}\widetilde{L}_p(\{w_j^{t+1}\},z^{t+1},h^{t+1},\{\lambda_l^t\},\{\phi_j^t\})-\nabla_{\lambda_l}\widetilde{L}_p(\{w_j^t\},z^t,h^t,\{\lambda_l^t\},\{\phi_j^t\}), \lambda_l^{t+1}-\lambda_l^t\right\rangle (1a)$$

$$+\sum_{l=1}^{|\mathbf{A}^t|}\left\langle \nabla_{\lambda_l}\widetilde{L}_p(\{w_j^t\},z^t,h^t,\{\lambda_l^t\},\{\phi_j^t\})-\nabla_{\lambda_l}\widetilde{L}_p(\{w_j^t\},z^t,h^t,\{\lambda_l^{t-1}\},\{\phi_j^{t-1}\}), v_{1,l}^{t+1}\right\rangle (1b)$$

$$+\sum_{l=1}^{|\mathbf{A}^t|}\left\langle \nabla_{\lambda_l}\widetilde{L}_p(\{w_j^t\},z^t,h^t,\{\lambda_l^t\},\{\phi_j^t\})-\nabla_{\lambda_l}\widetilde{L}_p(\{w_j^t\},z^t,h^t,\{\lambda_l^{t-1}\},\{\phi_j^{t-1}\}), \lambda_l^t-\lambda_l^{t-1}\right\rangle (1c).$$

$$\text{(A.18)}$$

Firstly, we focus on the $(1a)$ in Eq. (A.18), we can write $(1a)$ as:

$$\left\langle \nabla_{\lambda_l}\widetilde{L}_p(\{w_j^{t+1}\},z^{t+1},h^{t+1},\{\lambda_l^t\},\{\phi_j^t\})-\nabla_{\lambda_l}\widetilde{L}_p(\{w_j^t\},z^t,h^t,\{\lambda_l^t\},\{\phi_j^t\}), \lambda_l^{t+1}-\lambda_l^t\right\rangle$$

$$=\left\langle \nabla_{\lambda_l}L_p(\{w_j^{t+1}\},z^{t+1},h^{t+1},\{\lambda_l^t\},\{\phi_j^t\})-\nabla_{\lambda_l}L_p(\{w_j^t\},z^t,h^t,\{\lambda_l^t\},\{\phi_j^t\}), \lambda_l^{t+1}-\lambda_l^t\right\rangle$$

$$+(c_1^{t-1}-c_1^t)\left\langle \lambda_l^t, \lambda_l^{t+1}-\lambda_l^t\right\rangle$$

$$=\left\langle \nabla_{\lambda_l}L_p(\{w_j^{t+1}\},z^{t+1},h^{t+1},\{\lambda_l^t\},\{\phi_j^t\})-\nabla_{\lambda_l}L_p(\{w_j^t\},z^t,h^t,\{\lambda_l^t\},\{\phi_j^t\}), \lambda_l^{t+1}-\lambda_l^t\right\rangle$$

$$+\frac{c_1^{t-1}-c_1^t}{2}(||\lambda_l^{t+1}||^2-||\lambda_l^t||^2)-\frac{c_1^{t-1}-c_1^t}{2}||\lambda_l^{t+1}-\lambda_l^t||^2. $$

$$\text{(A.19)}$$

And according to Cauchy-Schwarz inequality and Assumption 1, we can obtain,

$$\left\langle \nabla_{\lambda_l} L_p(\{w_j^{t+1}\}, z^{t+1}, h^{t+1}, \{\lambda_l^t\}, \{\phi_j^t\}) - \nabla_{\lambda_l} L_p(\{w_j^t\}, z^t, h^t, \{\lambda_l^t\}, \{\phi_j^t\}), \lambda_l^{t+1} - \lambda_l^t \right\rangle$$

$$\leq \frac{L^2}{2a_1} \left( \sum_{j=1}^{N} ||w_j^{t+1} - w_j^t||^2 + ||z^{t+1} - z^t||^2 + ||h^{t+1} - h^t||^2 \right) + \frac{a_1}{2} ||\lambda_l^{t+1} - \lambda_l^t||^2,$$
(A.20)

where $a_1 > 0$ is a constant. Combining Eq. (A.19) with Eq. (A.20), we can obtain the upper bound of $(1a)$, that is,

$$\sum_{l=1}^{|\mathbf{A}^t|} \left\langle \nabla_{\lambda_l} \widetilde{L}_p(\{w_j^{t+1}\}, z^{t+1}, h^{t+1}, \{\lambda_l^t\}, \{\phi_j^t\}) - \nabla_{\lambda_l} \widetilde{L}_p(\{w_j^t\}, z^t, h^t, \{\lambda_l^t\}, \{\phi_j^t\}), \lambda_l^{t+1} - \lambda_l^t \right\rangle$$

$$\leq \sum_{l=1}^{|\mathbf{A}^t|} \left( \frac{L^2}{2a_1} \left( \sum_{j=1}^{N} ||w_j^{t+1} - w_j^t||^2 + ||z^{t+1} - z^t||^2 + ||h^{t+1} - h^t||^2 \right) + \frac{a_1}{2} ||\lambda_l^{t+1} - \lambda_l^t||^2 \right.$$

$$\left. + \frac{c_1^{t-1} - c_1^t}{2} (||\lambda_l^{t+1}||^2 - ||\lambda_l^t||^2) - \frac{c_1^{t-1} - c_1^t}{2} ||\lambda_l^{t+1} - \lambda_l^t||^2 \right).$$
(A.21)

Secondly, we focus on the $(1b)$ in Eq. (A.18). According to Cauchy-Schwarz inequality we can write the $(1b)$ as,

$$\sum_{l=1}^{|\mathbf{A}^t|} \left\langle \nabla_{\lambda_l} \widetilde{L}_p(\{w_j^t\}, z^t, h^t, \{\lambda_l^t\}, \{\phi_j^t\}) - \nabla_{\lambda_l} \widetilde{L}_p(\{w_j^t\}, z^t, h^t, \{\lambda_l^{t-1}\}, \{\phi_j^{t-1}\}), v_{1,l}^{t+1} \right\rangle$$

$$\leq \sum_{l=1}^{|\mathbf{A}^t|} \left( \frac{a_2}{2} ||\nabla_{\lambda_l} \widetilde{L}_p(\{w_j^t\}, z^t, h^t, \{\lambda_l^t\}, \{\phi_j^t\}) - \nabla_{\lambda_l} \widetilde{L}_p(\{w_j^t\}, z^t, h^t, \{\lambda_l^{t-1}\}, \{\phi_j^{t-1}\})||^2 + \frac{1}{2a_2} ||v_{1,l}^{t+1}||^2 \right),$$
(A.22)

where $a_2 > 0$ is a constant. Then, we focus on the $(1c)$ in Eq. (A.18). Firstly, $\forall \lambda_l$, we have,

$$||\nabla_{\lambda_l} \widetilde{L}_p(\{w_j^t\}, z^t, h^t, \{\lambda_l^t\}, \{\phi_j^t\}) - \nabla_{\lambda_l} \widetilde{L}_p(\{w_j^t\}, z^t, h^t, \{\lambda_l^{t-1}\}, \{\phi_j^{t-1}\})||$$

$$= ||\nabla_{\lambda_l} L_p(\{w_j^t\}, z^t, h^t, \{\lambda_l^t\}, \{\phi_j^t\}) - \nabla_{\lambda_l} L_p(\{w_j^t\}, z^t, h^t, \{\lambda_l^{t-1}\}, \{\phi_j^t\}) - c_1^{t-1}(\lambda_l^t - \lambda_l^{t-1})||$$

$$\leq (L + c_1^{t-1}) ||\lambda_l^t - \lambda_l^{t-1}||,$$
(A.23)

where the last inequality comes from Assumption 1 and the trigonometric inequality. Denoting $L_1' = L + c_1^0$, we can obtain,

$$||\nabla_{\lambda_l} \widetilde{L}_p(\{w_j^t\}, z^t, h^t, \{\lambda_l^t\}, \{\phi_j^t\}) - \nabla_{\lambda_l} \widetilde{L}_p(\{w_j^t\}, z^t, h^t, \{\lambda_l^{t-1}\}, \{\phi_j^{t-1}\})|| \leq L_1' ||\lambda_l^t - \lambda_l^{t-1}||.$$
(A.24)

Following from Eq. (A.24) and the strong concavity of $\widetilde{L}_p(\{w_j\}, z, h, \{\lambda_l\}, \{\phi_j\})$ *w.r.t* $\lambda_l$ [37, 52], we can obtain the upper bound of $(1c)$:

$$\sum_{l=1}^{|\mathbf{A}^t|} \left\langle \nabla_{\lambda_l} \widetilde{L}_p(\{w_j^t\}, z^t, h^t, \{\lambda_l^t\}, \{\phi_j^t\}) - \nabla_{\lambda_l} \widetilde{L}_p(\{w_j^t\}, z^t, h^t, \{\lambda_l^{t-1}\}, \{\phi_j^{t-1}\}), \lambda_l^t - \lambda_l^{t-1} \right\rangle$$

$$\leq \sum_{l=1}^{|\mathbf{A}^t|} \left( -\frac{1}{L_1' + c_1^{t-1}} ||\nabla_{\lambda_l} \widetilde{L}_p(\{w_j^t\}, z^t, h^t, \{\lambda_l^t\}, \{\phi_j^t\}) - \nabla_{\lambda_l} \widetilde{L}_p(\{w_j^t\}, z^t, h^t, \{\lambda_l^{t-1}\}, \{\phi_j^{t-1}\})||^2 \right.$$

$$\left. - \frac{c_1^{t-1} L_1'}{L_1' + c_1^{t-1}} ||\lambda_l^t - \lambda_l^{t-1}||^2 \right).$$
(A.25)

In addition, the following inequality can be obtained,

$$\frac{1}{\rho_1} \left\langle \lambda_l^t - \lambda_l^{t-1}, \lambda_l^{t+1} - \lambda_l^t \right\rangle \leq \frac{1}{2\rho_1} ||\lambda_l^{t+1} - \lambda_l^t||^2 - \frac{1}{2\rho_1} ||v_{1,l}^{t+1}||^2 + \frac{1}{2\rho_1} ||\lambda_l^t - \lambda_l^{t-1}||^2.$$
(A.26)

Combining Eq. (A.17), (A.18), (A.21), (A.22), (A.25), (A.26), $\frac{\rho_1}{2} \leq \frac{1}{L_1' + c_1^0}$, and setting $a_2 = \rho_1$, we have:

$$L_p(\{\boldsymbol{w}_j^{t+1}\}, \boldsymbol{z}^{t+1}, h^{t+1}, \{\lambda_l^{t+1}\}, \{\boldsymbol{\phi}_j^t\}) - L_p(\{\boldsymbol{w}_j^{t+1}\}, \boldsymbol{z}^{t+1}, h^{t+1}, \{\lambda_l^t\}, \{\boldsymbol{\phi}_j^t\})$$

$$\leq \sum_{l=1}^{|\mathbf{A}^t|} (\langle \nabla_{\lambda_l} \widetilde{L}_p(\{\boldsymbol{w}_j^{t+1}\}, \boldsymbol{z}^{t+1}, h^{t+1}, \{\lambda_l^t\}, \{\boldsymbol{\phi}_j^t\}) - \nabla_{\lambda_l} \widetilde{L}_p(\{\boldsymbol{w}_j^t\}, \boldsymbol{z}^t, h^t, \{\lambda_l^{t-1}\}, \{\boldsymbol{\phi}_j^{t-1}\}), \lambda_l^{t+1} - \lambda_l^t \rangle$$

$$+ \frac{1}{\rho_1} \langle \lambda_l^t - \lambda_l^{t-1}, \lambda_l^{t+1} - \lambda_l^t \rangle + \frac{c_1^t}{2} (||\lambda_l^{t+1}||^2 - ||\lambda_l^t||^2))$$

$$\leq \sum_{l=1}^{|\mathbf{A}^t|} (\frac{L^2}{2a_1} (\sum_{j=1}^N ||\boldsymbol{w}_j^{t+1} - \boldsymbol{w}_j^t||^2 + ||\boldsymbol{z}^{t+1} - \boldsymbol{z}^t||^2 + ||h^{t+1} - h^t||^2)$$

$$+ (\frac{a_1}{2} - \frac{c_1^{t-1} - c_1^t}{2} + \frac{1}{2\rho_1})||\lambda_l^{t+1} - \lambda_l^t||^2 + \frac{c_1^{t-1}}{2}(||\lambda_l^{t+1}||^2 - ||\lambda_l^t||^2) + \frac{1}{2\rho_1}||\lambda_l^t - \lambda_l^{t-1}||^2)$$

$$= \frac{|\mathbf{A}^t| L^2}{2a_1} (\sum_{j=1}^N ||\boldsymbol{w}_j^{t+1} - \boldsymbol{w}_j^t||^2 + ||\boldsymbol{z}^{t+1} - \boldsymbol{z}^t||^2 + ||h^{t+1} - h^t||^2)$$

$$+ (\frac{a_1}{2} - \frac{c_1^{t-1} - c_1^t}{2} + \frac{1}{2\rho_1}) \sum_{l=1}^{|\mathbf{A}^t|} ||\lambda_l^{t+1} - \lambda_l^t||^2 + \frac{c_1^{t-1}}{2} \sum_{l=1}^{|\mathbf{A}^t|} (||\lambda_l^{t+1}||^2 - ||\lambda_l^t||^2) + \frac{1}{2\rho_1} \sum_{l=1}^{|\mathbf{A}^t|} ||\lambda_l^t - \lambda_l^{t-1}||^2.$$

$$(A.27)$$

According to Eq. (15), $\forall \phi \in \Phi$, it follows that,

$$\left\langle \boldsymbol{\phi}_j^{t+1} - \boldsymbol{\phi}_j^t - \rho_2 \nabla_{\phi_j} \widetilde{L}_p(\{\boldsymbol{w}_j^{t+1}\}, \boldsymbol{z}^{t+1}, h^{t+1}, \{\lambda_l^{t+1}\}, \{\boldsymbol{\phi}_j^t\}), \phi - \boldsymbol{\phi}_j^{t+1} \right\rangle \geq 0. \qquad (A.28)$$

Choosing $\phi = \boldsymbol{\phi}_j^t$, we can obtain,

$$\left\langle \nabla_{\phi_j} \widetilde{L}_p(\{\boldsymbol{w}_j^{t+1}\}, \boldsymbol{z}^{t+1}, h^{t+1}, \{\lambda_l^{t+1}\}, \{\boldsymbol{\phi}_j^t\}) - \frac{1}{\rho_2}(\boldsymbol{\phi}_j^{t+1} - \boldsymbol{\phi}_j^t), \boldsymbol{\phi}_j^t - \boldsymbol{\phi}_j^{t+1} \right\rangle \leq 0. \qquad (A.29)$$

Likewise, we have,

$$\left\langle \nabla_{\phi_j} \widetilde{L}_p(\{\boldsymbol{w}_j^t\}, \boldsymbol{z}^t, h^t, \{\lambda_l^t\}, \{\boldsymbol{\phi}_j^{t-1}\}) - \frac{1}{\rho_2}(\boldsymbol{\phi}_j^t - \boldsymbol{\phi}_j^{t-1}), \boldsymbol{\phi}_j^{t+1} - \boldsymbol{\phi}_j^t \right\rangle \leq 0. \qquad (A.30)$$

Since $\widetilde{L}_p(\{\boldsymbol{w}_j\}, \boldsymbol{z}, h, \{\lambda_l\}, \{\boldsymbol{\phi}_j\})$ is concave with respect to $\phi_j$ and follows from Eq. (A.30):

$$\widetilde{L}_p(\{\boldsymbol{w}_j^{t+1}\}, \boldsymbol{z}^{t+1}, h^{t+1}, \{\lambda_l^{t+1}\}, \{\boldsymbol{\phi}_j^{t+1}\}) - \widetilde{L}_p(\{\boldsymbol{w}_j^{t+1}\}, \boldsymbol{z}^{t+1}, h^{t+1}, \{\lambda_l^{t+1}\}, \{\boldsymbol{\phi}_j^t\})$$

$$\leq \sum_{j=1}^N \left\langle \nabla_{\phi_j} \widetilde{L}_p(\{\boldsymbol{w}_j^{t+1}\}, \boldsymbol{z}^{t+1}, h^{t+1}, \{\lambda_l^{t+1}\}, \{\boldsymbol{\phi}_j^t\}), \boldsymbol{\phi}_j^{t+1} - \boldsymbol{\phi}_j^t \right\rangle$$

$$\leq \sum_{j=1}^N (\langle \nabla_{\phi_j} \widetilde{L}_p(\{\boldsymbol{w}_j^{t+1}\}, \boldsymbol{z}^{t+1}, h^{t+1}, \{\lambda_l^{t+1}\}, \{\boldsymbol{\phi}_j^t\}) - \nabla_{\phi_j} \widetilde{L}_p(\{\boldsymbol{w}_j^t\}, \boldsymbol{z}^t, h^t, \{\lambda_l^t\}, \{\boldsymbol{\phi}_j^{t-1}\}), \boldsymbol{\phi}_j^{t+1} - \boldsymbol{\phi}_j^t \rangle$$

$$+ \frac{1}{\rho_2} \langle \boldsymbol{\phi}_j^t - \boldsymbol{\phi}_j^{t-1}, \boldsymbol{\phi}_j^{t+1} - \boldsymbol{\phi}_j^t \rangle).$$

$$(A.31)$$

Denoting $\boldsymbol{v}_{2,l}^{t+1} = \boldsymbol{\phi}_j^{t+1} - \boldsymbol{\phi}_j^t - (\boldsymbol{\phi}_j^t - \boldsymbol{\phi}_j^{t-1})$, we can write the first term in the last inequality of Eq. (A.31) as

$$\sum_{j=1}^N \left\langle \nabla_{\phi_j} \widetilde{L}_p(\{\boldsymbol{w}_j^{t+1}\}, \boldsymbol{z}^{t+1}, h^{t+1}, \{\lambda_l^{t+1}\}, \{\boldsymbol{\phi}_j^t\}) - \nabla_{\phi_j} \widetilde{L}_p(\{\boldsymbol{w}_j^t\}, \boldsymbol{z}^t, h^t, \{\lambda_l^t\}, \{\boldsymbol{\phi}_j^{t-1}\}), \boldsymbol{\phi}_j^{t+1} - \boldsymbol{\phi}_j^t \right\rangle$$

$$= \sum_{j=1}^N \left\langle \nabla_{\phi_j} \widetilde{L}_p(\{\boldsymbol{w}_j^{t+1}\}, \boldsymbol{z}^{t+1}, h^{t+1}, \{\lambda_l^{t+1}\}, \{\boldsymbol{\phi}_j^t\}) - \nabla_{\phi_j} \widetilde{L}_p(\{\boldsymbol{w}_j^t\}, \boldsymbol{z}^t, h^t, \{\lambda_l^t\}, \{\boldsymbol{\phi}_j^t\}), \boldsymbol{\phi}_j^{t+1} - \boldsymbol{\phi}_j^t \right\rangle (2a)$$

$$+ \sum_{j=1}^N \left\langle \nabla_{\phi_j} \widetilde{L}_p(\{\boldsymbol{w}_j^t\}, \boldsymbol{z}^t, h^t, \{\lambda_l^t\}, \{\boldsymbol{\phi}_j^t\}) - \nabla_{\phi_j} \widetilde{L}_p(\{\boldsymbol{w}_j^t\}, \boldsymbol{z}^t, h^t, \{\lambda_l^t\}, \{\boldsymbol{\phi}_j^{t-1}\}), \boldsymbol{v}_{2,l}^{t+1} \right\rangle (2b)$$

$$+ \sum_{j=1}^N \left\langle \nabla_{\phi_j} \widetilde{L}_p(\{\boldsymbol{w}_j^t\}, \boldsymbol{z}^t, h^t, \{\lambda_l^t\}, \{\boldsymbol{\phi}_j^t\}) - \nabla_{\phi_j} \widetilde{L}_p(\{\boldsymbol{w}_j^t\}, \boldsymbol{z}^t, h^t, \{\lambda_l^t\}, \{\boldsymbol{\phi}_j^{t-1}\}), \boldsymbol{\phi}_j^t - \boldsymbol{\phi}_j^{t-1} \right\rangle (2c).$$

$$(A.32)$$

We firstly focus on the $(2a)$ in Eq. (A.32), we can write the $(2a)$ as,

$$
\left\langle \nabla_{\boldsymbol{\phi}_j} \widetilde{L}_p(\{\boldsymbol{w}_j^{t+1}\}, \boldsymbol{z}^{t+1}, h^{t+1}, \{\lambda_l^{t+1}\}, \{\boldsymbol{\phi}_j^t\}) - \nabla_{\boldsymbol{\phi}_j} \widetilde{L}_p(\{\boldsymbol{w}_j^t\}, \boldsymbol{z}^t, h^t, \{\lambda_l^t\}, \{\boldsymbol{\phi}_j^t\}), \boldsymbol{\phi}_j^{t+1} - \boldsymbol{\phi}_j^t \right\rangle
$$
$$
= \left\langle \nabla_{\boldsymbol{\phi}_j} L_p(\{\boldsymbol{w}_j^{t+1}\}, \boldsymbol{z}^{t+1}, h^{t+1}, \{\lambda_l^{t+1}\}, \{\boldsymbol{\phi}_j^t\}) - \nabla_{\boldsymbol{\phi}_j} L_p(\{\boldsymbol{w}_j^t\}, \boldsymbol{z}^t, h^t, \{\lambda_l^t\}, \{\boldsymbol{\phi}_j^t\}), \boldsymbol{\phi}_j^{t+1} - \boldsymbol{\phi}_j^t \right\rangle
$$
$$
+ (c_2^{t-1} - c_2^t) \left\langle \boldsymbol{\phi}_j^t, \boldsymbol{\phi}_j^{t+1} - \boldsymbol{\phi}_j^t \right\rangle
$$
$$
= \left\langle \nabla_{\boldsymbol{\phi}_j} L_p(\{\boldsymbol{w}_j^{t+1}\}, \boldsymbol{z}^{t+1}, h^{t+1}, \{\lambda_l^{t+1}\}, \{\boldsymbol{\phi}_j^t\}) - \nabla_{\boldsymbol{\phi}_j} L_p(\{\boldsymbol{w}_j^t\}, \boldsymbol{z}^t, h^t, \{\lambda_l^t\}, \{\boldsymbol{\phi}_j^t\}), \boldsymbol{\phi}_j^{t+1} - \boldsymbol{\phi}_j^t \right\rangle
$$
$$
+ \frac{c_2^{t-1} - c_2^t}{2} (||\boldsymbol{\phi}_j^{t+1}||^2 - ||\boldsymbol{\phi}_j^t||^2) - \frac{c_2^{t-1} - c_2^t}{2} ||\boldsymbol{\phi}_j^{t+1} - \boldsymbol{\phi}_j^t||^2).
$$

$$(A.33)$$

And according to Cauchy-Schwarz inequality and Assumption 1, we can obtain,

$$
\left\langle \nabla_{\boldsymbol{\phi}_j} L_p(\{\boldsymbol{w}_j^{t+1}\}, \boldsymbol{z}^{t+1}, h^{t+1}, \{\lambda_l^{t+1}\}, \{\boldsymbol{\phi}_j^t\}) - \nabla_{\boldsymbol{\phi}_j} L_p(\{\boldsymbol{w}_j^t\}, \boldsymbol{z}^t, h^t, \{\lambda_l^t\}, \{\boldsymbol{\phi}_j^t\}), \boldsymbol{\phi}_j^{t+1} - \boldsymbol{\phi}_j^t \right\rangle
$$
$$
= \left\langle \nabla_{\boldsymbol{\phi}_j} L_p(\{\boldsymbol{w}_j^{t+1}\}, \boldsymbol{z}^{t+1}, h^{t+1}, \{\lambda_l^t\}, \{\boldsymbol{\phi}_j^t\}) - \nabla_{\boldsymbol{\phi}_j} L_p(\{\boldsymbol{w}_j^t\}, \boldsymbol{z}^t, h^t, \{\lambda_l^t\}, \{\boldsymbol{\phi}_j^t\}), \boldsymbol{\phi}_j^{t+1} - \boldsymbol{\phi}_j^t \right\rangle
$$
$$
\leq \frac{L^2}{2a_3} \left( \sum_{j=1}^N ||\boldsymbol{w}_j^{t+1} - \boldsymbol{w}_j^t||^2 + ||\boldsymbol{z}^{t+1} - \boldsymbol{z}^t||^2 + ||h^{t+1} - h^t||^2 \right) + \frac{a_3}{2} ||\boldsymbol{\phi}_j^{t+1} - \boldsymbol{\phi}_j^t||^2,
$$

$$(A.34)$$

where $a_3 > 0$ is a constant. Thus, we can obtain the upper bound of $(2a)$ by combining the above Eq. (A.33) and Eq. (A.34),

$$
\sum_{j=1}^N \left\langle \nabla_{\boldsymbol{\phi}_j} \widetilde{L}_p(\{\boldsymbol{w}_j^{t+1}\}, \boldsymbol{z}^{t+1}, h^{t+1}, \{\lambda_l^{t+1}\}, \{\boldsymbol{\phi}_j^t\}) - \nabla_{\boldsymbol{\phi}_j} \widetilde{L}_p(\{\boldsymbol{w}_j^t\}, \boldsymbol{z}^t, h^t, \{\lambda_l^t\}, \{\boldsymbol{\phi}_j^t\}), \boldsymbol{\phi}_j^{t+1} - \boldsymbol{\phi}_j^t \right\rangle
$$
$$
= \sum_{j \in \mathbf{Q}^{t+1}} \left\langle \nabla_{\boldsymbol{\phi}_j} \widetilde{L}_p(\{\boldsymbol{w}_j^{t+1}\}, \boldsymbol{z}^{t+1}, h^{t+1}, \{\lambda_l^{t+1}\}, \{\boldsymbol{\phi}_j^t\}) - \nabla_{\boldsymbol{\phi}_j} \widetilde{L}_p(\{\boldsymbol{w}_j^t\}, \boldsymbol{z}^t, h^t, \{\lambda_l^t\}, \{\boldsymbol{\phi}_j^t\}), \boldsymbol{\phi}_j^{t+1} - \boldsymbol{\phi}_j^t \right\rangle
$$
$$
\leq \sum_{j \in \mathbf{Q}^{t+1}} \left( \frac{L^2}{2a_3} \left( \sum_{j=1}^N ||\boldsymbol{w}_j^{t+1} - \boldsymbol{w}_j^t||^2 + ||\boldsymbol{z}^{t+1} - \boldsymbol{z}^t||^2 + ||h^{t+1} - h^t||^2 \right) + \frac{a_3}{2} ||\boldsymbol{\phi}_j^{t+1} - \boldsymbol{\phi}_j^t||^2 \right.
$$
$$
\left. + \frac{c_2^{t-1} - c_2^t}{2} (||\boldsymbol{\phi}_j^{t+1}||^2 - ||\boldsymbol{\phi}_j^t||^2) - \frac{c_2^{t-1} - c_2^t}{2} ||\boldsymbol{\phi}_j^{t+1} - \boldsymbol{\phi}_j^t||^2 \right).
$$

$$(A.35)$$

Next we focus on the $(2b)$ in Eq. (A.32). According to Cauchy-Schwarz inequality we can write the $(2b)$ as

$$
\sum_{j=1}^N \left\langle \nabla_{\boldsymbol{\phi}_j} \widetilde{L}_p(\{\boldsymbol{w}_j^t\}, \boldsymbol{z}^t, h^t, \{\lambda_l^t\}, \{\boldsymbol{\phi}_j^t\}) - \nabla_{\boldsymbol{\phi}_j} \widetilde{L}_p(\{\boldsymbol{w}_j^t\}, \boldsymbol{z}^t, h^t, \{\lambda_l^t\}, \{\boldsymbol{\phi}_j^{t-1}\}), \boldsymbol{v}_{2,l}^{t+1} \right\rangle
$$
$$
\leq \sum_{j=1}^N \left( \frac{a_4}{2} ||\nabla_{\boldsymbol{\phi}_j} \widetilde{L}_p(\{\boldsymbol{w}_j^t\}, \boldsymbol{z}^t, h^t, \{\lambda_l^t\}, \{\boldsymbol{\phi}_j^t\}) - \nabla_{\boldsymbol{\phi}_j} \widetilde{L}_p(\{\boldsymbol{w}_j^t\}, \boldsymbol{z}^t, h^t, \{\lambda_l^t\}, \{\boldsymbol{\phi}_j^{t-1}\})||^2 + \frac{1}{2a_4} ||\boldsymbol{v}_{2,l}^{t+1}||^2 \right),
$$

$$(A.36)$$

where $a_4 > 0$ is a constant. Then, we focus on the $(2c)$ in Eq. (A.32), we have,

$$
||\nabla_{\boldsymbol{\phi}_j} \widetilde{L}_p(\{\boldsymbol{w}_j^t\}, \boldsymbol{z}^t, h^t, \{\lambda_l^t\}, \{\boldsymbol{\phi}_j^t\}) - \nabla_{\boldsymbol{\phi}_j} \widetilde{L}_p(\{\boldsymbol{w}_j^t\}, \boldsymbol{z}^t, h^t, \{\lambda_l^t\}, \{\boldsymbol{\phi}_j^{t-1}\})||
$$
$$
\leq ||\nabla_{\boldsymbol{\phi}_j} L_p(\{\boldsymbol{w}_j^t\}, \boldsymbol{z}^t, h^t, \{\lambda_l^t\}, \{\boldsymbol{\phi}_j^t\}) - \nabla_{\boldsymbol{\phi}_j} L_p(\{\boldsymbol{w}_j^t\}, \boldsymbol{z}^t, h^t, \{\lambda_l^t\}, \{\boldsymbol{\phi}_j^{t-1}\})|| + c_2^{t-1} ||\boldsymbol{\phi}_j^t - \boldsymbol{\phi}_j^{t-1}||
$$
$$
\leq (L + c_2^{t-1}) ||\boldsymbol{\phi}_j^t - \boldsymbol{\phi}_j^{t-1}||,
$$

$$(A.37)$$

where the last inequality comes from Assumption 1 and the trigonometric inequality. Denoting $L_2' = L + c_2^0$, we can obtain,

$$
||\nabla_{\boldsymbol{\phi}_j} \widetilde{L}_p(\{\boldsymbol{w}_j^t\}, \boldsymbol{z}^t, h^t, \{\lambda_l^t\}, \{\boldsymbol{\phi}_j^t\}) - \nabla_{\boldsymbol{\phi}_j} \widetilde{L}_p(\{\boldsymbol{w}_j^t\}, \boldsymbol{z}^t, h^t, \{\lambda_l^t\}, \{\boldsymbol{\phi}_j^{t-1}\})|| \leq L_2' ||\boldsymbol{\phi}_j^t - \boldsymbol{\phi}_j^{t-1}||.
$$

$$(A.38)$$

Following Eq. (A.38) and the strong concavity of $\widetilde{L}_p(\{\boldsymbol{w}_j\}, \boldsymbol{z}, h, \{\lambda_l\}, \{\boldsymbol{\phi}_j\})$ w.r.t $\boldsymbol{\phi}_j$, we can obtain the upper bound of $(2c)$,

$$
\sum_{j=1}^{N} \left\langle \nabla_{\phi_j} \widetilde{L}_p(\{\boldsymbol{w}_j^t\}, \boldsymbol{z}^t, h^t, \{\lambda_l^t\}, \{\boldsymbol{\phi}_j^t\}) - \nabla_{\phi_j} \widetilde{L}_p(\{\boldsymbol{w}_j^t\}, \boldsymbol{z}^t, h^t, \{\lambda_l^t\}, \{\boldsymbol{\phi}_j^{t-1}\}), \boldsymbol{\phi}_j^t - \boldsymbol{\phi}_j^{t-1} \right\rangle
$$
$$
\leq \sum_{j=1}^{N} \left( -\frac{1}{L_2' + c_2^{t-1}} || \nabla_{\phi_j} \widetilde{L}_p(\{\boldsymbol{w}_j^t\}, \boldsymbol{z}^t, h^t, \{\lambda_l^t\}, \{\boldsymbol{\phi}_j^t\}) - \nabla_{\phi_j} \widetilde{L}_p(\{\boldsymbol{w}_j^t\}, \boldsymbol{z}^t, h^t, \{\lambda_l^t\}, \{\boldsymbol{\phi}_j^{t-1}\}) ||^2 \right.
$$
$$
\left. -\frac{c_2^{t-1} L_2'}{L_2' + c_2^{t-1}} || \boldsymbol{\phi}_j^t - \boldsymbol{\phi}_j^{t-1} ||^2 \right).
$$
(A.39)

In addition, the following inequality can also be obtained,

$$
\sum_{j=1}^{N} \frac{1}{\rho_2} \left\langle \boldsymbol{\phi}_j^t - \boldsymbol{\phi}_j^{t-1}, \boldsymbol{\phi}_j^{t+1} - \boldsymbol{\phi}_j^t \right\rangle \leq \sum_{j=1}^{N} \left( \frac{1}{2\rho_2} || \boldsymbol{\phi}_j^{t+1} - \boldsymbol{\phi}_j^t ||^2 - \frac{1}{2\rho_2} || \boldsymbol{v}_{2,l}^{t+1} ||^2 + \frac{1}{2\rho_2} || \boldsymbol{\phi}_j^t - \boldsymbol{\phi}_j^{t-1} ||^2 \right).
$$
(A.40)

Combining Eq. (A.31), (A.32), (A.35), (A.36), (A.39), (A.40), $\frac{\rho_2}{2} \leq \frac{1}{L_2' + c_2^0}$, and setting $a_4 = \rho_2$, we have,

$$
L_p(\{\boldsymbol{w}_j^{t+1}\}, \boldsymbol{z}^{t+1}, h^{t+1}, \{\lambda_l^{t+1}\}, \{\boldsymbol{\phi}_j^{t+1}\}) - L_p(\{\boldsymbol{w}_j^{t+1}\}, \boldsymbol{z}^{t+1}, h^{t+1}, \{\lambda_l^{t+1}\}, \{\boldsymbol{\phi}_j^t\})
$$
$$
\leq \sum_{j=1}^{N} \left( \left\langle \nabla_{\phi_j} \widetilde{L}_p(\{\boldsymbol{w}_j^{t+1}\}, \boldsymbol{z}^{t+1}, h^{t+1}, \{\lambda_l^{t+1}\}, \{\boldsymbol{\phi}_j^t\}) - \nabla_{\phi_j} \widetilde{L}_p(\{\boldsymbol{w}_j^t\}, \boldsymbol{z}^t, h^t, \{\lambda_l^t\}, \{\boldsymbol{\phi}_j^{t-1}\}), \boldsymbol{\phi}_j^{t+1} - \boldsymbol{\phi}_j^t \right\rangle \right.
$$
$$
\left. + \frac{1}{\rho_2} \left\langle \boldsymbol{\phi}_j^t - \boldsymbol{\phi}_j^{t-1}, \boldsymbol{\phi}_j^{t+1} - \boldsymbol{\phi}_j^t \right\rangle + \frac{c_2^t}{2} (|| \boldsymbol{\phi}_j^{t+1} ||^2 - || \boldsymbol{\phi}_j^t ||^2) \right)
$$
$$
\leq \frac{|\mathbf{Q}^{t+1}| L^2}{2a_3} \left( \sum_{j=1}^{N} || \boldsymbol{w}_j^{t+1} - \boldsymbol{w}_j^t ||^2 + || \boldsymbol{z}^{t+1} - \boldsymbol{z}^t ||^2 + || h^{t+1} - h^t ||^2 \right)
$$
$$
+ \left( \frac{a_3}{2} - \frac{c_2^{t-1} - c_2^t}{2} + \frac{1}{2\rho_2} \right) \sum_{j=1}^{N} || \boldsymbol{\phi}_j^{t+1} - \boldsymbol{\phi}_j^t ||^2 + \frac{c_2^{t-1}}{2} \sum_{j=1}^{N} (|| \boldsymbol{\phi}_j^{t+1} ||^2 - || \boldsymbol{\phi}_j^t ||^2) + \frac{1}{2\rho_2} \sum_{j=1}^{N} || \boldsymbol{\phi}_j^t - \boldsymbol{\phi}_j^{t-1} ||^2.
$$
(A.41)

By combining Lemma 1 with Eq. (A.27) and Eq. (A.41), we conclude the proof of Lemma 2.

**Lemma 3** *Firstly, we denote $S_1^{t+1}$, $S_2^{t+1}$ and $F^{t+1}$ as,*

$$
S_1^{t+1} = \frac{4}{\rho_1{}^2 c_1^{t+1}} \sum_{l=1}^{|\mathbf{A}^t|} || \lambda_l^{t+1} - \lambda_l^t ||^2 - \frac{4}{\rho_1} \left( \frac{c_1^{t-1}}{c_1^t} - 1 \right) \sum_{l=1}^{|\mathbf{A}^t|} || \lambda_l^{t+1} ||^2,
$$
(A.42)

$$
S_2^{t+1} = \frac{4}{\rho_2{}^2 c_2^{t+1}} \sum_{j=1}^{N} || \boldsymbol{\phi}_j^{t+1} - \boldsymbol{\phi}_j^t ||^2 - \frac{4}{\rho_2} \left( \frac{c_2^{t-1}}{c_2^t} - 1 \right) \sum_{j=1}^{N} || \boldsymbol{\phi}_j^{t+1} ||^2,
$$
(A.43)

$$
F^{t+1} = L_p(\{\boldsymbol{w}_j^{t+1}\}, \boldsymbol{z}^{t+1}, h^{t+1}, \{\lambda_l^{t+1}\}, \{\boldsymbol{\phi}_j^{t+1}\}) + S_1^{t+1} + S_2^{t+1}
$$
$$
- \frac{7}{2\rho_1} \sum_{l=1}^{|\mathbf{A}^t|} || \lambda_l^{t+1} - \lambda_l^t ||^2 - \frac{c_1^t}{2} \sum_{l=1}^{|\mathbf{A}^t|} || \lambda_l^{t+1} ||^2 - \frac{7}{2\rho_2} \sum_{j=1}^{N} || \boldsymbol{\phi}_j^{t+1} - \boldsymbol{\phi}_j^t ||^2 - \frac{c_2^t}{2} \sum_{j=1}^{N} || \boldsymbol{\phi}_j^{t+1} ||^2,
$$
(A.44)

*then $\forall t \geq T_1 + \tau$, we have,*

$$F^{t+1} - F^t \leq (\frac{L+1}{2} - \frac{1}{\eta_w^t} + \frac{\rho_1 |\mathbf{A}^t| L^2}{2} + \frac{\rho_2 |\mathbf{Q}^{t+1}| L^2}{2} + \frac{8|\mathbf{A}^t| L^2}{\rho_1 (c_1^t)^2} + \frac{8NL^2}{\rho_2 (c_2^t)^2}) \sum_{j=1}^{N} ||\boldsymbol{w}_j^{t+1} - \boldsymbol{w}_j^t||^2$$

$$+ (\frac{L+3\tau k_1 NL^2}{2} - \frac{1}{\eta_z^t} + \frac{\rho_1 |\mathbf{A}^t| L^2}{2} + \frac{\rho_2 |\mathbf{Q}^{t+1}| L^2}{2} + \frac{8|\mathbf{A}^t| L^2}{\rho_1 (c_1^t)^2} + \frac{8NL^2}{\rho_2 (c_2^t)^2}) ||\boldsymbol{z}^{t+1} - \boldsymbol{z}^t||^2$$

$$+ (\frac{L+3\tau k_1 NL^2}{2} - \frac{1}{\eta_h^t} + \frac{\rho_1 |\mathbf{A}^t| L^2}{2} + \frac{\rho_2 |\mathbf{Q}^{t+1}| L^2}{2} + \frac{8|\mathbf{A}^t| L^2}{\rho_1 (c_1^t)^2} + \frac{8NL^2}{\rho_2 (c_2^t)^2}) ||h^{t+1} - h^t||^2$$

$$- (\frac{1}{10\rho_1} - \frac{3\tau k_1 NL^2}{2}) \sum_{l=1}^{|\mathbf{A}^t|} ||\lambda_l^{t+1} - \lambda_l^t||^2 - \frac{1}{10\rho_2} \sum_{j=1}^{N} ||\phi_j^{t+1} - \phi_j^t||^2 + \frac{c_1^{t-1} - c_1^t}{2} \sum_{l=1}^{|\mathbf{A}^t|} ||\lambda_l^{t+1}||^2$$

$$+ \frac{c_2^{t-1} - c_2^t}{2} \sum_{j=1}^{N} ||\phi_j^{t+1}||^2 + \frac{4}{\rho_1} (\frac{c_1^{t-2}}{c_1^{t-1}} - \frac{c_1^{t-1}}{c_1^t}) \sum_{l=1}^{|\mathbf{A}^t|} ||\lambda_l^t||^2 + \frac{4}{\rho_2} (\frac{c_2^{t-2}}{c_2^{t-1}} - \frac{c_2^{t-1}}{c_2^t}) \sum_{j=1}^{N} ||\phi_j^t||^2.$$

(A.45)

***Proof of Lemma 3:***

Let $a_1 = \frac{1}{\rho_1}$, $a_3 = \frac{1}{\rho_2}$ and substitute them into Lemma 2, $\forall t \geq T_1 + \tau$, we have,

$$L_p(\{\boldsymbol{w}_j^{t+1}\}, \boldsymbol{z}^{t+1}, h^{t+1}, \{\lambda_l^{t+1}\}, \{\phi_j^{t+1}\}) - L_p(\{\boldsymbol{w}_j^t\}, \boldsymbol{z}^t, h^t, \{\lambda_l^t\}, \{\phi_j^t\})$$

$$\leq (\frac{L+1}{2} - \frac{1}{\eta_w^t} + \frac{\rho_1 |\mathbf{A}^t| L^2}{2} + \frac{\rho_2 |\mathbf{Q}^{t+1}| L^2}{2}) \sum_{j=1}^{N} ||\boldsymbol{w}_j^{t+1} - \boldsymbol{w}_j^t||^2$$

$$+ (\frac{L+3\tau k_1 NL^2}{2} - \frac{1}{\eta_z^t} + \frac{\rho_1 |\mathbf{A}^t| L^2}{2} + \frac{\rho_2 |\mathbf{Q}^{t+1}| L^2}{2}) ||\boldsymbol{z}^{t+1} - \boldsymbol{z}^t||^2$$

$$+ (\frac{L+3\tau k_1 NL^2}{2} - \frac{1}{\eta_h^t} + \frac{\rho_1 |\mathbf{A}^t| L^2}{2} + \frac{\rho_2 |\mathbf{Q}^{t+1}| L^2}{2}) ||h^{t+1} - h^t||^2$$

$$+ (\frac{3\tau k_1 NL^2}{2} + \frac{1}{\rho_1} - \frac{c_1^{t-1} - c_1^t}{2}) \sum_{l=1}^{|\mathbf{A}^t|} ||\lambda_l^{t+1} - \lambda_l^t||^2$$

$$+ \frac{c_1^{t-1}}{2} \sum_{l=1}^{|\mathbf{A}^t|} (||\lambda_l^{t+1}||^2 - ||\lambda_l^t||^2) + \frac{1}{2\rho_1} \sum_{l=1}^{|\mathbf{A}^t|} ||\lambda_l^t - \lambda_l^{t-1}||^2 + (\frac{1}{\rho_2} - \frac{c_2^{t-1} - c_2^t}{2}) \sum_{j=1}^{N} ||\phi_j^{t+1} - \phi_j^t||^2$$

$$+ \frac{c_2^{t-1}}{2} \sum_{j=1}^{N} (||\phi_j^{t+1}||^2 - ||\phi_j^t||^2) + \frac{1}{2\rho_2} \sum_{j=1}^{N} ||\phi_j^t - \phi_j^{t-1}||^2.$$

(A.46)

According to Eq. (14), in $(t+1)^{\text{th}}$ iteration, it follows that:

$$\left\langle \lambda_l^{t+1} - \lambda_l^t - \rho_1 \nabla_{\lambda_l} \widetilde{L}_p(\{\boldsymbol{w}_j^{t+1}\}, \boldsymbol{z}^{t+1}, h^{t+1}, \{\lambda_l^t\}, \{\phi_j^t\}), \lambda_l^t - \lambda_l^{t+1} \right\rangle \geq 0. \qquad \text{(A.47)}$$

Similar to Eq. (A.47), in $t^{\text{th}}$ iteration, we have,

$$\left\langle \lambda_l^t - \lambda_l^{t-1} - \rho_1 \nabla_{\lambda_l} \widetilde{L}_p(\{\boldsymbol{w}_j^t\}, \boldsymbol{z}^t, h^t, \{\lambda_l^{t-1}\}, \{\phi_j^{t-1}\}), \lambda_l^{t+1} - \lambda_l^t \right\rangle \geq 0. \qquad \text{(A.48)}$$

$\forall t \geq T_1$, we can obtain the following inequality,

$$\sum_{l=1}^{|\mathbf{A}^t|} \frac{1}{\rho_1} \left\langle \boldsymbol{v}_{1,l}^{t+1}, \lambda_l^{t+1} - \lambda_l^t \right\rangle$$

$$\leq \sum_{l=1}^{|\mathbf{A}^t|} (\left\langle \nabla_{\lambda_l} \widetilde{L}_p(\{\boldsymbol{w}_j^{t+1}\}, \boldsymbol{z}^{t+1}, h^{t+1}, \{\lambda_l^t\}, \{\phi_j^t\}) - \nabla_{\lambda_l} \widetilde{L}_p(\{\boldsymbol{w}_j^t\}, \boldsymbol{z}^t, h^t, \{\lambda_l^t\}, \{\phi_j^t\}), \lambda_l^{t+1} - \lambda_l^t \right\rangle$$

$$+ \left\langle \nabla_{\lambda_l} \widetilde{L}_p(\{\boldsymbol{w}_j^t\}, \boldsymbol{z}^t, h^t, \{\lambda_l^t\}, \{\phi_j^t\}) - \nabla_{\lambda_l} \widetilde{L}_p(\{\boldsymbol{w}_j^t\}, \boldsymbol{z}^t, h^t, \{\lambda_l^{t-1}\}, \{\phi_j^{t-1}\}), \boldsymbol{v}_{1,l}^{t+1} \right\rangle$$

$$+ \left\langle \nabla_{\lambda_l} \widetilde{L}_p(\{\boldsymbol{w}_j^t\}, \boldsymbol{z}^t, h^t, \{\lambda_l^t\}, \{\phi_j^t\}) - \nabla_{\lambda_l} \widetilde{L}_p(\{\boldsymbol{w}_j^t\}, \boldsymbol{z}^t, h^t, \{\lambda_l^{t-1}\}, \{\phi_j^{t-1}\}), \lambda_l^t - \lambda_l^{t-1} \right\rangle).$$

(A.49)

Since we have the following equality,

$$\frac{1}{\rho_1} \left\langle \boldsymbol{v}_{1,l}^{t+1}, \lambda_l^{t+1} - \lambda_l^t \right\rangle = \frac{1}{2\rho_1} ||\lambda_l^{t+1} - \lambda_l^t||^2 + \frac{1}{2\rho_1} ||\boldsymbol{v}_{1,l}^{t+1}||^2 - \frac{1}{2\rho_1} ||\lambda_l^t - \lambda_l^{t-1}||^2, \qquad \text{(A.50)}$$

it follows that,

$$\sum_{l=1}^{|\mathbf{A}^t|} \left( \frac{1}{2\rho_1} ||\lambda_l^{t+1} - \lambda_l^t||^2 + \frac{1}{2\rho_1} ||\boldsymbol{v}_{1,l}^{t+1}||^2 - \frac{1}{2\rho_1} ||\lambda_l^t - \lambda_l^{t-1}||^2 \right)$$

$$\leq \sum_{l=1}^{|\mathbf{A}^t|} \left( \frac{L^2}{2b_1^t} \left( \sum_{j=1}^{N} ||\boldsymbol{w}_j^{t+1} - \boldsymbol{w}_j^t||^2 + ||\boldsymbol{z}^{t+1} - \boldsymbol{z}^t||^2 + ||h^{t+1} - h^t||^2 \right) + \frac{b_1^t}{2} ||\lambda_l^{t+1} - \lambda_l^t||^2 \right.$$

$$+ \frac{c_1^{t-1} - c_1^t}{2} \left( ||\lambda_l^{t+1}||^2 - ||\lambda_l^t||^2 \right) - \frac{c_1^{t-1} - c_1^t}{2} ||\lambda_l^{t+1} - \lambda_l^t||^2$$

$$+ \frac{\rho_1}{2} ||\nabla_{\lambda_l} \widetilde{L}_p(\{\boldsymbol{w}_j^t\}, \boldsymbol{z}^t, h^t, \{\lambda_l^t\}, \{\boldsymbol{\phi}_j^t\}) - \nabla_{\lambda_l} \widetilde{L}_p(\{\boldsymbol{w}_j^t\}, \boldsymbol{z}^t, h^t, \{\lambda_l^{t-1}\}, \{\boldsymbol{\phi}_j^{t-1}\})||^2 + \frac{1}{2\rho_1} ||\boldsymbol{v}_{1,l}^{t+1}||^2$$

$$- \frac{1}{L_1' + c_1^{t-1}} ||\nabla_{\lambda_l} \widetilde{L}_p(\{\boldsymbol{w}_j^t\}, \boldsymbol{z}^t, h^t, \{\lambda_l^t\}, \{\boldsymbol{\phi}_j^t\}) - \nabla_{\lambda_l} \widetilde{L}_p(\{\boldsymbol{w}_j^t\}, \boldsymbol{z}^t, h^t, \{\lambda_l^{t-1}\}, \{\boldsymbol{\phi}_j^{t-1}\})||^2$$

$$\left. - \frac{c_1^{t-1} L_1'}{L_1' + c_1^{t-1}} ||\lambda_l^t - \lambda_l^{t-1}||^2 \right), \tag{A.51}$$

where $b_1^t > 0$. According to the setting that $c_1^0 \leq L_1'$, we have $-\frac{c_1^{t-1} L_1'}{L_1' + c_1^{t-1}} \leq -\frac{c_1^{t-1} L_1'}{2L_1'} = -\frac{c_1^{t-1}}{2} \leq -\frac{c_1^t}{2}$. Multiplying both sides of the inequality Eq. (A.51) by $\frac{8}{\rho_1 c_1^t}$, we have,

$$\sum_{l=1}^{|\mathbf{A}^t|} \left( \frac{4}{\rho_1^2 c_1^t} ||\lambda_l^{t+1} - \lambda_l^t||^2 - \frac{4}{\rho_1} \left( \frac{c_1^{t-1} - c_1^t}{c_1^t} \right) ||\lambda_l^{t+1}||^2 \right)$$

$$\leq \sum_{l=1}^{|\mathbf{A}^t|} \left( \frac{4}{\rho_1^2 c_1^t} ||\lambda_l^t - \lambda_l^{t-1}||^2 - \frac{4}{\rho_1} \left( \frac{c_1^{t-1} - c_1^t}{c_1^t} \right) ||\lambda_l^t||^2 + \frac{4 b_1^t}{\rho_1 c_1^t} ||\lambda_l^{t+1} - \lambda_l^t||^2 - \frac{4}{\rho_1} ||\lambda_l^t - \lambda_l^{t-1}||^2 \right.$$

$$\left. + \frac{4 L^2}{\rho_1 c_1^t b_1^t} \left( \sum_{j=1}^{N} ||\boldsymbol{w}_j^{t+1} - \boldsymbol{w}_j^t||^2 + ||\boldsymbol{z}^{t+1} - \boldsymbol{z}^t||^2 + ||h^{t+1} - h^t||^2 \right) \right). \tag{A.52}$$

Setting $b_1^t = \frac{c_1^t}{2}$ in Eq. (A.52) and using the definition of $S_1^t$, we have,

$$S_1^{t+1} - S_1^t$$

$$\leq \sum_{l=1}^{|\mathbf{A}^t|} \left( \frac{4}{\rho_1} \left( \frac{c_1^{t-2}}{c_1^{t-1}} - \frac{c_1^{t-1}}{c_1^t} \right) ||\lambda_l^t||^2 + \frac{8 L^2}{\rho_1 (c_1^t)^2} \left( \sum_{j=1}^{N} ||\boldsymbol{w}_j^{t+1} - \boldsymbol{w}_j^t||^2 + ||\boldsymbol{z}^{t+1} - \boldsymbol{z}^t||^2 + ||h^{t+1} - h^t||^2 \right) \right.$$

$$\left. + \left( \frac{2}{\rho_1} + \frac{4}{\rho_1^2} \left( \frac{1}{c_1^{t+1}} - \frac{1}{c_1^t} \right) \right) ||\lambda_l^{t+1} - \lambda_l^t||^2 - \frac{4}{\rho_1} ||\lambda_l^t - \lambda_l^{t-1}||^2 \right)$$

$$= \sum_{l=1}^{|\mathbf{A}^t|} \frac{4}{\rho_1} \left( \frac{c_1^{t-2}}{c_1^{t-1}} - \frac{c_1^{t-1}}{c_1^t} \right) ||\lambda_l^t||^2 + \sum_{l=1}^{|\mathbf{A}^t|} \left( \frac{2}{\rho_1} + \frac{4}{\rho_1^2} \left( \frac{1}{c_1^{t+1}} - \frac{1}{c_1^t} \right) \right) ||\lambda_l^{t+1} - \lambda_l^t||^2$$

$$- \sum_{l=1}^{|\mathbf{A}^t|} \frac{4}{\rho_1} ||\lambda_l^t - \lambda_l^{t-1}||^2 + \frac{8 |\mathbf{A}^t| L^2}{\rho_1 (c_1^t)^2} \left( \sum_{j=1}^{N} ||\boldsymbol{w}_j^{t+1} - \boldsymbol{w}_j^t||^2 + ||\boldsymbol{z}^{t+1} - \boldsymbol{z}^t||^2 + ||h^{t+1} - h^t||^2 \right). \tag{A.53}$$

Likewise, according to Eq. (15), we have that,

$$\frac{1}{\rho_2} \left\langle \boldsymbol{v}_{2,l}^{t+1}, \boldsymbol{\phi}_j^{t+1} - \boldsymbol{\phi}_j^t \right\rangle$$

$$\leq \left\langle \nabla_{\boldsymbol{\phi}_j} \widetilde{L}_p(\{\boldsymbol{w}_j^{t+1}\}, \boldsymbol{z}^{t+1}, h^{t+1}, \{\lambda_l^{t+1}\}, \{\boldsymbol{\phi}_j^t\}) - \nabla_{\boldsymbol{\phi}_j} \widetilde{L}_p(\{\boldsymbol{w}_j^t\}, \boldsymbol{z}^t, h^t, \{\lambda_l^t\}, \{\boldsymbol{\phi}_j^{t-1}\}), \boldsymbol{\phi}_j^{t+1} - \boldsymbol{\phi}_j^t \right\rangle$$

$$= \left\langle \nabla_{\boldsymbol{\phi}_j} \widetilde{L}_p(\{\boldsymbol{w}_j^{t+1}\}, \boldsymbol{z}^{t+1}, h^{t+1}, \{\lambda_l^{t+1}\}, \{\boldsymbol{\phi}_j^t\}) - \nabla_{\boldsymbol{\phi}_j} \widetilde{L}_p(\{\boldsymbol{w}_j^t\}, \boldsymbol{z}^t, h^t, \{\lambda_l^t\}, \{\boldsymbol{\phi}_j^t\}), \boldsymbol{\phi}_j^{t+1} - \boldsymbol{\phi}_j^t \right\rangle$$

$$+ \left\langle \nabla_{\boldsymbol{\phi}_j} \widetilde{L}_p(\{\boldsymbol{w}_j^t\}, \boldsymbol{z}^t, h^t, \{\lambda_l^t\}, \{\boldsymbol{\phi}_j^t\}) - \nabla_{\boldsymbol{\phi}_j} \widetilde{L}_p(\{\boldsymbol{w}_j^t\}, \boldsymbol{z}^t, h^t, \{\lambda_l^t\}, \{\boldsymbol{\phi}_j^{t-1}\}), \boldsymbol{v}_{2,l}^{t+1} \right\rangle$$

$$+ \left\langle \nabla_{\boldsymbol{\phi}_j} \widetilde{L}_p(\{\boldsymbol{w}_j^t\}, \boldsymbol{z}^t, h^t, \{\lambda_l^t\}, \{\boldsymbol{\phi}_j^t\}) - \nabla_{\boldsymbol{\phi}_j} \widetilde{L}_p(\{\boldsymbol{w}_j^t\}, \boldsymbol{z}^t, h^t, \{\lambda_l^t\}, \{\boldsymbol{\phi}_j^{t-1}\}), \boldsymbol{\phi}_j^t - \boldsymbol{\phi}_j^{t-1} \right\rangle. \tag{A.54}$$

In addition, since

$$\frac{1}{\rho_2} \left\langle \boldsymbol{v}_{2,l}^{t+1}, \boldsymbol{\phi}_j^{t+1} - \boldsymbol{\phi}_j^t \right\rangle = \frac{1}{2\rho_2} ||\boldsymbol{\phi}_j^{t+1} - \boldsymbol{\phi}_j^t||^2 + \frac{1}{2\rho_2} ||\boldsymbol{v}_{2,l}^{t+1}||^2 - \frac{1}{2\rho_2} ||\boldsymbol{\phi}_j^t - \boldsymbol{\phi}_j^{t-1}||^2, \tag{A.55}$$

it follows that,

$$
\frac{1}{2\rho_2}||\boldsymbol{\phi}_j^{t+1}-\boldsymbol{\phi}_j^t||^2 + \frac{1}{2\rho_2}||\boldsymbol{v}_{2,l}^{t+1}||^2 - \frac{1}{2\rho_2}||\boldsymbol{\phi}_j^t-\boldsymbol{\phi}_j^{t-1}||^2
$$

$$
\leq \frac{L^2}{2b_2^t}\Big(\sum_{j=1}^N ||\boldsymbol{w}_j^{t+1}-\boldsymbol{w}_j^t||^2 + ||\boldsymbol{z}^{t+1}-\boldsymbol{z}^t||^2 + ||h^{t+1}-h^t||^2\Big) + \frac{b_2^t}{2}||\boldsymbol{\phi}_j^{t+1}-\boldsymbol{\phi}_j^t||^2
$$

$$
+\frac{c_2^{t-1}-c_2^t}{2}(||\boldsymbol{\phi}_j^{t+1}||^2-||\boldsymbol{\phi}_j^t||^2) - \frac{c_2^{t-1}-c_2^t}{2}||\boldsymbol{\phi}_j^{t+1}-\boldsymbol{\phi}_j^t||^2 - \frac{c_2^t L_2'}{L_2'+c_2^{t-1}}||\boldsymbol{\phi}_j^t-\boldsymbol{\phi}_j^{t-1}||^2
$$

$$
+\frac{\rho_2}{2}||\nabla_{\boldsymbol{\phi}_j}\widetilde{L}_p(\{\boldsymbol{w}_j^t\},\boldsymbol{z}^t,h^t,\{\lambda_l^t\},\{\boldsymbol{\phi}_j^t\}) - \nabla_{\boldsymbol{\phi}_j}\widetilde{L}_p(\{\boldsymbol{w}_j^t\},\boldsymbol{z}^t,h^t,\{\lambda_l^t\},\{\boldsymbol{\phi}_j^{t-1}\})||^2 + \frac{1}{2\rho_2}||\boldsymbol{v}_{2,l}^{t+1}||^2
$$

$$
-\frac{1}{L_2'+c_2^{t-1}}||\nabla_{\boldsymbol{\phi}_j}\widetilde{L}_p(\{\boldsymbol{w}_j^t\},\boldsymbol{z}^t,h^t,\{\lambda_l^t\},\{\boldsymbol{\phi}_j^t\}) - \nabla_{\boldsymbol{\phi}_j}\widetilde{L}_p(\{\boldsymbol{w}_j^t\},\boldsymbol{z}^t,h^t,\{\lambda_l^t\},\{\boldsymbol{\phi}_j^{t-1}\})||^2.
$$

(A.56)

According to the setting $c_2^0 \leq L_2'$, we have $-\frac{c_2^{t-1}L_2'}{L_2'+c_2^{t-1}} \leq -\frac{c_2^{t-1}L_2'}{2L_2'} = -\frac{c_2^{t-1}}{2} \leq -\frac{c_2^t}{2}$. Multiplying both sides of the inequality Eq. (A.56) by $\frac{8}{\rho_2 c_2^t}$, we have,

$$
\frac{4}{\rho_2{}^2 c_2^t}||\boldsymbol{\phi}_j^{t+1}-\boldsymbol{\phi}_j^t||^2 - \frac{4}{\rho_2}\Big(\frac{c_2^{t-1}-c_2^t}{c_2^t}\Big)||\boldsymbol{\phi}_j^{t+1}||^2
$$

$$
\leq \frac{4}{\rho_2{}^2 c_2^t}||\boldsymbol{\phi}_j^t-\boldsymbol{\phi}_j^{t-1}||^2 - \frac{4}{\rho_2}\Big(\frac{c_2^{t-1}-c_2^t}{c_2^t}\Big)||\boldsymbol{\phi}_j^t||^2 + \frac{4b_2^t}{\rho_2 c_2^t}||\boldsymbol{\phi}_j^{t+1}-\boldsymbol{\phi}_j^t||^2 - \frac{4}{\rho_2}||\boldsymbol{\phi}_j^t-\boldsymbol{\phi}_j^{t-1}||^2
$$

(A.57)

$$
+\frac{4L^2}{\rho_2 c_2^t b_2^t}\Big(\sum_{j=1}^N ||\boldsymbol{w}_j^{t+1}-\boldsymbol{w}_j^t||^2 + ||\boldsymbol{z}^{t+1}-\boldsymbol{z}^t||^2 + ||h^{t+1}-h^t||^2\Big).
$$

Setting $b_2^t = \frac{c_2^t}{2}$ in Eq. (A.57) and using the definition of $S_2^t$, we can obtain,

$$
S_2^{t+1}-S_2^t
$$

$$
\leq \sum_{j=1}^N \Big(\frac{4}{\rho_2}\Big(\frac{c_2^{t-2}}{c_2^{t-1}}-\frac{c_2^{t-1}}{c_2^t}\Big)||\boldsymbol{\phi}_j^t||^2 + \frac{8L^2}{\rho_2(c_2^t)^2}\Big(\sum_{j=1}^N ||\boldsymbol{w}_j^{t+1}-\boldsymbol{w}_j^t||^2 + ||\boldsymbol{z}^{t+1}-\boldsymbol{z}^t||^2 + \sum_{l=1}^{|\mathbf{A}^t|}||\lambda_l^{t+1}-\lambda_l^t||^2\Big)
$$

$$
+\Big(\frac{2}{\rho_2}+\frac{2}{\rho_2{}^2}\Big(\frac{1}{c_2^{t+1}}-\frac{1}{c_2^t}\Big)\Big)||\boldsymbol{\phi}_j^{t+1}-\boldsymbol{\phi}_j^t||^2 - \frac{4}{\rho_2}||\boldsymbol{\phi}_j^t-\boldsymbol{\phi}_j^{t-1}||^2\Big)
$$

$$
=\sum_{j=1}^N \frac{4}{\rho_2}\Big(\frac{c_2^{t-2}}{c_2^{t-1}}-\frac{c_2^{t-1}}{c_2^t}\Big)||\boldsymbol{\phi}_j^t||^2 + \sum_{j=1}^N\Big(\frac{2}{\rho_2}+\frac{4}{\rho_2{}^2}\Big(\frac{1}{c_2^{t+1}}-\frac{1}{c_2^t}\Big)\Big)||\boldsymbol{\phi}_j^{t+1}-\boldsymbol{\phi}_j^t||^2
$$

$$
-\sum_{j=1}^N \frac{4}{\rho_2}||\boldsymbol{\phi}_j^t-\boldsymbol{\phi}_j^{t-1}||^2 + \frac{8NL^2}{\rho_2(c_2^t)^2}\Big(\sum_{j=1}^N ||\boldsymbol{w}_j^{t+1}-\boldsymbol{w}_j^t||^2 + ||\boldsymbol{z}^{t+1}-\boldsymbol{z}^t||^2 + ||h^{t+1}-h^t||^2\Big).
$$

(A.58)

According to the setting about $c_1^t$ and $c_2^t$, we have $\frac{\rho_1}{10} \geq \frac{1}{c_1^{t+1}} - \frac{1}{c_1^t}, \frac{\rho_2}{10} \geq \frac{1}{c_2^{t+1}} - \frac{1}{c_2^t}, \forall t \geq T_1$. Using the definition of $F^{t+1}$ and combining it with Eq. (A.53) and Eq. (A.58), $\forall t \geq T_1 + \tau$, we have,

$$
F^{t+1}-F^t
$$

$$
\leq \Big(\frac{L+1}{2}-\frac{1}{\eta_{\boldsymbol{w}}^t}+\frac{\rho_1|\mathbf{A}^t|L^2}{2}+\frac{\rho_2|\mathbf{Q}^{t+1}|L^2}{2}+\frac{8|\mathbf{A}^t|L^2}{\rho_1(c_1^t)^2}+\frac{8NL^2}{\rho_2(c_2^t)^2}\Big)\sum_{j=1}^N ||\boldsymbol{w}_j^{t+1}-\boldsymbol{w}_j^t||^2
$$

$$
+\Big(\frac{L+3\tau k_1 NL^2}{2}-\frac{1}{\eta_{\boldsymbol{z}}^t}+\frac{\rho_1|\mathbf{A}^t|L^2}{2}+\frac{\rho_2|\mathbf{Q}^{t+1}|L^2}{2}+\frac{8|\mathbf{A}^t|L^2}{\rho_1(c_1^t)^2}+\frac{8NL^2}{\rho_2(c_2^t)^2}\Big)||\boldsymbol{z}^{t+1}-\boldsymbol{z}^t||^2
$$

$$
+\Big(\frac{L+3\tau k_1 NL^2}{2}-\frac{1}{\eta_h^t}+\frac{\rho_1|\mathbf{A}^t|L^2}{2}+\frac{\rho_2|\mathbf{Q}^{t+1}|L^2}{2}+\frac{8|\mathbf{A}^t|L^2}{\rho_1(c_1^t)^2}+\frac{8NL^2}{\rho_2(c_2^t)^2}\Big)||h^{t+1}-h^t||^2
$$

(A.59)

$$
-\Big(\frac{1}{10\rho_1}-\frac{3\tau k_1 NL^2}{2}\Big)\sum_{l=1}^{|\mathbf{A}^t|}||\lambda_l^{t+1}-\lambda_l^t||^2 - \frac{1}{10\rho_2}\sum_{j=1}^N ||\boldsymbol{\phi}_j^{t+1}-\boldsymbol{\phi}_j^t||^2 + \frac{c_1^{t-1}-c_1^t}{2}\sum_{l=1}^{|\mathbf{A}^t|}||\lambda_l^{t+1}||^2
$$

$$
+\frac{c_2^{t-1}-c_2^t}{2}\sum_{j=1}^N ||\boldsymbol{\phi}_j^{t+1}||^2 + \frac{4}{\rho_1}\Big(\frac{c_1^{t-2}}{c_1^{t-1}}-\frac{c_1^{t-1}}{c_1^t}\Big)\sum_{l=1}^{|\mathbf{A}^t|}||\lambda_l^t||^2 + \frac{4}{\rho_2}\Big(\frac{c_2^{t-2}}{c_2^{t-1}}-\frac{c_2^{t-1}}{c_2^t}\Big)\sum_{j=1}^N ||\boldsymbol{\phi}_j^t||^2.
$$

Next, we will combine Lemma 1, Lemma 2 with Lemma 3 to derive Theorem 1. Firstly, we make some definitions about our problem.

**Definition A.3** *The stationarity gap at $t^{th}$ iteration is defined as:*

$$\nabla G^t = \begin{bmatrix} \{\frac{1}{\alpha_w^t}(\boldsymbol{w}_j^t - \mathcal{P}_{\mathcal{W}}(\boldsymbol{w}_j^t - \alpha_w^t \nabla_{\boldsymbol{w}_j} L_p(\{\boldsymbol{w}_j^t\}, \boldsymbol{z}^t, h^t, \{\lambda_l^t\}, \{\boldsymbol{\phi}_j^t\})))\} \\ \frac{1}{\eta_z^t}(\boldsymbol{z}^t - \mathcal{P}_{\mathcal{Z}}(\boldsymbol{z}^t - \eta_z^t \nabla_{\boldsymbol{z}} L_p(\{\boldsymbol{w}_j^t\}, \boldsymbol{z}^t, h^t, \{\lambda_l^t\}, \{\boldsymbol{\phi}_j^t\}))) \\ \frac{1}{\eta_h^t}(h^t - \mathcal{P}_{\mathcal{H}}(h^t - \eta_h^t \nabla_h L_p(\{\boldsymbol{w}_j^t\}, \boldsymbol{z}^t, h^t, \{\lambda_l^t\}, \{\boldsymbol{\phi}_j^t\}))) \\ \{\frac{1}{\rho_1}(\lambda_l^t - \mathcal{P}_{\boldsymbol{\Lambda}}(\lambda_l^t + \rho_1 \nabla_{\lambda_l} L_p(\{\boldsymbol{w}_j^t\}, \boldsymbol{z}^t, h^t, \{\lambda_l^t\}, \{\boldsymbol{\phi}_j^t\})))\} \\ \{\frac{1}{\rho_2}(\boldsymbol{\phi}_j^t - \mathcal{P}_{\boldsymbol{\Phi}}(\boldsymbol{\phi}_j^t + \rho_2 \nabla_{\boldsymbol{\phi}_j} L_p(\{\boldsymbol{w}_j^t\}, \boldsymbol{z}^t, h^t, \{\lambda_l^t\}, \{\boldsymbol{\phi}_j^t\})))\} \end{bmatrix}. \tag{A.60}$$

*And we also define:*

$$(\nabla G^t)_{\boldsymbol{w}_j} = \frac{1}{\alpha_w^t}(\boldsymbol{w}_j^t - \mathcal{P}_{\mathcal{W}}(\boldsymbol{w}_j^t - \alpha_w^t \nabla_{\boldsymbol{w}_j} L_p(\{\boldsymbol{w}_j^t\}, \boldsymbol{z}^t, h^t, \{\lambda_l^t\}, \{\boldsymbol{\phi}_j^t\}))),$$

$$(\nabla G^t)_{\boldsymbol{z}} = \frac{1}{\eta_z^t}(\boldsymbol{z}^t - \mathcal{P}_{\mathcal{Z}}(\boldsymbol{z}^t - \eta_z^t \nabla_{\boldsymbol{z}} L_p(\{\boldsymbol{w}_j^t\}, \boldsymbol{z}^t, h^t, \{\lambda_l^t\}, \{\boldsymbol{\phi}_j^t\}))),$$

$$(\nabla G^t)_h = \frac{1}{\eta_h^t}(h^t - \mathcal{P}_{\mathcal{H}}(h^t - \eta_h^t \nabla_h L_p(\{\boldsymbol{w}_j^t\}, \boldsymbol{z}^t, h^t, \{\lambda_l^t\}, \{\boldsymbol{\phi}_j^t\}))), \tag{A.61}$$

$$(\nabla G^t)_{\lambda_l} = \frac{1}{\rho_1}(\lambda_l^t - \mathcal{P}_{\boldsymbol{\Lambda}}(\lambda_l^t + \rho_1 \nabla_{\lambda_l} L_p(\{\boldsymbol{w}_j^t\}, \boldsymbol{z}^t, h^t, \{\lambda_l^t\}, \{\boldsymbol{\phi}_j^t\}))),$$

$$(\nabla G^t)_{\boldsymbol{\phi}_j} = \frac{1}{\rho_2}(\boldsymbol{\phi}_j^t - \mathcal{P}_{\boldsymbol{\Phi}}(\boldsymbol{\phi}_j^t + \rho_2 \nabla_{\boldsymbol{\phi}_j} L_p(\{\boldsymbol{w}_j^t\}, \boldsymbol{z}^t, h^t, \{\lambda_l^t\}, \{\boldsymbol{\phi}_j^t\}))).$$

*It follows that,*

$$||\nabla G^t||^2 = \sum_{j=1}^N ||(\nabla G^t)_{\boldsymbol{w}_j}||^2 + ||(\nabla G^t)_{\boldsymbol{z}}||^2 + ||(\nabla G^t)_h||^2 + \sum_{l=1}^{|\mathbf{A}^t|} ||(\nabla G^t)_{\lambda_l}||^2 + \sum_{j=1}^N ||(\nabla G^t)_{\boldsymbol{\phi}_j}||^2. \tag{A.62}$$

**Definition A.4** *At $t^{th}$ iteration, the stationarity gap w.r.t $\widetilde{L}_p(\{\boldsymbol{w}_j\}, \boldsymbol{z}, h, \{\lambda_l\}, \{\boldsymbol{\phi}_j\})$ is defined as:*

$$\nabla \widetilde{G}^t = \begin{bmatrix} \{\frac{1}{\alpha_w^t}(\boldsymbol{w}_j^t - \mathcal{P}_{\mathcal{W}}(\boldsymbol{w}_j^t - \alpha_w^t \nabla_{\boldsymbol{w}_j} \widetilde{L}_p(\{\boldsymbol{w}_j^t\}, \boldsymbol{z}^t, h^t, \{\lambda_l^t\}, \{\boldsymbol{\phi}_j^t\})))\} \\ \frac{1}{\eta_z^t}(\boldsymbol{z}^t - \mathcal{P}_{\mathcal{Z}}(\boldsymbol{z}^t - \eta_z^t \nabla_{\boldsymbol{z}} \widetilde{L}_p(\{\boldsymbol{w}_j^t\}, \boldsymbol{z}^t, h^t, \{\lambda_l^t\}, \{\boldsymbol{\phi}_j^t\}))) \\ \frac{1}{\eta_h^t}(h^t - \mathcal{P}_{\mathcal{H}}(h^t - \eta_h^t \nabla_h \widetilde{L}_p(\{\boldsymbol{w}_j^t\}, \boldsymbol{z}^t, h^t, \{\lambda_l^t\}, \{\boldsymbol{\phi}_j^t\}))) \\ \{\frac{1}{\rho_1}(\lambda_l^t - \mathcal{P}_{\boldsymbol{\Lambda}}(\lambda_l^t + \rho_1 \nabla_{\lambda_l} \widetilde{L}_p(\{\boldsymbol{w}_j^t\}, \boldsymbol{z}^t, h^t, \{\lambda_l^t\}, \{\boldsymbol{\phi}_j^t\})))\} \\ \{\frac{1}{\rho_2}(\boldsymbol{\phi}_j^t - \mathcal{P}_{\boldsymbol{\Phi}}(\boldsymbol{\phi}_j^t + \rho_2 \nabla_{\boldsymbol{\phi}_j} \widetilde{L}_p(\{\boldsymbol{w}_j^t\}, \boldsymbol{z}^t, h^t, \{\lambda_l^t\}, \{\boldsymbol{\phi}_j^t\})))\} \end{bmatrix}. \tag{A.63}$$

*We further define:*

$$(\nabla \widetilde{G}^t)_{\boldsymbol{w}_j} = \frac{1}{\alpha_w^t}(\boldsymbol{w}_j^t - \mathcal{P}_{\mathcal{W}}(\boldsymbol{w}_j^t - \alpha_w^t \nabla_{\boldsymbol{w}_j} \widetilde{L}_p(\{\boldsymbol{w}_j^t\}, \boldsymbol{z}^t, h^t, \{\lambda_l^t\}, \{\boldsymbol{\phi}_j^t\}))),$$

$$(\nabla \widetilde{G}^t)_{\boldsymbol{z}} = \frac{1}{\eta_z^t}(\boldsymbol{z}^t - \mathcal{P}_{\mathcal{Z}}(\boldsymbol{z}^t - \eta_z^t \nabla_{\boldsymbol{z}} \widetilde{L}_p(\{\boldsymbol{w}_j^t\}, \boldsymbol{z}^t, h^t, \{\lambda_l^t\}, \{\boldsymbol{\phi}_j^t\}))),$$

$$(\nabla \widetilde{G}^t)_h = \frac{1}{\eta_h^t}(h^t - \mathcal{P}_{\mathcal{H}}(h^t - \eta_h^t \nabla_h \widetilde{L}_p(\{\boldsymbol{w}_j^t\}, \boldsymbol{z}^t, h^t, \{\lambda_l^t\}, \{\boldsymbol{\phi}_j^t\}))), \tag{A.64}$$

$$(\nabla \widetilde{G}^t)_{\lambda_l} = \frac{1}{\rho_1}(\lambda_l^t - \mathcal{P}_{\boldsymbol{\Lambda}}(\lambda_l^t + \rho_1 \nabla_{\lambda_l} \widetilde{L}_p(\{\boldsymbol{w}_j^t\}, \boldsymbol{z}^t, h^t, \{\lambda_l^t\}, \{\boldsymbol{\phi}_j^t\}))),$$

$$(\nabla \widetilde{G}^t)_{\boldsymbol{\phi}_j} = \frac{1}{\rho_2}(\boldsymbol{\phi}_j^t - \mathcal{P}_{\boldsymbol{\Phi}}(\boldsymbol{\phi}_j^t + \rho_2 \nabla_{\boldsymbol{\phi}_j} \widetilde{L}_p(\{\boldsymbol{w}_j^t\}, \boldsymbol{z}^t, h^t, \{\lambda_l^t\}, \{\boldsymbol{\phi}_j^t\}))).$$

*It follows that,*

$$||\nabla \widetilde{G}^t||^2 = \sum_{j=1}^N ||(\nabla \widetilde{G}^t)_{\boldsymbol{w}_j}||^2 + ||(\nabla \widetilde{G}^t)_{\boldsymbol{z}}||^2 + ||(\nabla \widetilde{G}^t)_h||^2 + \sum_{l=1}^{|\mathbf{A}^t|} ||(\nabla \widetilde{G}^t)_{\lambda_l}||^2 + \sum_{j=1}^N ||(\nabla \widetilde{G}^t)_{\boldsymbol{\phi}_j}||^2. \tag{A.65}$$

**Definition A.5** *In our asynchronous algorithm, for the worker $j$ in $t^{th}$ iteration, we define the last iteration where worker $j$ was active as $\widetilde{t}_j$. And we define the next iteration that worker $j$ will be active as $\overline{t}_j$. For the iteration index set that worker $j$ is active from $T_1^{th}$ to $(T_1 + T + \tau)^{th}$ iteration, we define it as $\mathcal{V}_j(T)$. And the $i^{th}$ element in $\mathcal{V}_j(T)$ is defined as $\hat{v}_j(i)$.*

*Proof of Theorem 1:*

Firstly, setting:

$$a_5^t = \frac{4|\mathbf{A}^t|(\gamma-2)L^2}{\rho_1(c_1^t)^2} + \frac{4N(\gamma-2)L^2}{\rho_2(c_2^t)^2} + \frac{\rho_2(N-|\mathbf{Q}^{t+1}|)L^2}{2} - \frac{1}{2}, \tag{A.66}$$

$$a_6^t = \frac{4|\mathbf{A}^t|(\gamma-2)L^2}{\rho_1(c_1^t)^2} + \frac{4N(\gamma-2)L^2}{\rho_2(c_2^t)^2} + \frac{\rho_2(N-|\mathbf{Q}^{t+1}|)L^2}{2} - \frac{3\tau k_1 N L^2}{2}, \tag{A.67}$$

where $\gamma$ is a constant which satisfies $\gamma > 2$ and $\frac{4(\gamma-2)L^2}{\rho_1(c_1^0)^2} + \frac{4N(\gamma-2)L^2}{\rho_2(c_2^0)^2} + \frac{\rho_2(N-S)L^2}{2} \geq \max\{\frac{1}{2}, \frac{3\tau k_1 N L^2}{2}\}$. It is seen that the $a_5^t, a_6^t$ are nonnegative sequences. Since $\forall t \geq 0$, $|\mathbf{A}^0| \leq |\mathbf{A}^t|$, $(c_1^0)^2 \geq (c_1^t)^2$, $(c_2^0)^2 \geq (c_2^t)^2$, and we assume that $|\mathbf{Q}^{t+1}| = S, \forall t$, thus we have $a_5^0 \leq a_5^t, a_6^0 \leq a_6^t, \forall t$. According to the setting of $\eta_{\boldsymbol{w}}^t, \eta_{\boldsymbol{z}}^t, \eta_h^t$ and $c_1^t, c_2^t$, we have,

$$\frac{L+1}{2} - \frac{1}{\eta_{\boldsymbol{w}}^t} + \frac{\rho_1|\mathbf{A}^t|L^2}{2} + \frac{\rho_2|\mathbf{Q}^{t+1}|L^2}{2} + \frac{8|\mathbf{A}^t|L^2}{\rho_1(c_1^t)^2} + \frac{8NL^2}{\rho_2(c_2^t)^2} = -a_5^t, \tag{A.68}$$

$$\frac{L+3\tau k_1 N L^2}{2} - \frac{1}{\eta_{\boldsymbol{z}}^t} + \frac{\rho_1|\mathbf{A}^t|L^2}{2} + \frac{\rho_2|\mathbf{Q}^{t+1}|L^2}{2} + \frac{8|\mathbf{A}^t|L^2}{\rho_1(c_1^t)^2} + \frac{8NL^2}{\rho_2(c_2^t)^2} = -a_6^t, \tag{A.69}$$

$$\frac{L+3\tau k_1 N L^2}{2} - \frac{1}{\eta_h^t} + \frac{\rho_1|\mathbf{A}^t|L^2}{2} + \frac{\rho_2|\mathbf{Q}^{t+1}|L^2}{2} + \frac{8|\mathbf{A}^t|L^2}{\rho_1(c_1^t)^2} + \frac{8NL^2}{\rho_2(c_2^t)^2} = -a_6^t. \tag{A.70}$$

Combining Eq. (A.68), (A.69), (A.70) with Lemma 3, $\forall t \geq T_1 + \tau$, it follows that,

$$a_5^t \sum_{j=1}^N ||\boldsymbol{w}_j^{t+1} - \boldsymbol{w}_j^t||^2 + a_6^t||\boldsymbol{z}^{t+1} - \boldsymbol{z}^t||^2 + a_6^t||h^{t+1} - h^t||^2$$
$$+ (\frac{1}{10\rho_1} - \frac{3\tau k_1 N L^2}{2})\sum_{l=1}^{|\mathbf{A}^t|} ||\lambda_l^{t+1} - \lambda_l^t||^2 + \frac{1}{10\rho_2}\sum_{j=1}^N ||\boldsymbol{\phi}_j^{t+1} - \boldsymbol{\phi}_j^t||^2$$
$$\leq F^t - F^{t+1} + \frac{c_1^{t-1}-c_1^t}{2}\sum_{l=1}^{|\mathbf{A}^t|} ||\lambda_l^{t+1}||^2 + \frac{c_2^{t-1}-c_2^t}{2}\sum_{j=1}^N ||\boldsymbol{\phi}_j^{t+1}||^2$$
$$+ \frac{4}{\rho_1}(\frac{c_1^{t-2}}{c_1^{t-1}} - \frac{c_1^{t-1}}{c_1^t})\sum_{l=1}^{|\mathbf{A}^t|} ||\lambda_l^t||^2 + \frac{4}{\rho_2}(\frac{c_2^{t-2}}{c_2^{t-1}} - \frac{c_2^{t-1}}{c_2^t})\sum_{j=1}^N ||\boldsymbol{\phi}_j^t||^2. \tag{A.71}$$

Combining the definition of $(\nabla\widetilde{G}^t)_{\boldsymbol{w}_j}$ with trigonometric inequality, Cauchy-Schwarz inequality and Assumption 1 and 2, $\forall t \geq T_1 + \tau$, we have,

$$||(\nabla\widetilde{G}^t)_{\boldsymbol{w}_j}||^2 \leq \frac{2}{\eta_{\boldsymbol{w}}^{\overline{t}_j2}}||\boldsymbol{w}_j^{\overline{t}_j} - \boldsymbol{w}_j^t||^2 + 6\tau k_1 L^2(||\boldsymbol{z}^{t+1} - \boldsymbol{z}^t||^2 + ||h^{t+1} - h^t||^2 + \sum_{l=1}^{|\mathbf{A}^t|} ||\lambda_l^{t+1} - \lambda_l^t||^2). \tag{A.72}$$

Combining the definition of $(\nabla\widetilde{G}^t)_{\boldsymbol{z}}$ with trigonometric inequality and Cauchy-Schwarz inequality, we can obtain the following inequality,

$$||(\nabla\widetilde{G}^t)_{\boldsymbol{z}}||^2 \leq 2L^2 \sum_{j=1}^N ||\boldsymbol{w}_j^{t+1} - \boldsymbol{w}_j^t||^2 + \frac{2}{(\eta_{\boldsymbol{z}}^t)^2}||\boldsymbol{z}^{t+1} - \boldsymbol{z}^t||^2. \tag{A.73}$$

Likewise, combining the definition of $(\nabla\widetilde{G}^t)_h$ with trigonometric inequality and Cauchy-Schwarz inequality, we have that,

$$||(\nabla\widetilde{G}^t)_h||^2 \leq 2L^2(\sum_{j=1}^N ||\boldsymbol{w}_j^{t+1} - \boldsymbol{w}_j^t||^2 + ||\boldsymbol{z}^{t+1} - \boldsymbol{z}^t||^2) + \frac{2}{(\eta_h^t)^2}||h^{t+1} - h^t||^2. \tag{A.74}$$

Combining the definition of $(\nabla \widetilde{G}^t)_{\lambda_l}$ with trigonometric inequality and Cauchy-Schwarz inequality,

$$||(\nabla \widetilde{G}^t)_{\lambda_l}||^2$$
$$\leq \tfrac{3}{\rho_1{}^2}||\lambda_l^{t+1}-\lambda_l^t||^2+3L^2(\sum_{j=1}^{N}||\boldsymbol{w}_j^{t+1}-\boldsymbol{w}_j^t||^2+||\boldsymbol{z}^{t+1}-\boldsymbol{z}^t||^2+||h^{t+1}-h^t||^2)+3(c_1^{t-1}-c_1^t)^2||\lambda_l^t||^2$$
$$\leq \tfrac{3}{\rho_1{}^2}||\lambda_l^{t+1}-\lambda_l^t||^2+3L^2(\sum_{j=1}^{N}||\boldsymbol{w}_j^{t+1}-\boldsymbol{w}_j^t||^2+||\boldsymbol{z}^{t+1}-\boldsymbol{z}^t||^2+||h^{t+1}-h^t||^2)+3((c_1^{t-1})^2-(c_1^t)^2)||\lambda_l^t||^2.$$

(A.75)

Combining the definition of $(\nabla \widetilde{G}^t)_{\boldsymbol{\phi}_j}$ with Cauchy-Schwarz inequality and Assumption 2, we have,

$$||(\nabla \widetilde{G}^t)_{\boldsymbol{\phi}_j}||^2$$
$$\leq \tfrac{3}{\rho_2{}^2}||\boldsymbol{\phi}_j^{\overline{t_j}}-\boldsymbol{\phi}_j^t||^2+3L^2(\sum_{j=1}^{N}||\boldsymbol{w}_j^{\overline{t_j}}-\boldsymbol{w}_j^t||^2+||\boldsymbol{z}^{\overline{t_j}}-\boldsymbol{z}^t||^2)+3(c_2^{\widetilde{t_j}-1}-c_2^{\overline{t_j}-1})^2||\boldsymbol{\phi}_j^t||^2$$
$$\leq \tfrac{3}{\rho_2{}^2}||\boldsymbol{\phi}_j^{\overline{t_j}}-\boldsymbol{\phi}_j^t||^2+3L^2(\sum_{j=1}^{N}||\boldsymbol{w}_j^{\overline{t_j}}-\boldsymbol{w}_j^t||^2+\tau k_1(||\boldsymbol{z}^{t+1}-\boldsymbol{z}^t||^2+||h^{t+1}-h^t||^2+\sum_{l=1}^{|\mathbf{A}^t|}||\lambda_l^{t+1}-\lambda_l^t||^2))$$
$$+3((c_2^{\widetilde{t_j}-1})^2-(c_2^{\overline{t_j}-1})^2)||\boldsymbol{\phi}_j^t||^2.$$

(A.76)

According to the Definition A.4 as well as Eq. (A.72), (A.73), (A.74), (A.75) and Eq. (A.76), $\forall t \geq T_1 + \tau$, we have that,

$$||\nabla \widetilde{G}^t||^2 = \sum_{j=1}^{N}||(\nabla \widetilde{G}^t)_{\boldsymbol{w}_j}||^2+||(\nabla \widetilde{G}^t)_{\boldsymbol{z}}||^2+||(\nabla \widetilde{G}^t)_h||^2+\sum_{l=1}^{|\mathbf{A}^t|}||(\nabla \widetilde{G}^t)_{\lambda_l}||^2+\sum_{j=1}^{N}||(\nabla \widetilde{G}^t)_{\boldsymbol{\phi}_j}||^2$$
$$\leq (\tfrac{2}{\eta_{\boldsymbol{w}}{}^2}+3NL^2)\sum_{j=1}^{N}||\boldsymbol{w}_j^{\overline{t_j}}-\boldsymbol{w}_j^t||^2+(4+3|\mathbf{A}^t|)L^2\sum_{j=1}^{N}||\boldsymbol{w}_j^{t+1}-\boldsymbol{w}_j^t||^2$$
$$+(\tfrac{2}{(\eta_{\boldsymbol{z}}^t)^2}+(2+9\tau k_1 N+3|\mathbf{A}^t|)L^2)||\boldsymbol{z}^{t+1}-\boldsymbol{z}^t||^2+(\tfrac{2}{(\eta_h^t)^2}+(9\tau k_1 N+3|\mathbf{A}^t|)L^2)||h^{t+1}-h^t||^2$$
$$+\sum_{l=1}^{|\mathbf{A}^t|}(\tfrac{3}{\rho_1{}^2}+9\tau k_1 NL^2)||\lambda_l^{t+1}-\lambda_l^t||^2+\sum_{l=1}^{|\mathbf{A}^t|}3((c_1^{t-1})^2-(c_1^t)^2)||\lambda_l^t||^2$$
$$+\sum_{j=1}^{N}\tfrac{3}{\rho_2{}^2}||\boldsymbol{\phi}_j^{\overline{t_j}}-\boldsymbol{\phi}_j^t||^2+\sum_{j=1}^{N}3((c_2^{\widetilde{t_j}-1})^2-(c_2^{\overline{t_j}-1})^2)||\boldsymbol{\phi}_j^t||^2.$$

(A.77)

We set constants $d_1$, $d_2$, $d_3$ as,

$$d_1 = \frac{2k_\tau \tau+(4+3M+3k_\tau \tau N)L^2\underline{\eta_{\boldsymbol{w}}}^2}{\underline{\eta_{\boldsymbol{w}}}^2(a_5^0)^2} \geq \frac{2k_\tau \tau+(4+3|\mathbf{A}^t|+3k_\tau \tau N)L^2\underline{\eta_{\boldsymbol{w}}}^2}{\underline{\eta_{\boldsymbol{w}}}^2(a_5^t)^2},$$ (A.78)

$$d_2 = \frac{2+(2+9\tau k_1 N+3M)L^2\underline{\eta_{\boldsymbol{z}}}^2}{\underline{\eta_{\boldsymbol{z}}}^2(a_6^0)^2} \geq \frac{2+(2+9\tau k_1 N+3|\mathbf{A}^t|)L^2(\eta_{\boldsymbol{z}}^t)^2}{(\eta_{\boldsymbol{z}}^t)^2(a_6^t)^2},$$ (A.79)

$$d_3 = \frac{2+(9\tau k_1 N+3M)L^2\underline{\eta_h}^2}{\underline{\eta_h}^2(a_6^0)^2} \geq \frac{2+(9\tau k_1 N+3|\mathbf{A}^t|)L^2(\eta_h^t)^2}{(\eta_h^t)^2(a_6^t)^2},$$ (A.80)

where $k_\tau$, $\underline{\eta_{\boldsymbol{z}}}$ and $\underline{\eta_h}$ are positive constants. $\underline{\eta_{\boldsymbol{z}}} = \frac{2}{L+\rho_1 ML^2+\rho_2 NL^2+8(\frac{M\gamma L^2}{\rho_1 \varepsilon_1^2}+\frac{N\gamma L^2}{\rho_2 \varepsilon_2^2})} \leq \eta_{\boldsymbol{z}}^t$ and $\underline{\eta_h} = \frac{2}{L+\rho_1 ML^2+\rho_2 NL^2+8(\frac{M\gamma L^2}{\rho_1 \varepsilon_1^2}+\frac{N\gamma L^2}{\rho_2 \varepsilon_2^2})} \leq \eta_h^t, \forall t$. Thus, combining Eq. (A.77) with Eq. (A.78), (A.79), (A.80), $\forall t \geq T_1 + \tau$, we have,

$$||\nabla \widetilde{G}^t||^2 \leq \sum_{j=1}^{N}d_1(a_5^t)^2||\boldsymbol{w}_j^{t+1}-\boldsymbol{w}_j^t||^2+d_2(a_6^t)^2||\boldsymbol{z}^{t+1}-\boldsymbol{z}^t||^2+d_3(a_6^t)^2||h^{t+1}-h^t||^2$$
$$+\sum_{l=1}^{|\mathbf{A}^t|}(\tfrac{3}{\rho_1{}^2}+9\tau k_1 NL^2)||\lambda_l^{t+1}-\lambda_l^t||^2+\sum_{l=1}^{|\mathbf{A}^t|}3((c_1^{t-1})^2-(c_1^t)^2)||\lambda_l^t||^2+\sum_{j=1}^{N}\tfrac{3}{\rho_2{}^2}||\boldsymbol{\phi}_j^{\overline{t_j}}-\boldsymbol{\phi}_j^t||^2$$
$$+\sum_{j=1}^{N}3((c_2^{\widetilde{t_j}-1})^2-(c_2^{\overline{t_j}-1})^2)||\boldsymbol{\phi}_j^t||^2+(\tfrac{2}{\eta_{\boldsymbol{w}}{}^2}+3NL^2)\sum_{j=1}^{N}||\boldsymbol{w}_j^{\overline{t_j}}-\boldsymbol{w}_j^t||^2$$
$$-(\tfrac{2k_\tau \tau}{\eta_{\boldsymbol{w}}{}^2}+3k_\tau \tau NL^2)\sum_{j=1}^{N}||\boldsymbol{w}_j^{t+1}-\boldsymbol{w}_j^t||^2.$$

(A.81)

Let $d_4^t$ denote a nonnegative sequence:

$$d_4^t = \frac{1}{\max\{d_1 a_5^t, d_2 a_6^t, d_3 a_6^t, \frac{\frac{30}{\rho_1} + 90\rho_1 \tau k_1 NL^2}{1 - 15\rho_1 \tau k_1 NL^2}, \frac{30\tau}{\rho_2}\}}. \tag{A.82}$$

It is seen that $d_4^0 \geq d_4^t, \forall t \geq 0$. And we denote the lower bound of $d_4^t$ as $\underline{d_4}$, it appears that $d_4^t \geq \underline{d_4} \geq 0, \forall t \geq 0$. And we set the constant $k_\tau$ satisfies $k_\tau \geq \frac{d_4^0(\frac{2}{\overline{\eta_w}^2} + 3NL^2)}{\underline{d_4}(\frac{2}{\overline{\eta_w}^2} + 3NL^2)}$, where $\overline{\eta_w}$ is the step-size in terms of $w_j$ in the first iteration (it is seen that $\overline{\eta_w} \geq \eta_w^t, \forall t$). Then, $\forall t \geq T_1 + \tau$, we can obtain the following inequality from Eq. (A.81) and Eq. (A.82):

$$d_4^t \|\nabla \widetilde{G}^t\|^2 \leq a_5^t \sum_{j=1}^{N} \|w_j^{t+1} - w_j^t\|^2 + a_6^t \|z^{t+1} - z^t\|^2 + a_6^t \|h^{t+1} - h^t\|^2$$

$$+ (\frac{1}{10\rho_1} - \frac{3\tau k_1 NL^2}{2}) \sum_{l=1}^{|\mathbf{A}^t|} \|\lambda_l^{t+1} - \lambda_l^t\|^2 + \frac{1}{10\tau\rho_2} \sum_{j=1}^{N} \|\phi_j^{\overline{t_j}} - \phi_j^t\|^2$$

$$+ 3d_4^t((c_1^{t-1})^2 - (c_1^t)^2) \sum_{l=1}^{|\mathbf{A}^t|} \|\lambda_l^t\|^2 + 3d_4^t \sum_{j=1}^{N} ((c_2^{\widetilde{t_j}-1})^2 - (c_2^{\overline{t_j}-1})^2) \|\phi_j^t\|^2$$

$$+ d_4^t(\frac{2}{\underline{\eta_w}^2} + 3NL^2) \sum_{j=1}^{N} \|w_j^{\overline{t_j}} - w_j^t\|^2 - d_4^t(\frac{2k_\tau \tau}{\underline{\eta_w}^2} + 3k_\tau \tau NL^2) \sum_{j=1}^{N} \|w_j^{t+1} - w_j^t\|^2. \tag{A.83}$$

Combining Eq. (A.83) with Eq. (A.71) and according to the setting $\|\lambda_l^t\|^2 \leq \sigma_1^2$, $\|\phi_j^t\|^2 \leq \sigma_2^2$ (where $\sigma_1^2 = \alpha_3^2$, $\sigma_2^2 = p\alpha_4^2$) and $d_4^0 \geq d_4^t \geq \underline{d_4}$, thus, $\forall t \geq T_1 + \tau$, we have,

$$d_4^t \|\nabla \widetilde{G}^t\|^2$$
$$\leq F^t - F^{t+1} + \frac{c_1^{t-1} - c_1^t}{2} M\sigma_1^2 + \frac{c_2^{t-1} - c_2^t}{2} N\sigma_2^2 + \frac{4}{\rho_1}(\frac{c_1^{t-2}}{c_1^{t-1}} - \frac{c_1^{t-1}}{c_1^t}) M\sigma_1^2$$

$$+ \frac{4}{\rho_2}(\frac{c_2^{t-2}}{c_2^{t-1}} - \frac{c_2^{t-1}}{c_2^t}) N\sigma_2^2 + 3d_4^0((c_1^{t-1})^2 - (c_1^t)^2) M\sigma_1^2 + 3d_4^0 \sum_{j=1}^{N} ((c_2^{\widetilde{t_j}-1})^2 - (c_2^{\overline{t_j}-1})^2)\sigma_2^2$$

$$+ \frac{1}{10\tau\rho_2} \sum_{j=1}^{N} \|\phi_j^{\overline{t_j}} - \phi_j^t\|^2 - \frac{1}{10\rho_2} \sum_{j=1}^{N} \|\phi_j^{t+1} - \phi_j^t\|^2$$

$$+ d_4^0(\frac{2}{\underline{\eta_w}^2} + 3NL^2) \sum_{j=1}^{N} \|w_j^{\overline{t_j}} - w_j^t\|^2 - \underline{d_4}(\frac{2k_\tau \tau}{\underline{\eta_w}^2} + 3k_\tau \tau NL^2) \sum_{j=1}^{N} \|w_j^{t+1} - w_j^t\|^2. \tag{A.84}$$

Denoting $\widetilde{T}(\varepsilon)$ as $\widetilde{T}(\varepsilon) = \min\{t \mid \|\nabla \widetilde{G}^{T_1+t}\| \leq \frac{\varepsilon}{2}, t \geq \tau\}$. Summing up Eq. (A.84) from $t = T_1 + \tau$ to $t = T_1 + \widetilde{T}(\varepsilon)$, we have,

$$\sum_{t=T_1+\tau}^{T_1+\widetilde{T}(\varepsilon)} d_4^t \|\nabla \widetilde{G}^t\|^2$$

$$\leq F^{T_1+\tau} - \underline{L} + \frac{4}{\rho_1}(\frac{c_1^{T_1+\tau-2}}{c_1^{T_1+\tau-1}} + \frac{c_1^{T_1+\tau-1}}{c_1^{T_1+\tau}}) M\sigma_1^2 + \frac{c_1^{T_1+\tau-1}}{2} M\sigma_1^2 + \frac{7}{2\rho_1} M\sigma_3^2 + 3d_4^0(c_1^0)^2 M\sigma_1^2$$

$$+ \frac{4}{\rho_2}(\frac{c_1^{T_1+\tau-2}}{c_1^{T_1+\tau-1}} + \frac{c_1^{T_1+\tau-1}}{c_1^{T_1+\tau}}) N\sigma_2^2 + \frac{c_2^{T_1+\tau-1}}{2} N\sigma_2^2 + \frac{7}{2\rho_2} N\sigma_4^2 + \sum_{j=1}^{N} \sum_{t=T_1+\tau}^{T_1+\widetilde{T}(\varepsilon)} 3d_4^0((c_2^{\widetilde{t_j}-1})^2 - (c_2^{\overline{t_j}-1})^2)\sigma_2^2$$

$$+ \frac{c_1^{T_1+\tau}}{2} M\sigma_1^2 + \frac{c_2^{T_1+\tau}}{2} N\sigma_2^2 + \frac{1}{10\tau\rho_2} \sum_{j=1}^{N} \sum_{t=T_1+\tau}^{T_1+\widetilde{T}(\varepsilon)} \|\phi_j^{\overline{t_j}} - \phi_j^t\|^2 - \frac{1}{10\rho_2} \sum_{j=1}^{N} \sum_{t=T_1+\tau}^{T_1+\widetilde{T}(\varepsilon)} \|\phi_j^{t+1} - \phi_j^t\|^2$$

$$+ d_4^0(\frac{2}{\underline{\eta_w}^2} + 3NL^2) \sum_{j=1}^{N} \sum_{t=T_1+\tau}^{T_1+\widetilde{T}(\varepsilon)} \|w_j^{\overline{t_j}} - w_j^t\|^2 - \underline{d_4}(\frac{2k_\tau \tau}{\underline{\eta_w}^2} + 3k_\tau \tau NL^2) \sum_{j=1}^{N} \sum_{t=T_1+\tau}^{T_1+\widetilde{T}(\varepsilon)} \|w_j^{t+1} - w_j^t\|^2, \tag{A.85}$$

where $\sigma_3 = \max\{||\lambda_1 - \lambda_2||\,|\lambda_1, \lambda_2 \in \boldsymbol{\Lambda}\}$, $\sigma_4 = \max\{||\boldsymbol{\phi}_1 - \boldsymbol{\phi}_2||\,|\boldsymbol{\phi}_1, \boldsymbol{\phi}_2 \in \boldsymbol{\Phi}\}$ and $\underline{L} = \min\limits_{\{\boldsymbol{w}_j \in \mathcal{W}\},\boldsymbol{z} \in \mathcal{Z},h \in \mathcal{H},\{\lambda_l \in \boldsymbol{\Lambda}\},\{\boldsymbol{\phi}_j \in \boldsymbol{\Phi}\}} L_p(\{\boldsymbol{w}_j\}, \boldsymbol{z}, h, \{\lambda_l\}, \{\boldsymbol{\phi}_j\})$, which satisfy that, $\forall t \geq T_1 + \tau$,

$$F^{t+1} \geq \underline{L} - \frac{4}{\rho_1}\frac{c_1^{T_1+\tau-1}}{c_1^{T_1+\tau}}M\sigma_1{}^2 - \frac{4}{\rho_2}\frac{c_2^{T_1+\tau-1}}{c_2^{T_1+\tau}}N\sigma_2{}^2 - \frac{7}{2\rho_1}M\sigma_3{}^2 - \frac{7}{2\rho_2}N\sigma_4{}^2 - \frac{c_1^{T_1+\tau}}{2}M\sigma_1{}^2 - \frac{c_2^{T_1+\tau}}{2}N\sigma_2{}^2.$$

$$(A.86)$$

For each worker $j$, the iterations between the last iteration and the next iteration where it is active is no more than $\tau$, $i.e.$, $\overline{t}_j - \widetilde{t}_j \leq \tau$, we have,

$$\sum_{t=T_1+\tau}^{T_1+\widetilde{T}(\varepsilon)} 3d_4^0((c_2^{\widetilde{t}_j-1})^2 - (c_2^{\overline{t}_j-1})^2)\sigma_2{}^2$$

$$\leq \tau \sum_{\substack{\hat{v}_j(i) \in \mathcal{V}_j(\widetilde{T}(\varepsilon)), \\ T_1+\tau \leq \hat{v}_j(i) \leq T_1+\widetilde{T}(\varepsilon)}} 3d_4^0((c_2^{\hat{v}_j(i)-1})^2 - (c_2^{\hat{v}_j(i+1)-1})^2)\sigma_2{}^2 \qquad (A.87)$$

$$\leq 3\tau d_4^0(c_2^0)^2\sigma_2{}^2.$$

Since the idle workers do not update their variables in each iteration, for any $t$ that satisfies $\hat{v}_j(i-1) \leq t < \hat{v}_j(i)$, we have $\boldsymbol{\phi}_j^t = \boldsymbol{\phi}_j^{\hat{v}_j(i)-1}$. And for $t \notin \mathcal{V}_j(T)$, we have $||\boldsymbol{\phi}_j^t - \boldsymbol{\phi}_j^{t-1}||^2 = 0$. Combing with $\hat{v}_j(i) - \hat{v}_j(i-1) \leq \tau$, we can obtain that,

$$\sum_{j=1}^{N}\sum_{t=T_1+\tau}^{T_1+\widetilde{T}(\varepsilon)}||\boldsymbol{\phi}_j^{\overline{t}_j} - \boldsymbol{\phi}_j^t||^2 \leq \tau \sum_{j=1}^{N}\sum_{\substack{\hat{v}_j(i) \in \mathcal{V}_j(\widetilde{T}(\varepsilon)), \\ T_1+\tau+1 \leq \hat{v}_j(i)}}||\boldsymbol{\phi}_j^{\hat{v}_j(i)} - \boldsymbol{\phi}_j^{\hat{v}_j(i)-1}||^2$$

$$= \tau \sum_{j=1}^{N}\sum_{t=T_1+\tau}^{T_1+\widetilde{T}(\varepsilon)}||\boldsymbol{\phi}_j^{t+1} - \boldsymbol{\phi}_j^t||^2 + \tau \sum_{j=1}^{N}\sum_{t=T_1+\widetilde{T}(\varepsilon)+1}^{T_1+\widetilde{T}(\varepsilon)+\tau-1}||\boldsymbol{\phi}_j^{t+1} - \boldsymbol{\phi}_j^t||^2 \quad (A.88)$$

$$\leq \tau \sum_{j=1}^{N}\sum_{t=T_1+\tau}^{T_1+\widetilde{T}(\varepsilon)}||\boldsymbol{\phi}_j^{t+1} - \boldsymbol{\phi}_j^t||^2 + 4\tau(\tau-1)N\sigma_2{}^2.$$

Similarly, for any $t$ that satisfies $\hat{v}_j(i-1) \leq t < \hat{v}_j(i)$, we have $\boldsymbol{w}_j^t = \boldsymbol{w}_j^{\hat{v}_j(i)-1}$. And for $t \notin \mathcal{V}_j(T)$, we have $||\boldsymbol{w}_j^t - \boldsymbol{w}_j^{t-1}||^2 = 0$. Combing with $\hat{v}_j(i) - \hat{v}_j(i-1) \leq \tau$, we can obtain,

$$\sum_{j=1}^{N}\sum_{t=T_1+\tau}^{T_1+\widetilde{T}(\varepsilon)}||\boldsymbol{w}_j^{\overline{t}_j} - \boldsymbol{w}_j^t||^2 \leq \tau \sum_{j=1}^{N}\sum_{\substack{\hat{v}_j(i) \in \mathcal{V}_j(\widetilde{T}(\varepsilon)), \\ T_1+\tau+1 \leq \hat{v}_j(i)}}||\boldsymbol{w}_j^{\hat{v}_j(i)} - \boldsymbol{w}_j^{\hat{v}_j(i)-1}||^2$$

$$= \tau \sum_{j=1}^{N}\sum_{t=T_1+\tau}^{T_1+\widetilde{T}(\varepsilon)}||\boldsymbol{w}_j^{t+1} - \boldsymbol{w}_j^t||^2 + \tau \sum_{j=1}^{N}\sum_{t=T_1+\widetilde{T}(\varepsilon)+1}^{T_1+\widetilde{T}(\varepsilon)+\tau-1}||\boldsymbol{w}_j^{t+1} - \boldsymbol{w}_j^t||^2$$

$$\leq \tau \sum_{j=1}^{N}\sum_{t=T_1+\tau}^{T_1+\widetilde{T}(\varepsilon)}||\boldsymbol{w}_j^{t+1} - \boldsymbol{w}_j^t||^2 + 4\tau(\tau-1)pN\alpha_1{}^2.$$

$$(A.89)$$

It follows from Eq. (A.85), (A.87), (A.88), (A.89) that,

$$\sum_{t=T_1+\tau}^{T_1+\widetilde{T}(\varepsilon)} d_4^t||\nabla \widetilde{G}^t||^2$$

$$\leq F^{T_1+\tau} - \underline{L} + \frac{4}{\rho_1}\Big(\frac{c_1^{T_1+\tau-2}}{c_1^{T_1+\tau-1}} + \frac{c_1^{T_1+\tau-1}}{c_1^{T_1+\tau}}\Big)M\sigma_1{}^2 + \frac{c_1^{T_1+\tau-1}}{2}M\sigma_1{}^2 + \frac{7}{2\rho_1}M\sigma_3{}^2 + 3d_4^0(c_1^0)^2M\sigma_1{}^2$$

$$+ \frac{4}{\rho_2}\Big(\frac{c_1^{T_1+\tau-2}}{c_1^{T_1+\tau-1}} + \frac{c_1^{T_1+\tau-1}}{c_1^{T_1+\tau}}\Big)N\sigma_2{}^2 + \frac{c_2^{T_1+\tau-1}}{2}N\sigma_2{}^2 + \frac{7}{2\rho_2}N\sigma_4{}^2 + 3\tau d_4^0(c_2^0)^2N\sigma_2{}^2$$

$$+ \frac{c_1^{T_1+\tau}}{2}M\sigma_1{}^2 + \frac{c_2^{T_1+\tau}}{2}N\sigma_2{}^2 + \Big(\frac{2N\sigma_2{}^2}{5\rho_2} + 4d_4^0\big(\frac{2}{\eta_{\boldsymbol{w}}{}^2} + 3NL^2\big)pN\alpha_1{}^2\tau\Big)(\tau-1)$$

$$= \overline{d} + k_d(\tau - 1),$$

$$(A.90)$$

where $\bar{d}$ and $k_d$ are constants. And constant $d_5$ is given by,

$$d_5 = \max\{\tfrac{d_1}{a_6^0}, \tfrac{d_2}{a_5^0}, \tfrac{d_3}{a_5^0}, \tfrac{\frac{30}{\rho_1}+90\rho_1\tau k_1 NL^2}{(1-15\rho_1\tau k_1 NL^2)a_5^0 a_6^0}, \tfrac{30\tau}{\rho_2 a_5^0 a_6^0}\}$$

$$\geq \max\{\tfrac{d_1}{a_6^t}, \tfrac{d_2}{a_5^t}, \tfrac{d_3}{a_5^t}, \tfrac{\frac{30}{\rho_1}+90\rho_1\tau k_1 NL^2}{(1-15\rho_1\tau k_1 NL^2)a_5^t a_6^t}, \tfrac{30\tau}{\rho_2 a_5^t a_6^t}\} \tag{A.91}$$

$$= \tfrac{1}{d_4^t a_5^t a_6^t}.$$

Thus, we can obtain that,

$$\sum_{t=T_1+\tau}^{T_1+\widetilde{T}(\varepsilon)} \frac{1}{d_5 a_5^t a_6^t}||\nabla\widetilde{G}^{T_1+\widetilde{T}(\varepsilon)}||^2 \leq \sum_{t=T_1+\tau}^{T_1+\widetilde{T}(\varepsilon)} \frac{1}{d_5 a_5^t a_6^t}||\nabla\widetilde{G}^t||^2 \leq \sum_{t=T_1+\tau}^{T_1+\widetilde{T}(\varepsilon)} d_4^t||\nabla\widetilde{G}^t||^2 \leq \bar{d}+k_d(\tau-1). \tag{A.92}$$

And it follows from Eq. (A.92) that,

$$||\nabla\widetilde{G}^{T_1+\widetilde{T}(\varepsilon)}||^2 \leq \frac{(\bar{d}+k_d(\tau-1))d_5}{\sum\limits_{t=T_1+\tau}^{T_1+\widetilde{T}(\varepsilon)} \frac{1}{a_5^t a_6^t}}. \tag{A.93}$$

According to the setting of $c_1^t$, $c_2^t$ and Eq. (A.66), (A.67), we have,

$$\frac{1}{a_5^t a_6^t} \geq \frac{1}{(4(\gamma-2)L^2(M\rho_1 + N\rho_2)(t+1)^{\frac{1}{3}} + \frac{\rho_2(N-S)L^2}{2})^2}. \tag{A.94}$$

Summing up $\frac{1}{a_5^t a_6^t}$ from $t = T_1+\tau$ to $t = T_1+\widetilde{T}(\varepsilon)$, it follows that,

$$\sum_{t=T_1+\tau}^{T_1+\widetilde{T}(\varepsilon)} \frac{1}{a_5^t a_6^t} \geq \sum_{t=T_1+\tau}^{T_1+\widetilde{T}(\varepsilon)} \frac{1}{(4(\gamma-2)L^2(M\rho_1+N\rho_2)(t+1)^{\frac{1}{3}} + \frac{\rho_2(N-S)L^2}{2})^2}$$

$$\geq \sum_{t=T_1+\tau}^{T_1+\widetilde{T}(\varepsilon)} \frac{1}{(4(\gamma-2)L^2(M\rho_1+N\rho_2)(t+1)^{\frac{1}{3}} + \frac{\rho_2(N-S)L^2}{2}(t+1)^{\frac{1}{3}})^2} \tag{A.95}$$

$$\geq \frac{(T_1+\widetilde{T}(\varepsilon))^{\frac{1}{3}} - (T_1+\tau)^{\frac{1}{3}}}{(4(\gamma-2)L^2(M\rho_1+N\rho_2) + \frac{\rho_2(N-S)L^2}{2})^2}.$$

The second inequality in Eq. (A.95) is due to that $\forall t \geq T_1 + \tau$, we have,

$$4(\gamma-2)L^2(M\rho_1+N\rho_2)(t+1)^{\frac{1}{3}} + \frac{\rho_2(N-S)L^2}{2} \leq (4(\gamma-2)L^2(M\rho_1+N\rho_2) + \frac{\rho_2(N-S)L^2}{2})(t+1)^{\frac{1}{3}}. \tag{A.96}$$

The last inequality in Eq. (A.95) follows from the fact that $\sum\limits_{t=T_1+\tau}^{T_1+\widetilde{T}(\varepsilon)} \frac{1}{(t+1)^{\frac{2}{3}}} \geq (T_1+\widetilde{T}(\varepsilon))^{\frac{1}{3}} - (T_1+\tau)^{\frac{1}{3}}$.

Thus, plugging Eq. (A.95) into Eq. (A.93), we can obtain:

$$||\nabla\widetilde{G}^{T_1+\widetilde{T}(\varepsilon)}||^2 \leq \frac{(\bar{d}+k_d(\tau-1))d_5}{\sum\limits_{t=T_1+\tau}^{T_1+\widetilde{T}(\varepsilon)} \frac{1}{a_5^t a_6^t}} \leq \frac{(4(\gamma-2)L^2(M\rho_1+N\rho_2) + \frac{\rho_2(N-S)L^2}{2})^2(\bar{d}+k_d(\tau-1))d_5}{(T_1+\widetilde{T}(\varepsilon))^{\frac{1}{3}} - (T_1+\tau)^{\frac{1}{3}}}. \tag{A.97}$$

According to the definition of $\widetilde{T}(\varepsilon)$, we have:

$$T_1+\widetilde{T}(\varepsilon) \geq \left(\frac{4(4(\gamma-2)L^2(M\rho_1+N\rho_2) + \frac{\rho_2(N-S)L^2}{2})^2(\bar{d}+k_d(\tau-1))d_5}{\varepsilon^2} + (T_1+\tau)^{\frac{1}{3}}\right)^3. \tag{A.98}$$

Combining the definition of $\nabla G^t$ and $\nabla \widetilde{G}^t$ with trigonometric inequality, we then get:

$$||\nabla G^t|| - ||\nabla \widetilde{G}^t|| \le ||\nabla G^t - \nabla \widetilde{G}^t|| \le \sqrt{\sum_{l=1}^{|\mathbf{A}^t|} ||c_1^{t-1}\lambda_l^t||^2 + \sum_{j=1}^{N} ||c_2^{t-1}\phi_j^t||^2}. \qquad (A.99)$$

Denoting constant $d_6$ as $d_6 = 4(\gamma - 2)L^2(M\rho_1 + N\rho_2)$. If $t > (\frac{4M\sigma_1^2}{\rho_1^2} + \frac{4N\sigma_2^2}{\rho_2^2})^3 \frac{1}{\varepsilon^6}$, then we have

$\sqrt{\sum_{l=1}^{|\mathbf{A}^t|} ||c_1^{t-1}\lambda_l^t||^2 + \sum_{j=1}^{N} ||c_2^{t-1}\phi_j^t||^2} \le \frac{\varepsilon}{2}$. Combining it with Eq. (A.98), we can conclude that there exists a

$$T(\varepsilon) \sim \mathcal{O}(\max\{(\frac{4M\sigma_1^2}{\rho_1^2} + \frac{4N\sigma_2^2}{\rho_2^2})^3 \frac{1}{\varepsilon^6}, (\frac{4(d_6 + \frac{\rho_2(N-S)L^2}{2})^2(\bar{d} + k_d(\tau - 1))d_5}{\varepsilon^2} + (T_1 + \tau)^{\frac{1}{3}})^3\}), \tag{A.100}$$

such that $||\nabla G^t|| \le ||\nabla \widetilde{G}^t|| + \sqrt{\sum_{l=1}^{|\mathbf{A}^t|} ||c_1^{t-1}\lambda_l^t||^2 + \sum_{j=1}^{N} ||c_2^{t-1}\phi_j^t||^2} \le \varepsilon$, which concludes our proof.

# B  Time Efficiency Comparison

In a distributed communication network, the communication and computation delays of workers are inevitable. Due to differences in system configuration, communication and computation delays vary across different workers, the existence of lagging workers (*i.e.*, stragglers and stale workers) is inevitable. For synchronous algorithm, it will lead to idling and wastage of computing resources since the master only updates the variables after receiving updates from all workers (as illustrated in Figure B1). Different from the synchronous algorithm, the asynchronous algorithm allows the master updates the variables whenever it receives updates from a subset of workers, which is more efficient. In this section, we compare the time for our asynchronous and synchronous algorithms to return an $\varepsilon$-stationary point.

**Fact 1** *Let $\mathcal{T}_1$ and $T_1(\varepsilon)$ denote the convergence time and iterations for the proposed asynchronous algorithm. Let $\mathcal{T}_2$ and $T_2(\varepsilon)$ denote the convergence time and iterations for the synchronous algorithm, we have,*

$$\frac{\mathcal{T}_1}{\mathcal{T}_2} = \frac{T_1(\varepsilon) \times S}{T_2(\varepsilon) \times (\frac{\widetilde{d}}{\hat{d}_1} + \frac{\widetilde{d}}{\hat{d}_2} \cdots + \frac{\widetilde{d}}{\hat{d}_N})}, \tag{B.101}$$

*where $\widetilde{d}$ is the maximum (computation + communication) delay of all workers.*

### Proof of Fact 1:

In this part, we do not consider the delays of master. Suppose that there are $N$ workers in a distributed system and the number of active workers is $S$. For brevity, we assume the delay for each work remains the same during the iteration. Let $[\hat{d}_1, \hat{d}_2, \cdots \hat{d}_N] \in \mathbb{R}^N$ denote the (computation + communication) delay for $N$ workers. And we define the maximum delay of all workers as $\widetilde{d}$. For the $j^{\text{th}}$ worker, it has communicated with the master $\frac{\mathcal{T}_1}{\hat{d}_j}$ times during time $\mathcal{T}_1$. Thus, for time $\mathcal{T}_1$, it satisfies that,

$$\frac{\mathcal{T}_1}{\hat{d}_1} + \frac{\mathcal{T}_1}{\hat{d}_2} + \cdots + \frac{\mathcal{T}_1}{\hat{d}_N} = T_1(\varepsilon) \times S. \tag{B.102}$$

For the synchronous algorithm, the time $\mathcal{T}_2$ needs to satisfy that:

$$\mathcal{T}_2 = T_2(\varepsilon) \times \widetilde{d}. \tag{B.103}$$

Thus, we have that,

$$\frac{\mathcal{T}_1}{\mathcal{T}_2} = \frac{T_1(\varepsilon) \times S}{T_2(\varepsilon) \times (\frac{\widetilde{d}}{\hat{d}_1} + \frac{\widetilde{d}}{\hat{d}_2} \cdots + \frac{\widetilde{d}}{\hat{d}_N})}. \tag{B.104}$$

For the special case that the asynchronous algorithm degrades to the synchronous algorithm, *i.e.*, the master is required to update its parameters only after receiving the updates from all workers, we have $S = N$, $T_1(\varepsilon) = T_2(\varepsilon)$ and $\hat{d}_j = \widetilde{d}, \forall j = 1, \cdots, N$ since all workers are required to wait the slowest worker. Back to Eq. (B.104), we can obtain that $\frac{\mathcal{T}_1}{\mathcal{T}_2} = 1$.

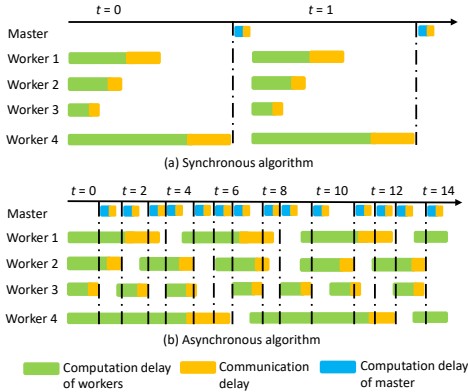

(a) Synchronous algorithm

(b) Asynchronous algorithm

■ Computation delay   ■ Communication   ■ Computation delay
of workers      delay         of master

Figure B1: The illustration of synchronous and asynchronous algorithms. $t$ represents the number of iterations. In the asynchronous algorithm (at the bottom), the master begins to update its parameters after receiving the update from one worker.

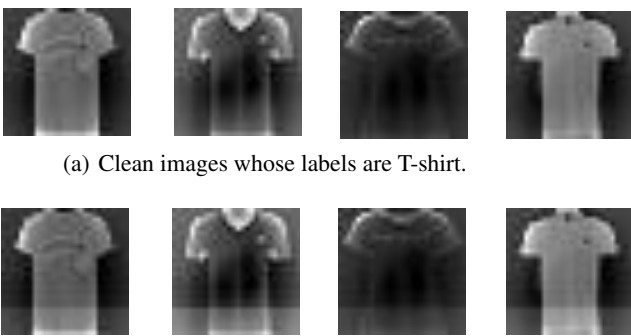

(a) Clean images whose labels are T-shirt.

(b) Attacked images whose target labels are Pullover.

Figure C1: Backdoor attacks on Fashion MNIST dataset. Through adding triggers on local patch of clean images, the attacked images are misclassified as the target labels.

## C    Experiments

In this section, we present the detailed results of our experiments. We first give a detailed description of the datasets and baseline methods used in our experiments.

### C.1    Datasets and Baseline Methods

In this section, we provide a detailed introduction to datasets and baseline methods. The number of workers and categories of every dataset are summarized in Table C1.

**Datasets**:

1. **SHL dataset**: The SHL dataset was collected using four cellphones on four body locations where people usually carry cellphones. The SHL dataset provides multimodal locomotion and transportation data collected in real-world settings using eight various modes of transportation. We separated the data into six workers with varied proportions based on the four body locations of smartphones to imitate the different tendencies of workers (users) in positioning cellphones.

2. **Person Activity dataset**: Data contains recordings of five participants performing eleven different activities. Each participant wears four sensors in four different body locations (ankle left, ankle right, belt, and chest) while performing the activities. Each participant corresponds to one worker in the experiment.

Table C1: The number of workers and categories of datasets

|  | SHL | Person Activity | SC-MA | Fashion MNIST |
|---|---|---|---|---|
| Number of workers | 6 | 5 | 15 | 3 |
| Number of categories | 8 | 11 | 7 | 3 |

Table C2: Model structure that used for SHL dataset.

| No. | Layer type | Number of neurons | Activation |
|---|---|---|---|
| 1 | Fully-connected | 96 | ReLU |
| 2 | Fully-connected | 48 | ReLU |
| 3 | Fully-connected | 24 | ReLU |
| 4 | Output | 8 | Softmax |

3. **Single Chest-Mounted Accelerometer dataset**: Data was collected from fifteen participants engaged in seven distinct activities. Each participant (worker) wears an accelerometer mounted on the chest.

4. **Fashion MNIST**: Similar to MNIST, Fashion MNIST is a dataset where images are grouped into ten categories of clothing. The subset of the data labeled with Pullover, Shirt and T-shirt are extracted as three workers and each worker consists of one class of clothing.

**Baseline Methods**:

1. **Ind$_j$**: It learns the model from an individual worker $j$.

2. **Mix$_{\text{Even}}$**: It learns the model from all workers with even weights using the proposed distributed algorithm.

3. **FedAvg**: It learns the model from all workers with even weights. It aggregates the local model parameters from workers through using model averaging.

4. **AFL**: It aims to address the fairness issues in federated learning. AFL adopts the strategy that alternately update the model parameters and the weight of each worker through alternating projected gradient descent/ascent.

5. **DRFA-Prox**: It aims to mitigate the data heterogeneity issue in federated learning. Compared with AFL, it is communication-efficient which requires fewer communication rounds. Moreover, it leverages the prior distribution and introduces it as a regularizer in the objective function.

6. **ASPIRE-EASE(-)**: The proposed ASPIRE-EASE without asynchronous setting.

7. **ASPIRE-CP**: The proposed ASPIRE with cutting plane method.

8. **ASPIRE-EASE$_{\text{per}}$**: The proposed ASPIRE-EASE with periodic communication.

## C.2 Training Details

In our empirical studies, since the downstream tasks are multi-class classification, the cross entropy loss is used on each worker (*i.e.*, $\mathcal{L}_j(\cdot), \forall j$). For SHL, Person Activity and SM-AC datasets, we adopt the deep multilayer perceptron [49] as the base model. Specifically, we exhibit the model structures that are used for SHL, Person Activity and SM-AC datasets in Table C2, Table C3 and Table C4. And we use the same logistic regression model as in [35, 16] for the Fashion MNIST dataset. In the experiments, we use the SGD optimizer for model training, and we implement our model with PyTorch and conduct all the experiments on a server with two TITAN V GPUs.

## C.3 Additional Results

We first show the detailed experiment settings about robustness against malicious attacks. We conduct experiments in the setting where there are malicious workers which attempt to mislead the model

Table C3: Model structure that used for Peson Activity dataset.

| No. | Layer type | Number of neurons | Activation |
|-----|-----------|-------------------|------------|
| 1 | Fully-connected | 64 | ReLU |
| 2 | Fully-connected | 32 | ReLU |
| 3 | Fully-connected | 16 | ReLU |
| 4 | Output | 11 | Softmax |

Table C4: Model structure that used for SC-MA dataset.

| No. | Layer type | Number of neurons | Activation |
|-----|-----------|-------------------|------------|
| 1 | Fully-connected | 32 | ReLU |
| 2 | Fully-connected | 16 | ReLU |
| 3 | Output | 7 | Softmax |

training process. The backdoor attack [1, 48] is adopted in the experiment which aims to bury the backdoor during the training phase of the model. The buried backdoor will be activated by the preset trigger. When the backdoor is not activated, the attacked model performs normally to other local models. When the backdoor is activated, the output of the attack model is misled as the target label which is pre-specified by the attacker. In the experiment, one worker is chosen as the malicious worker. We add triggers to a small part of the data and change their primal labels to target labels (*e.g.*, triggers are added on the local patch of clean images on the Fashion MNIST dataset, which are shown in Figure C1). Furthermore, the malicious worker can purposefully raise the training loss to mislead the master. To evaluate the model's robustness against malicious attacks, following [14], we calculate the success attack rate of the backdoor attacks. The success attack rate can be calculated by checking how many instances in the backdoor dataset can be misled into the target labels. The lower success attack rate indicates better robustness against backdoor attacks. The success attack rates of different models on three datasets are reported in Table 2. In Table 2, we observe that AFL can be attacked easily since it could assign higher weights to malicious workers. Compared to AFL, FedAvg and $\mathrm{Mix_{Even}}$ achieve relatively lower success attack rates since they assign equal weights to the malicious workers and other workers. DRFA-Prox can achieve even lower success attack rates since it can leverage the prior distribution to assign lower weights for malicious workers. The proposed ASPIRE-EASE achieves the lowest success attack rates since it can leverage the prior distribution more effectively. Specifically, it will assign lower weights to malicious workers with tight theoretical guarantees.

We also report additional experiment results on SHL and Fashion MNIST datasets. We first show that the proposed ASPIRE-EASE can flexibly control the level of robustness by adjusting $\Gamma$, which is presented in Figure C2. It is seen that the robustness of ASPIRE-EASE can be gradually enhanced when $\Gamma$ increases. Next, the comparison of convergence speed by considering different communication and computation delays for each worker is exhibited in Figure C3. We can observe that the proposed ASPIRE-EASE is generally the most efficient since the ASPIRE is an asynchronous algorithm and the proposed EASE is effective. Finally, to further demonstrate the efficiency of EASE, we compare ASPIRE-EASE with ASPIRE-CP concerning the number of cutting planes used during the training. As shown in Theorem 1, a smaller number of cutting planes (which corresponds to a smaller $M$) will need fewer iterations to achieve convergence. In Figure C4, we can see that ASPIRE-EASE uses fewer cutting planes and thus is more efficient.

## D   Solve PD-DRO in Centralized Manner

Considering to solve the PD-DRO problem in Eq. (4) in centralized manner, we can rewrite the problem in Eq. (4) as:

$$\min_{\boldsymbol{w}\in\mathcal{W}} \max_{\mathbf{p}\in\mathcal{P}} \sum_{j=1}^{N} p_j f_j(\boldsymbol{w}) \tag{D.105}$$

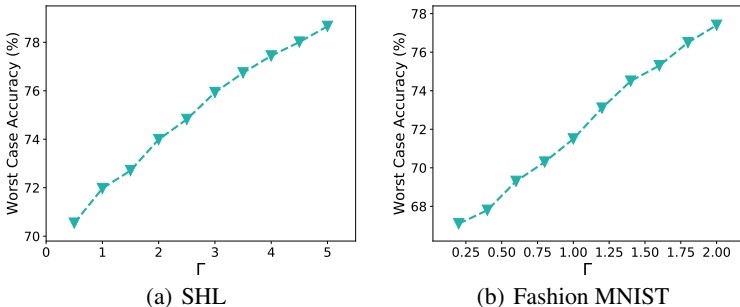

(a) SHL        (b) Fashion MNIST

Figure C2: $\Gamma$ control the degree of robustness (worst case performance in the problem) on (a) SHL, (b) Fashion MNIST datasets.

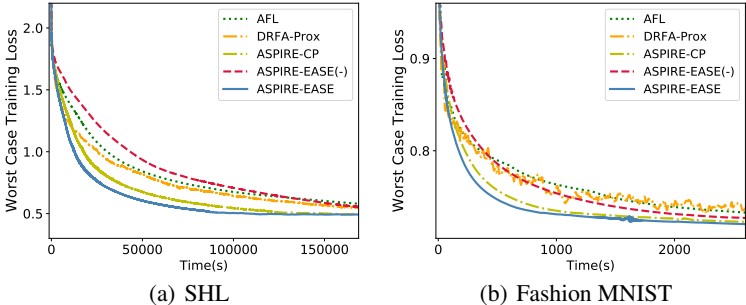

(a) SHL        (b) Fashion MNIST

Figure C3: Comparison of the convergence time on worst case worker on (a) SHL, (b) Fashion MNIST datasets.

where $\boldsymbol{w} \in \mathbb{R}^p$ is the model parameter. Utilizing the cutting plane method, we can obtain the approximate problem of Eq. (D.105),

$$\min_{\boldsymbol{w} \in \boldsymbol{\mathcal{W}}, h \in \boldsymbol{\mathcal{H}}} \quad h \tag{D.106}$$

$$\text{s.t.} \sum_{j=1}^{N} (\bar{p} + a_{l,j}) f_j(\boldsymbol{w}) - h \leq 0, \forall \boldsymbol{a}_l \in \mathbf{A}^t,$$

$$\text{var.} \quad \boldsymbol{w}, h.$$

Thus, the Lagrangian function of Eq. (D.106) can be written as:

$$L_p(\boldsymbol{w}, h, \{\lambda_l\}) = h + \sum_{l=1}^{|\mathbf{A}^t|} \lambda_l \Big( \sum_{j=1}^{N} (\bar{p} + a_{l,j}) f_j(\boldsymbol{w}) - h \Big). \tag{D.107}$$

Following [52], the regularized version of (D.107) is employed to update all variables as follows,

$$\widetilde{L}_p(\boldsymbol{w}, h, \{\lambda_l\}) = h + \sum_{l=1}^{|\mathbf{A}^t|} \lambda_l \Big( \sum_{j=1}^{N} (\bar{p} + a_{l,j}) f_j(\boldsymbol{w}) - h \Big) - \sum_{l=1}^{|\mathbf{A}^t|} \frac{c_1^t}{2} ||\lambda_l||^2, \tag{D.108}$$

where $c_1^t$ denotes the regularization term in $(t+1)^{\text{th}}$ iteration. To avoid enumerating the whole dataset, the mini-batch loss $\hat{f}_j(\boldsymbol{w}) = \sum_{i=1}^{m} \frac{1}{m} \mathcal{L}_j(\mathbf{x}_j^i, y_j^i; \boldsymbol{w})$ can be used, where $m$ is the mini-batch size. It is evident that $\mathbb{E}[\hat{f}_j(\boldsymbol{w})] = f_j(\boldsymbol{w})$ and $\mathbb{E}[\nabla \hat{f}_j(\boldsymbol{w})] = \nabla f_j(\boldsymbol{w})$. The centralized algorithm, which aims to solve problem in Eq. (4) in centralized manner, proceeds as follows in $(t+1)^{\text{th}}$ iteration:

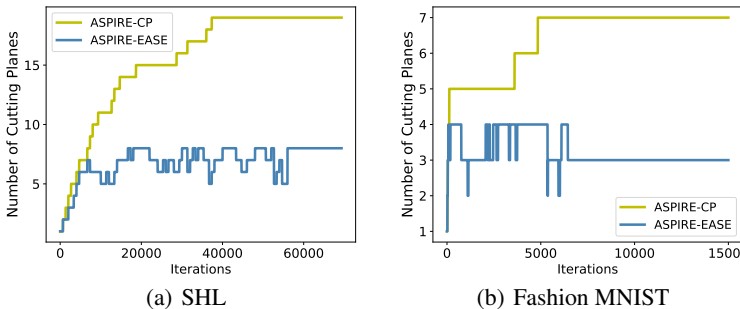

(a) SHL  (b) Fashion MNIST

Figure C4: Comparison of ASPIRE-CP and ASPIRE-EASE regarding the number of cutting planes on (a) SHL, (b) Fashion MNIST datasets. ASPIRE-CP represents ASPIRE with cutting plane method.

1. Updating the model parameter $w$ as follows,

$$w^{t+1} = \mathcal{P}_{\mathcal{W}}(w^t - \eta_w^t \nabla_w \widetilde{L}_p(w^t, h^t, \{\lambda_l^t\})), \tag{D.109}$$

where $\eta_w^t$ represents the step-size and $\mathcal{P}_{\mathcal{W}}$ represents the projection onto the convex set $\mathcal{W}$.

2. Updating the additional variable $h$ as follows,

$$h^{t+1} = \mathcal{P}_{\mathcal{H}}(h^t - \eta_h^t \nabla_h \widetilde{L}_p(w^{t+1}, h^t, \{\lambda_l^t\})), \tag{D.110}$$

where $\eta_h^t$ represents the step-size and $\mathcal{P}_{\mathcal{H}}$ represents the projection onto the convex set $\mathcal{H}$.

3. Updating the dual variable $\lambda_l$ as follows,

$$\lambda_l^{t+1} = \mathcal{P}_{\Lambda}(\lambda_l^t + \rho_1 \nabla_{\lambda_l} \widetilde{L}_p(w^{t+1}, h^{t+1}, \{\lambda_l^t\})), \ l = 1, \cdots, |\mathbf{A}^t|, \tag{D.111}$$

where $\rho_1$ represents the step-size and $\mathcal{P}_{\Lambda}$ represents the projection onto the convex set $\Lambda$.

Then, during $T_1$ iterations, EASE is utilized to update the set $\mathbf{A}^{t+1}$ every $k$ iterations.

**Definition D.1** *Following [52, 32, 53], the stationarity gap at $t^{th}$ iteration is defined as,*

$$\nabla G^t = \begin{bmatrix} \frac{1}{\eta_w^t}(w^t - \mathcal{P}_{\mathcal{W}}(w^t - \eta_w^t \nabla_w L_p(w^t, h^t, \{\lambda_l^t\}))) \\ \frac{1}{\eta_h^t}(h^t - \mathcal{P}_{\mathcal{H}}(h^t - \eta_h^t \nabla_h L_p(w^t, h^t, \{\lambda_l^t\}))) \\ \{\frac{1}{\rho_1}(\lambda_l^t - \mathcal{P}_{\Lambda}(\lambda_l^t + \rho_1 \nabla_{\lambda_l} L_p(w^t, h^t, \{\lambda_l^t\})))\} \end{bmatrix}. \tag{D.112}$$

*And we also define:*

$$\begin{aligned}
(\nabla G^t)_w &= \frac{1}{\eta_w^t}(w^t - \mathcal{P}_{\mathcal{W}}(w^t - \eta_w^t \nabla_w L_p(w^t, h^t, \{\lambda_l^t\}))), \\
(\nabla G^t)_h &= \frac{1}{\eta_h^t}(h^t - \mathcal{P}_{\mathcal{H}}(h^t - \eta_h^t \nabla_h L_p(w^t, h^t, \{\lambda_l^t\}))), \\
(\nabla G^t)_{\lambda_l} &= \frac{1}{\rho_1}(\lambda_l^t - \mathcal{P}_{\Lambda}(\lambda_l^t + \rho_1 \nabla_{\lambda_l} L_p(w^t, h^t, \{\lambda_l^t\}))).
\end{aligned} \tag{D.113}$$

*It follows that:*

$$||\nabla G^t||^2 = ||(\nabla G^t)_w||^2 + ||(\nabla G^t)_h||^2 + \sum_{l=1}^{|\mathbf{A}^t|} ||(\nabla G^t)_{\lambda_l}||^2. \tag{D.114}$$

**Definition D.2** *At $t^{th}$ iteration, the stationarity gap w.r.t $\widetilde{L}_p(w, h, \{\lambda_l\})$ is defined as:*

$$\nabla \widetilde{G}^t = \begin{bmatrix} \frac{1}{\eta_w^t}(w^t - \mathcal{P}_{\mathcal{W}}(w^t - \eta_w^t \nabla_w \widetilde{L}_p(w^t, h^t, \{\lambda_l^t\}))) \\ \frac{1}{\eta_h^t}(h^t - \mathcal{P}_{\mathcal{H}}(h^t - \eta_h^t \nabla_h \widetilde{L}_p(w^t, h^t, \{\lambda_l^t\}))) \\ \{\frac{1}{\rho_1}(\lambda_l^t - \mathcal{P}_{\Lambda}(\lambda_l^t + \rho_1 \nabla_{\lambda_l} \widetilde{L}_p(w^t, h^t, \{\lambda_l^t\})))\} \end{bmatrix}. \tag{D.115}$$

*And we also define:*

$$(\nabla\widetilde{G}^t)_{\boldsymbol{w}} = \frac{1}{\eta_{\boldsymbol{w}}^t}(\boldsymbol{w}^t - \mathcal{P}_{\boldsymbol{\mathcal{W}}}(\boldsymbol{w}^t - \eta_{\boldsymbol{w}}^t \nabla_{\boldsymbol{w}}\widetilde{L}_p(\boldsymbol{w}^t, h^t, \{\lambda_l^t\}))),$$

$$(\nabla\widetilde{G}^t)_h = \frac{1}{\eta_h^t}(h^t - \mathcal{P}_{\boldsymbol{\mathcal{H}}}(h^t - \eta_h^t \nabla_h\widetilde{L}_p(\boldsymbol{w}^t, h^t, \{\lambda_l^t\}))), \tag{D.116}$$

$$(\nabla\widetilde{G}^t)_{\lambda_l} = \frac{1}{\rho_1}(\lambda_l^t - \mathcal{P}_{\boldsymbol{\Lambda}}(\lambda_l^t + \rho_1 \nabla_{\lambda_l}\widetilde{L}_p(\boldsymbol{w}^t, h^t, \{\lambda_l^t\}))).$$

*It follows that:*

$$||\nabla\widetilde{G}^t||^2 = ||(\nabla\widetilde{G}^t)_{\boldsymbol{w}}||^2 + ||(\nabla\widetilde{G}^t)_h||^2 + \sum_{l=1}^{|\mathbf{A}^t|} ||(\nabla\widetilde{G}^t)_{\lambda_l}||^2. \tag{D.117}$$

**Assumption D.1** $L_p$ *has Lipschitz continuous gradients. We assume that there exists $L > 0$ satisfying that,*

$$||\nabla_\theta L_p(\boldsymbol{w}, h, \{\lambda_l\}) - \nabla_\theta L_p(\hat{\boldsymbol{w}}, \hat{h}, \{\hat{\lambda}_l\})|| \leq L||[\boldsymbol{w} - \hat{\boldsymbol{w}}; h - \hat{h}; \boldsymbol{\lambda}_{\mathrm{cat}} - \hat{\boldsymbol{\lambda}}_{\mathrm{cat}}]||,$$

*where $\theta \in \{\boldsymbol{w}, h, \{\lambda_l\}\}$, $[;]$ represents the concatenation and $\boldsymbol{\lambda}_{\mathrm{cat}} - \hat{\boldsymbol{\lambda}}_{\mathrm{cat}} = [\lambda_1 - \hat{\lambda}_1; \cdots; \lambda_{|\mathbf{A}^t|} - \hat{\lambda}_{|\mathbf{A}^t|}] \in \mathbb{R}^{|\mathbf{A}^t|}$.*

**Setting D.1** $|\mathbf{A}^t| \leq M, \forall t$, *i.e., an upper bound is set for the number of cutting planes.*

**Setting D.2** $c_1^t = \frac{1}{\rho_1(t+1)^{\frac{1}{4}}} \geq \underline{c}_1$ *is nonnegative non-increasing sequence, where $\underline{c}_1 > 0$ meets $\underline{c}_1{}^2 \leq \frac{\varepsilon^2}{4M}$.*

**Theorem D.1** *Suppose Assumption D.1 holds. We set $\eta_{\boldsymbol{w}}^t = \eta_h^t = \frac{2}{L + \rho_1|\mathbf{A}^t|L^2 + 8\frac{|\mathbf{A}^t|\gamma L^2}{\rho_1(c_1^t)^2}}$, and we set constant $\rho_1 \leq \frac{2}{L + 2c_1^0}$. For a given $\varepsilon$, we have:*

$$T(\varepsilon) \sim \mathcal{O}(\max\{(\frac{16(\gamma - 2)L^2 M \rho_1 \bar{d} d_5}{\varepsilon^2} + (T_1 + 2)^{\frac{1}{2}})^2, \frac{16M^2\sigma_1^4}{\rho_1^4}\frac{1}{\varepsilon^4}\}), \tag{D.118}$$

*where $\sigma_1$, $\gamma$, $\bar{d}$, $d_5$ and $T_1$ are constants.*

**Lemma D.1** *Suppose Assumption D.1 holds, we have:*

$$L_p(\boldsymbol{w}^{t+1}, h^t, \{\lambda_l^t\}) - L_p(\boldsymbol{w}^t, h^t, \{\lambda_l^t\}) \leq (\frac{L}{2} - \frac{1}{\eta_{\boldsymbol{w}}^t})||\boldsymbol{w}^{t+1} - \boldsymbol{w}^t||^2, \tag{D.119}$$

$$L_p(\boldsymbol{w}^{t+1}, h^{t+1}, \{\lambda_l^t\}) - L_p(\boldsymbol{w}^{t+1}, h^t, \{\lambda_l^t\}) \leq (\frac{L}{2} - \frac{1}{\eta_h^t})||h^{t+1} - h^t||^2. \tag{D.120}$$

*Proof:*

According to Assumption D.1, we have,

$$L_p(\boldsymbol{w}^{t+1}, h^t, \{\lambda_l^t\}) - L_p(\boldsymbol{w}^t, h^t, \{\lambda_l^t\})$$
$$\leq \langle \nabla_{\boldsymbol{w}} L_p(\boldsymbol{w}^t, h^t, \{\lambda_l^t\}), \boldsymbol{w}^{t+1} - \boldsymbol{w}^t \rangle + \frac{L}{2}||\boldsymbol{w}^{t+1} - \boldsymbol{w}^t||^2. \tag{D.121}$$

According to the optimal condition for Eq. (D.109) and $\nabla_{\boldsymbol{w}}\widetilde{L}_p(\boldsymbol{w}^t, h^t, \{\lambda_l^t\}) = \nabla_{\boldsymbol{w}}L_p(\boldsymbol{w}^t, h^t, \{\lambda_l^t\})$, we have,

$$\langle \boldsymbol{w}^{t+1} - \boldsymbol{w}^t, \nabla_{\boldsymbol{w}}L_p(\boldsymbol{w}^t, h^t, \{\lambda_l^t\}) \rangle \leq -\frac{1}{\eta_{\boldsymbol{w}}^t}||\boldsymbol{w}^{t+1} - \boldsymbol{w}^t||^2. \tag{D.122}$$

Combining Eq. (D.121) with Eq. (D.122), we have that,

$$L_p(\boldsymbol{w}^{t+1}, h^t, \{\lambda_l^t\}) - L_p(\boldsymbol{w}^t, h^t, \{\lambda_l^t\}) \leq (\frac{L}{2} - \frac{1}{\eta_{\boldsymbol{w}}^t})||\boldsymbol{w}^{t+1} - \boldsymbol{w}^t||^2.$$

Similar to Eq. (D.119), we can easily have Eq. (D.120).

**Lemma D.2** *Suppose Assumption D.1 holds, $\forall t \geq T_1$, we have:*

$$L_p(\boldsymbol{w}^{t+1}, h^{t+1}, \{\lambda_l^{t+1}\}) - L_p(\boldsymbol{w}^t, h^t, \{\lambda_l^t\})$$

$$\leq (\tfrac{L}{2} - \tfrac{1}{\eta_w^t} + \tfrac{|\mathbf{A}^t|L^2}{2a_1})||\boldsymbol{w}^{t+1} - \boldsymbol{w}^t||^2 + (\tfrac{L}{2} - \tfrac{1}{\eta_h^t} + \tfrac{|\mathbf{A}^t|L^2}{2a_1})||h^{t+1} - h^t||^2$$

$$+ (\tfrac{a_1}{2} - \tfrac{c_1^{t-1} - c_1^t}{2} + \tfrac{1}{2\rho_1}) \sum_{l=1}^{|\mathbf{A}^t|} ||\lambda_l^{t+1} - \lambda_l^t||^2 + \tfrac{c_1^{t-1}}{2} \sum_{l=1}^{|\mathbf{A}^t|} (||\lambda_l^{t+1}||^2 - ||\lambda_l^t||^2) + \tfrac{1}{2\rho_1} \sum_{l=1}^{|\mathbf{A}^t|} ||\lambda_l^t - \lambda_l^{t-1}||^2. \tag{D.123}$$

*Proof:*

According to Eq. (D.111), in $(t+1)^{\text{th}}$ iteration, $\forall \lambda \in \boldsymbol{\Lambda}$, it follows that,

$$\left\langle \lambda_l^{t+1} - \lambda_l^t - \rho_1 \nabla_{\lambda_l} \widetilde{L}_p(\boldsymbol{w}^{t+1}, h^{t+1}, \{\lambda_l^t\}), \lambda - \lambda_l^{t+1} \right\rangle \geq 0. \tag{D.124}$$

Let $\lambda = \lambda_l^t$, we can obtain,

$$\left\langle \nabla_{\lambda_l} \widetilde{L}_p(\boldsymbol{w}^{t+1}, h^{t+1}, \{\lambda_l^t\}) - \frac{1}{\rho_1}(\lambda_l^{t+1} - \lambda_l^t), \lambda_l^t - \lambda_l^{t+1} \right\rangle \leq 0. \tag{D.125}$$

Likewise, in $t^{\text{th}}$ iteration, we have that,

$$\left\langle \nabla_{\lambda_l} \widetilde{L}_p(\boldsymbol{w}^t, h^t, \{\lambda_l^{t-1}\}) - \frac{1}{\rho_1}(\lambda_l^t - \lambda_l^{t-1}), \lambda_l^{t+1} - \lambda_l^t \right\rangle \leq 0. \tag{D.126}$$

$\forall t \geq T_1$, since $\widetilde{L}_p(\boldsymbol{w}, h, \{\lambda_l\})$ is concave with respect to $\lambda_l$, we have,

$$\widetilde{L}_p(\boldsymbol{w}^{t+1}, h^{t+1}, \{\lambda_l^{t+1}\}) - \widetilde{L}_p(\boldsymbol{w}^{t+1}, h^{t+1}, \{\lambda_l^t\})$$

$$\leq \sum_{l=1}^{|\mathbf{A}^t|} \left\langle \nabla_{\lambda_l} \widetilde{L}_p(\boldsymbol{w}^{t+1}, h^{t+1}, \{\lambda_l^t\}), \lambda_l^{t+1} - \lambda_l^t \right\rangle$$

$$\leq \sum_{l=1}^{|\mathbf{A}^t|} (\left\langle \nabla_{\lambda_l} \widetilde{L}_p(\boldsymbol{w}^{t+1}, h^{t+1}, \{\lambda_l^t\}) - \nabla_{\lambda_l} \widetilde{L}_p(\boldsymbol{w}^t, h^t, \{\lambda_l^{t-1}\}), \lambda_l^{t+1} - \lambda_l^t \right\rangle$$

$$+ \tfrac{1}{\rho_1} \left\langle \lambda_l^t - \lambda_l^{t-1}, \lambda_l^{t+1} - \lambda_l^t \right\rangle). \tag{D.127}$$

Denoting $\boldsymbol{v}_{1,l}^{t+1} = \lambda_l^{t+1} - \lambda_l^t - (\lambda_l^t - \lambda_l^{t-1})$, we have,

$$\sum_{l=1}^{|\mathbf{A}^t|} (\left\langle \nabla_{\lambda_l} \widetilde{L}_p(\boldsymbol{w}^{t+1}, h^{t+1}, \{\lambda_l^t\}) - \nabla_{\lambda_l} \widetilde{L}_p(\boldsymbol{w}^t, h^t, \{\lambda_l^{t-1}\}), \lambda_l^{t+1} - \lambda_l^t \right\rangle$$

$$= \sum_{l=1}^{|\mathbf{A}^t|} \left\langle \nabla_{\lambda_l} \widetilde{L}_p(\boldsymbol{w}^{t+1}, h^{t+1}, \{\lambda_l^t\}) - \nabla_{\lambda_l} \widetilde{L}_p(\boldsymbol{w}^t, h^t, \{\lambda_l^t\}), \lambda_l^{t+1} - \lambda_l^t \right\rangle (1a)$$

$$+ \sum_{l=1}^{|\mathbf{A}^t|} \left\langle \nabla_{\lambda_l} \widetilde{L}_p(\boldsymbol{w}^t, h^t, \{\lambda_l^t\}) - \nabla_{\lambda_l} \widetilde{L}_p(\boldsymbol{w}^t, h^t, \{\lambda_l^{t-1}\}), \boldsymbol{v}_{1,l}^{t+1} \right\rangle (1b)$$

$$+ \sum_{l=1}^{|\mathbf{A}^t|} \left\langle \nabla_{\lambda_l} \widetilde{L}_p(\boldsymbol{w}^t, h^t, \{\lambda_l^t\}) - \nabla_{\lambda_l} \widetilde{L}_p(\boldsymbol{w}^t, h^t, \{\lambda_l^{t-1}\}), \lambda_l^t - \lambda_l^{t-1} \right\rangle (1c). \tag{D.128}$$

We firstly focus on $(1a)$ in Eq. (D.128). According to the Cauchy-Schwarz inequality and Assumption D.1, we have,

$$\sum_{l=1}^{|\mathbf{A}^t|} \left\langle \nabla_{\lambda_l} \widetilde{L}_p(\boldsymbol{w}^{t+1}, h^{t+1}, \{\lambda_l^t\}) - \nabla_{\lambda_l} \widetilde{L}_p(\boldsymbol{w}^t, h^t, \{\lambda_l^t\}), \lambda_l^{t+1} - \lambda_l^t \right\rangle$$

$$\leq \sum_{l=1}^{|\mathbf{A}^t|} (\tfrac{L^2}{2a_1}(||\boldsymbol{w}^{t+1} - \boldsymbol{w}^t||^2 + ||h^{t+1} - h^t||^2) + \tfrac{a_1}{2}||\lambda_l^{t+1} - \lambda_l^t||^2$$

$$+ \tfrac{c_1^{t-1} - c_1^t}{2}(||\lambda_l^{t+1}||^2 - ||\lambda_l^t||^2) - \tfrac{c_1^{t-1} - c_1^t}{2}||\lambda_l^{t+1} - \lambda_l^t||^2), \tag{D.129}$$

where $a_1 > 0$ is a constant. Secondly, according to Cauchy-Schwarz inequality we write $(1b)$ in Eq. (D.128) as,

$$\sum_{l=1}^{|\mathbf{A}^t|} \left\langle \nabla_{\lambda_l} \widetilde{L}_p(\boldsymbol{w}^t, h^t, \{\lambda_l^t\}) - \nabla_{\lambda_l} \widetilde{L}_p(\boldsymbol{w}^t, h^t, \{\lambda_l^{t-1}\}), \boldsymbol{v}_{1,l}^{t+1} \right\rangle$$

$$\leq \sum_{l=1}^{|\mathbf{A}^t|} \left( \frac{a_2}{2} || \nabla_{\lambda_l} \widetilde{L}_p(\boldsymbol{w}^t, h^t, \{\lambda_l^t\}) - \nabla_{\lambda_l} \widetilde{L}_p(\boldsymbol{w}^t, h^t, \{\lambda_l^{t-1}\}) ||^2 + \frac{1}{2a_2} ||\boldsymbol{v}_{1,l}^{t+1}||^2 \right), \tag{D.130}$$

where $a_2 > 0$ is a constant. Then, we focus on the $(1c)$ in Eq. (D.128). Denoting $L_1' = L + c_1^0$, according to Assumption D.1, trigonometric inequality and the strong concavity of $\widetilde{L}_p(\boldsymbol{w}, h, \{\lambda_l\})$ $w.r.t$ $\lambda_l$ [37, 52], we have,

$$\sum_{l=1}^{|\mathbf{A}^t|} \left\langle \nabla_{\lambda_l} \widetilde{L}_p(\boldsymbol{w}^t, h^t, \{\lambda_l^t\}) - \nabla_{\lambda_l} \widetilde{L}_p(\boldsymbol{w}^t, h^t, \{\lambda_l^{t-1}\}), \lambda_l^t - \lambda_l^{t-1} \right\rangle$$

$$\leq \sum_{l=1}^{|\mathbf{A}^t|} \left( -\frac{1}{L_1' + c_1^{t-1}} || \nabla_{\lambda_l} \widetilde{L}_p(\boldsymbol{w}^t, h^t, \{\lambda_l^t\}) - \nabla_{\lambda_l} \widetilde{L}_p(\boldsymbol{w}^t, h^t, \{\lambda_l^{t-1}\}), \boldsymbol{v}_{1,l}^{t+1}||^2 - \frac{c_1^{t-1} L_1'}{L_1' + c_1^{t-1}} ||\lambda_l^t - \lambda_l^{t-1}||^2 \right). \tag{D.131}$$

In addition, we can obtain the following inequality,

$$\frac{1}{\rho_1} \left\langle \lambda_l^t - \lambda_l^{t-1}, \lambda_l^{t+1} - \lambda_l^t \right\rangle \leq \frac{1}{2\rho_1} ||\lambda_l^{t+1} - \lambda_l^t||^2 - \frac{1}{2\rho_1} ||\boldsymbol{v}_{1,l}^{t+1}||^2 + \frac{1}{2\rho_1} ||\lambda_l^t - \lambda_l^{t-1}||^2. \tag{D.132}$$

Combining Eq. (D.127), (D.129), (D.130), (D.131), (D.132) with $\frac{\rho_1}{2} \leq \frac{1}{L_1' + c_1^0}$, and setting $a_2 = \rho_1$, $\forall t \geq T_1$, we have,

$$L_p(\boldsymbol{w}^{t+1}, h^{t+1}, \{\lambda_l^{t+1}\}) - L_p(\boldsymbol{w}^{t+1}, h^{t+1}, \{\lambda_l^t\})$$

$$\leq \frac{|\mathbf{A}^t| L^2}{2a_1} (||\boldsymbol{w}^{t+1} - \boldsymbol{w}^t||^2 + ||h^{t+1} - h^t||^2) + \left( \frac{a_1}{2} - \frac{c_1^{t-1} - c_1^t}{2} + \frac{1}{2\rho_1} \right) \sum_{l=1}^{|\mathbf{A}^t|} ||\lambda_l^{t+1} - \lambda_l^t||^2 \tag{D.133}$$

$$+ \frac{c_1^{t-1}}{2} \sum_{l=1}^{|\mathbf{A}^t|} (||\lambda_l^{t+1}||^2 - ||\lambda_l^t||^2) + \frac{1}{2\rho_1} \sum_{l=1}^{|\mathbf{A}^t|} ||\lambda_l^t - \lambda_l^{t-1}||^2.$$

By combining Lemma D.1 with Eq. (D.133), we conclude the proof of Lemma D.2.

**Lemma D.3** *Denote:*

$$S_1^{t+1} = \frac{4}{\rho_1^2 c_1^{t+1}} \sum_{l=1}^{|\mathbf{A}^t|} ||\lambda_l^{t+1} - \lambda_l^t||^2 - \frac{4}{\rho_1} \left( \frac{c_1^{t-1}}{c_1^t} - 1 \right) \sum_{l=1}^{|\mathbf{A}^t|} ||\lambda_l^{t+1}||^2, \tag{D.134}$$

$$F^{t+1} = L_p(\boldsymbol{w}^{t+1}, h^{t+1}, \{\lambda_l^{t+1}\}) + S_1^{t+1} - \frac{7}{2\rho_1} \sum_{l=1}^{|\mathbf{A}^t|} ||\lambda_l^{t+1} - \lambda_l^t||^2 - \frac{c_1^t}{2} \sum_{l=1}^{|\mathbf{A}^t|} ||\lambda_l^{t+1}||^2, \tag{D.135}$$

*then $\forall t \geq T_1$, we have:*

$$F^{t+1} - F^t$$

$$\leq \left( \frac{L}{2} - \frac{1}{\eta_w^t} + \frac{\rho_1 |\mathbf{A}^t| L^2}{2} + \frac{8 |\mathbf{A}^t| L^2}{\rho_1 (c_1^t)^2} \right) ||\boldsymbol{w}^{t+1} - \boldsymbol{w}^t||^2 + \left( \frac{L}{2} - \frac{1}{\eta_h^t} + \frac{\rho_1 |\mathbf{A}^t| L^2}{2} + \frac{8 |\mathbf{A}^t| L^2}{\rho_1 (c_1^t)^2} \right) ||h^{t+1} - h^t||^2$$

$$- \frac{1}{10\rho_1} \sum_{l=1}^{|\mathbf{A}^t|} ||\lambda_l^{t+1} - \lambda_l^t||^2 + \frac{c_1^{t-1} - c_1^t}{2} \sum_{l=1}^{|\mathbf{A}^t|} ||\lambda_l^{t+1}||^2 + \frac{4}{\rho_1} \left( \frac{c_1^{t-2}}{c_1^{t-1}} - \frac{c_1^{t-1}}{c_1^t} \right) \sum_{l=1}^{|\mathbf{A}^t|} ||\lambda_l^t||^2. \tag{D.136}$$

*Proof:*

Let $a_1 = \frac{1}{\rho_1}$ and substitute it into Lemma D.2, $\forall t \geq T_1$, we have,

$$L_p(\boldsymbol{w}^{t+1}, h^{t+1}, \{\lambda_l^{t+1}\}) - L_p(\boldsymbol{w}^t, h^t, \{\lambda_l^t\})$$

$$\leq \left( \frac{L}{2} - \frac{1}{\eta_w^t} + \frac{\rho_1 |\mathbf{A}^t| L^2}{2} \right) ||\boldsymbol{w}^{t+1} - \boldsymbol{w}^t||^2 + \left( \frac{L}{2} - \frac{1}{\eta_h^t} + \frac{\rho_1 |\mathbf{A}^t| L^2}{2} \right) ||h^{t+1} - h^t||^2$$

$$+ \left( -\frac{c_1^{t-1} - c_1^t}{2} + \frac{1}{\rho_1} \right) \sum_{l=1}^{|\mathbf{A}^t|} ||\lambda_l^{t+1} - \lambda_l^t||^2 + \frac{c_1^{t-1}}{2} \sum_{l=1}^{|\mathbf{A}^t|} (||\lambda_l^{t+1}||^2 - ||\lambda_l^t||^2) + \frac{1}{2\rho_1} \sum_{l=1}^{|\mathbf{A}^t|} ||\lambda_l^t - \lambda_l^{t-1}||^2. \tag{D.137}$$

Firstly, $\forall t \geq T_1$, we can obtain the following inequality,

$$\sum_{l=1}^{|\mathbf{A}^t|} \frac{1}{\rho_1} \left\langle \boldsymbol{v}_{1,l}^{t+1}, \lambda_l^{t+1} - \lambda_l^t \right\rangle$$

$$\leq \sum_{l=1}^{|\mathbf{A}^t|} (\left\langle \nabla_{\lambda_l} \widetilde{L}_p(\boldsymbol{w}^{t+1}, h^{t+1}, \{\lambda_l^t\}) - \nabla_{\lambda_l} \widetilde{L}_p(\boldsymbol{w}^t, h^t, \{\lambda_l^t\}), \lambda_l^{t+1} - \lambda_l^t \right\rangle \tag{D.138}$$

$$+ \left\langle \nabla_{\lambda_l} \widetilde{L}_p(\boldsymbol{w}^t, h^t, \{\lambda_l^t\}) - \nabla_{\lambda_l} \widetilde{L}_p(\boldsymbol{w}^t, h^t, \{\lambda_l^{t-1}\}), \boldsymbol{v}_{1,l}^{t+1} \right\rangle$$

$$+ \left\langle \nabla_{\lambda_l} \widetilde{L}_p(\boldsymbol{w}^t, h^t, \{\lambda_l^t\}) - \nabla_{\lambda_l} \widetilde{L}_p(\boldsymbol{w}^t, h^t, \{\lambda_l^{t-1}\}), \lambda_l^t - \lambda_l^{t-1} \right\rangle ).$$

Since

$$\frac{1}{\rho_1} \left\langle \boldsymbol{v}_{1,l}^{t+1}, \lambda_l^{t+1} - \lambda_l^t \right\rangle = \frac{1}{2\rho_1} ||\lambda_l^{t+1} - \lambda_l^t||^2 + \frac{1}{2\rho_1} ||\boldsymbol{v}_{1,l}^{t+1}||^2 - \frac{1}{2\rho_1} ||\lambda_l^t - \lambda_l^{t-1}||^2, \tag{D.139}$$

it follows that,

$$\sum_{l=1}^{|\mathbf{A}^t|} (\frac{1}{2\rho_1} ||\lambda_l^{t+1} - \lambda_l^t||^2 + \frac{1}{2\rho_1} ||\boldsymbol{v}_{1,l}^{t+1}||^2 - \frac{1}{2\rho_1} ||\lambda_l^t - \lambda_l^{t-1}||^2)$$

$$\leq \sum_{l=1}^{|\mathbf{A}^t|} (\frac{L^2}{2b_1^t} (||\boldsymbol{w}^{t+1} - \boldsymbol{w}^t||^2 + ||h^{t+1} - h^t||^2) + \frac{b_1^t}{2} ||\lambda_l^{t+1} - \lambda_l^t||^2$$

$$+ \frac{c_1^{t-1} - c_1^t}{2} (||\lambda_l^{t+1}||^2 - ||\lambda_l^t||^2) - \frac{c_1^{t-1} - c_1^t}{2} ||\lambda_l^{t+1} - \lambda_l^t||^2$$

$$+ \frac{\rho_1}{2} ||\nabla_{\lambda_l} \widetilde{L}_p(\boldsymbol{w}^t, h^t, \{\lambda_l^t\}) - \nabla_{\lambda_l} \widetilde{L}_p(\boldsymbol{w}^t, h^t, \{\lambda_l^{t-1}\}), \boldsymbol{v}_{1,l}^{t+1}||^2 + \frac{1}{2\rho_1} ||\boldsymbol{v}_{1,l}^{t+1}||^2$$

$$- \frac{1}{L_1' + c_1^{t-1}} ||\nabla_{\lambda_l} \widetilde{L}_p(\boldsymbol{w}^t, h^t, \{\lambda_l^t\}) - \nabla_{\lambda_l} \widetilde{L}_p(\boldsymbol{w}^t, h^t, \{\lambda_l^{t-1}\}), \boldsymbol{v}_{1,l}^{t+1}||^2 - \frac{c_1^{t-1} L_1'}{L_1' + c_1^{t-1}} ||\lambda_l^t - \lambda_l^{t-1}||^2), \tag{D.140}$$

where $b_1^t > 0$. According to the setting that $c_1^0 \leq L_1'$, we have $-\frac{c_1^{t-1} L_1'}{L_1' + c_1^{t-1}} \leq -\frac{c_1^{t-1} L_1'}{2L_1'} = -\frac{c_1^{t-1}}{2} \leq -\frac{c_1^t}{2}$. Multiplying both sides of the inequality Eq. (D.140) by $\frac{8}{\rho_1 c_1^t}$ and setting $b_1^t = \frac{c_1^t}{2}, \forall t \geq T_1$, we have,

$$S_1^{t+1} - S_1^t \leq \sum_{l=1}^{|\mathbf{A}^t|} \frac{4}{\rho_1} (\frac{c_1^{t-2}}{c_1^{t-1}} - \frac{c_1^{t-1}}{c_1^t}) ||\lambda_l^t||^2 + \sum_{l=1}^{|\mathbf{A}^t|} (\frac{2}{\rho_1} + \frac{4}{\rho_1^2} (\frac{1}{c_1^{t+1}} - \frac{1}{c_1^t})) ||\lambda_l^{t+1} - \lambda_l^t||^2$$

$$- \sum_{l=1}^{|\mathbf{A}^t|} \frac{4}{\rho_1} ||\lambda_l^t - \lambda_l^{t-1}||^2 + \frac{8 |\mathbf{A}^t| L^2}{\rho_1 (c_1^t)^2} (||\boldsymbol{w}^{t+1} - \boldsymbol{w}^t||^2 + ||h^{t+1} - h^t||^2). \tag{D.141}$$

According to the setting about $c_1^t$, we have $\frac{\rho_1}{10} \geq \frac{1}{c_1^{t+1}} - \frac{1}{c_1^t}, \forall t \geq T_1$. Using the definition of $F^{t+1}$ and combining it with Eq. (D.141) and Eq. (D.137), $\forall t \geq T_1$, we have,

$$F^{t+1} - F^t$$

$$\leq (\frac{L}{2} - \frac{1}{\eta_{\boldsymbol{w}}^t} + \frac{\rho_1 |\mathbf{A}^t| L^2}{2} + \frac{8 |\mathbf{A}^t| L^2}{\rho_1 (c_1^t)^2}) ||\boldsymbol{w}^{t+1} - \boldsymbol{w}^t||^2 + (\frac{L}{2} - \frac{1}{\eta_h^t} + \frac{\rho_1 |\mathbf{A}^t| L^2}{2} + \frac{8 |\mathbf{A}^t| L^2}{\rho_1 (c_1^t)^2}) ||h^{t+1} - h^t||^2$$

$$- \frac{1}{10\rho_1} \sum_{l=1}^{|\mathbf{A}^t|} ||\lambda_l^{t+1} - \lambda_l^t||^2 + \frac{c_1^{t-1} - c_1^t}{2} \sum_{l=1}^{|\mathbf{A}^t|} ||\lambda_l^{t+1}||^2 + \frac{4}{\rho_1} (\frac{c_1^{t-2}}{c_1^{t-1}} - \frac{c_1^{t-1}}{c_1^t}) \sum_{l=1}^{|\mathbf{A}^t|} ||\lambda_l^t||^2.$$

***Proof of Theorem D.1:***

Firstly, we set that $a_5^t = \frac{4 |\mathbf{A}^t| (\gamma - 2) L^2}{\rho_1 (c_1^t)^2}$, where $\gamma > 2$ is a constant. According to the setting of $\eta_{\boldsymbol{w}}^t, \eta_h^t$ and $c_1^t$, we have,

$$\frac{L}{2} - \frac{1}{\eta_{\boldsymbol{w}}^t} + \frac{\rho_1 |\mathbf{A}^t| L^2}{2} + \frac{8 |\mathbf{A}^t| L^2}{\rho_1 (c_1^t)^2} = -a_5^t, \quad \frac{L}{2} - \frac{1}{\eta_h^t} + \frac{\rho_1 |\mathbf{A}^t| L^2}{2} + \frac{8 |\mathbf{A}^t| L^2}{\rho_1 (c_1^t)^2} = -a_5^t. \tag{D.142}$$

Combining with Lemma D.3, $\forall t \geq T_1$, it follows that,

$$a_5^t ||\boldsymbol{w}^{t+1} - \boldsymbol{w}^t||^2 + a_5^t ||h^{t+1} - h^t||^2 + \frac{1}{10\rho_1} \sum_{l=1}^{|\mathbf{A}^t|} ||\lambda_l^{t+1} - \lambda_l^t||^2$$

$$\leq F^t - F^{t+1} + \frac{c_1^{t-1} - c_1^t}{2} \sum_{l=1}^{|\mathbf{A}^t|} ||\lambda_l^{t+1}||^2 + \frac{4}{\rho_1} (\frac{c_1^{t-2}}{c_1^{t-1}} - \frac{c_1^{t-1}}{c_1^t}) \sum_{l=1}^{|\mathbf{A}^t|} ||\lambda_l^t||^2. \tag{D.143}$$

According to the definition of $(\nabla \widetilde{G}(t))_{\boldsymbol{w}}$, we have,

$$||(\nabla \widetilde{G}^t)_{\boldsymbol{w}}||^2 \leq \frac{1}{(\eta_{\boldsymbol{w}}^t)^2}||\boldsymbol{w}^{t+1}-\boldsymbol{w}^t||^2. \tag{D.144}$$

Combining the definition of $(\nabla \widetilde{G}(t))_h$ with trigonometric, Cauchy-Schwarz inequality and Assumption D.1, we have,

$$||(\nabla \widetilde{G}^t)_h||^2 \leq 2L^2||\boldsymbol{w}^{t+1}-\boldsymbol{w}^t||^2+\frac{2}{(\eta_h^t)^2}||h^{t+1}-h^t||^2. \tag{D.145}$$

Combining the definition of $(\nabla \widetilde{G}^t)_{\lambda_l}$ with trigonometric inequality and Cauchy-Schwarz inequality,

$$||(\nabla \widetilde{G}^t)_{\lambda_l}||^2 \leq \frac{3}{\rho_1^2}||\lambda_l^{t+1}-\lambda_l^t||^2+3L^2(||\boldsymbol{w}^{t+1}-\boldsymbol{w}^t||^2+||h^{t+1}-h^t||^2)+3((c_1^{t-1})^2-(c_1^t)^2)||\lambda_l^t||^2. \tag{D.146}$$

According to the Definition D.2 as well as Eq. (D.144), (D.145) and Eq. (D.146), we can obtain,

$$||\nabla \widetilde{G}^t||^2 \leq (\frac{1}{(\eta_{\boldsymbol{w}}^t)^2}+2L^2+3|\mathbf{A}^t|L^2)||\boldsymbol{w}^{t+1}-\boldsymbol{w}^t||^2+(\frac{2}{(\eta_h^t)^2}+3|\mathbf{A}^t|L^2)||h^{t+1}-h^t||^2$$
$$+\sum_{l=1}^{|\mathbf{A}^t|}\frac{3}{\rho_1^2}||\lambda_l^{t+1}-\lambda_l^t||^2+\sum_{l=1}^{|\mathbf{A}^t|}3((c_1^{t-1})^2-(c_1^t)^2)||\lambda_l^t||^2. \tag{D.147}$$

We set constants $d_1$, $d_2$ as

$$d_1 = \frac{1+(2+3M)L^2\underline{\eta_{\boldsymbol{w}}}^2}{\underline{\eta_{\boldsymbol{w}}}^2(a_5^0)^2} \geq \frac{1+(2+3|\mathbf{A}^t|)L^2(\eta_{\boldsymbol{w}}^t)^2}{(\eta_{\boldsymbol{w}}^t)^2(a_5^t)^2}, \tag{D.148}$$

$$d_2 = \frac{2+3ML^2\underline{\eta_h}^2}{\underline{\eta_h}^2(a_5^0)^2} \geq \frac{2+3|\mathbf{A}^t|L^2(\eta_h^t)^2}{(\eta_h^t)^2(a_5^t)^2}, \tag{D.149}$$

where $\underline{\eta_{\boldsymbol{w}}} = \frac{2}{L+\rho_1 ML^2+8\frac{M\gamma L^2}{\rho_1 c_1^2}} \leq \eta_{\boldsymbol{w}}^t$ and $\underline{\eta_h} = \frac{2}{L+\rho_1 ML^2+8\frac{M\gamma L^2}{\rho_1 c_1^2}} \leq \eta_h^t, \forall t$ are positive constants.

Thus, combining Eq. (D.147) with Eq. (D.148) and Eq. (D.149), we can obtain,

$$||\nabla \widetilde{G}^t||^2 \leq d_1(a_5^t)^2||\boldsymbol{w}^{t+1}-\boldsymbol{w}^t||^2+d_2(a_5^t)^2||h^{t+1}-h^t||^2$$
$$+\sum_{l=1}^{|\mathbf{A}^t|}\frac{3}{\rho_1^2}||\lambda_l^{t+1}-\lambda_l^t||^2+\sum_{l=1}^{|\mathbf{A}^t|}3((c_1^{t-1})^2-(c_1^t)^2)||\lambda_l^t||^2. \tag{D.150}$$

Let $d_3^t$ denote a nonnegative sequence, $d_3^t = \frac{1}{\max\{d_1 a_5^t, d_2 a_5^t, \frac{30}{\rho_1}\}}$, and we have,

$$d_3^t||\nabla \widetilde{G}^t||^2 \leq a_5^t||\boldsymbol{w}^{t+1}-\boldsymbol{w}^t||^2+a_5^t||h^{t+1}-h^t||^2$$
$$+\frac{1}{10\rho_1}\sum_{l=1}^{|\mathbf{A}^t|}||\lambda_l^{t+1}-\lambda_l^t||^2+3d_3^t((c_1^{t-1})^2-(c_1^t)^2)\sum_{l=1}^{|\mathbf{A}^t|}||\lambda_l^t||^2. \tag{D.151}$$

Combining Eq. (D.151) with Eq. (D.143) and according to the setting $||\lambda_l^t||^2 \leq \sigma_1^2$ (where $\sigma_1^2 = \alpha_3^2$) and $d_3^0 \geq d_3^t, \forall t \geq T_1$, we have that,

$$d_3^t||\nabla \widetilde{G}^t||^2 \leq F^t-F^{t+1}+\frac{c_1^{t-1}-c_1^t}{2}M\sigma_1^2+\frac{4}{\rho_1}(\frac{c_1^{t-2}}{c_1^{t-1}}-\frac{c_1^{t-1}}{c_1^t})M\sigma_1^2+3d_3^0((c_1^{t-1})^2-(c_1^t)^2)M\sigma_1^2. \tag{D.152}$$

Denoting $\widetilde{T}(\varepsilon)$ as $\widetilde{T}(\varepsilon) = \min\{t \mid ||\nabla \widetilde{G}^{T_1+t}|| \leq \frac{\varepsilon}{2}, t \geq 2\}$. Summing up Eq. (D.152) from $t = T_1+2$ to $t = T_1+\widetilde{T}(\varepsilon)$, we have,

$$\sum_{t=T_1+2}^{T_1+\widetilde{T}(\varepsilon)}d_3^t||\nabla \widetilde{G}^t||^2 \leq F^{T_1+2}-\underline{L}+\frac{4}{\rho_1}(\frac{c_1^{T_1}}{c_1^{T_1+1}}+\frac{c_1^{T_1+1}}{c_1^{T_1+2}})M\sigma_1^2+\frac{c_1^{T_1+1}}{2}M\sigma_1^2$$
$$+\frac{7}{2\rho_1}M\sigma_3^2+\frac{c_1^{T_1+2}}{2}M\sigma_1^2+3d_3^0(c_1^0)^2M\sigma_1^2$$
$$= \overline{d}, \tag{D.153}$$

where $\sigma_3 = \max\{||\lambda_1 - \lambda_2|| \,|\, \lambda_1, \lambda_2 \in \mathbf{\Lambda}\}$ and $\underline{L} = \min_{\boldsymbol{w} \in \mathcal{W}, h \in \mathcal{H}, \{\lambda_l \in \mathbf{\Lambda}\}} L_p(\boldsymbol{w}, h, \{\lambda_l\})$, which satisfy that,

$$F^{t+1} \geq \underline{L} - \frac{4}{\rho_1} \frac{c_1^{T_1+1}}{c_1^{T_1+2}} M\sigma_1{}^2 - \frac{7}{2\rho_1} M\sigma_3{}^2 - \frac{c_1^{T_1+2}}{2} M\sigma_1{}^2, \ \forall t \geq T_1 + 2, \qquad (D.154)$$

and $\overline{d}$ is a constant. Constant $d_5$ is given by,

$$d_5 = \max\{d_1, d_2, \frac{30}{\rho_1 a_5^0}\} \geq \max\{d_1, d_2, \frac{30}{\rho_1 a_5^t}\} = \frac{1}{d_3^t a_5^t} \ . \qquad (D.155)$$

Thus, we can obtain that,

$$\sum_{t=T_1+2}^{T_1+\widetilde{T}(\varepsilon)} \frac{1}{d_5 a_5^t} ||\nabla \widetilde{G}^{T_1+\widetilde{T}(\varepsilon)}||^2 \leq \sum_{t=T_1+2}^{T_1+\widetilde{T}(\varepsilon)} d_3^t ||\nabla \widetilde{G}^t||^2 \leq \overline{d} \ . \qquad (D.156)$$

Summing up $\frac{1}{a_5^t}$ from $t = T_1 + 2$ to $t = T_1 + \widetilde{T}(\varepsilon)$, it follows that,

$$\sum_{t=T_1+2}^{T_1+\widetilde{T}(\varepsilon)} \frac{1}{a_5^t} \geq \sum_{t=T_1+2}^{T_1+\widetilde{T}(\varepsilon)} \frac{1}{4(\gamma-2)L^2 M\rho_1 (t+1)^{\frac{1}{2}}} \geq \frac{(T_1+\widetilde{T}(\varepsilon))^{\frac{1}{2}} - (T_1+2)^{\frac{1}{2}}}{4(\gamma-2)L^2 M\rho_1}. \qquad (D.157)$$

Combining Eq. (D.156), (D.157) with the definition of $\widetilde{T}(\varepsilon)$, we have that,

$$T_1 + \widetilde{T}(\varepsilon) \geq (\frac{16(\gamma-2)L^2 M\rho_1 \,\overline{d}\, d_5}{\varepsilon^2} + (T_1+2)^{\frac{1}{2}})^2. \qquad (D.158)$$

According to trigonometric inequality, we then get $||\nabla G^t|| - ||\nabla \widetilde{G}^t|| \leq ||\nabla G^t - \nabla \widetilde{G}^t|| \leq \sqrt{\sum_{l=1}^{|\mathbf{A}^t|} ||c_1^{t-1}\lambda_l^t||^2}$. If $t > \frac{16 M^2 \sigma_1{}^4}{\rho_1{}^4} \frac{1}{\varepsilon^4}$, we have $\sqrt{\sum_{l=1}^{|\mathbf{A}^t|} ||c_1^{t-1}\lambda_l^t||^2} \leq \frac{\varepsilon}{2}$. Combining it with Eq. (D.158), we can conclude that there exists a

$$T(\varepsilon) \sim \mathcal{O}(\max\{(\frac{16(\gamma-2)L^2 M\rho_1 \,\overline{d}\, d_5}{\varepsilon^2} + (T_1+2)^{\frac{1}{2}})^2, \frac{16 M^2 \sigma_1{}^4}{\rho_1{}^4} \frac{1}{\varepsilon^4}\}), \qquad (D.159)$$

such that $||\nabla G^t|| \leq \varepsilon$, which concludes our proof.

# E   Convergence Rate Analysis

In this section, we compare the convergence results of the proposed method against the existing methods in the literature (with centralized and distributed setting). GDmax [25] is proposed recently, which can be utilized to solve the nonconvex-concave minimax problems (related to the setting of our problem) with iteration complexity $\mathcal{O}(\frac{1}{\varepsilon^6})$ to obtain the $\varepsilon$-stationary point (*i.e.*, $||\Phi(\cdot)||^2 \leq \varepsilon^2$, where $\Phi(\cdot) = \max_y f(\cdot, y)$). However, GDmax is nested-loop which has to solve the inner subproblem every iteration [52]. Gradient descent-ascent (GDA) method [31] is proposed, which performs alternating gradient descent-ascent every iteration. The iteration complexity of GDA to obtain the $\varepsilon$-stationary point (*i.e.*, $||\Phi(\cdot)||^2 \leq \varepsilon^2$) for nonconvex-concave minimax problems is upper bounded by $\mathcal{O}(\frac{1}{\varepsilon^6})$. COVER [38] is proposed to solve the distributionally robust optimization with nonconvex objectives, which can obtain the $\varepsilon$-stationary point (*i.e.*, $||\mathcal{G}_\eta(\cdot)||^2 \leq \varepsilon^2$, where $\mathcal{G}_\eta$ is a proximal gradient measure) with the complexity $\mathcal{O}(\frac{1}{\varepsilon^3})$. Nevertheless, all the algorithms mentioned above do not discuss about the distributed algorithms. Recently, GCIVR [22] is proposed to solve the distributionally robust optimization problem in centralized and distributed manners. GCIVR is effective, which can respectively obtain the $\varepsilon$-stationary point (*i.e.*, $||\mathcal{G}_\eta(\cdot)||^2 \leq \varepsilon^2$) with the complexity $\mathcal{O}(\min\{\frac{\sqrt{N}}{\varepsilon^2}, \frac{1}{\varepsilon^3}\})$ and $\mathcal{O}(\min\{\frac{\sqrt{N}}{p\varepsilon^2} + \frac{\sqrt{N}}{\varepsilon^2}, \frac{1}{p\varepsilon^3} + \frac{1}{\varepsilon^3}\})$ ($p$ is the number of workers, in this problem $p = N$) in centralized and distributed manners when the objective is nonconvex.

Table E1: Convergence rate of algorithms related to our work (with centralized and distributed setting).

| Method | Centralized | Synchronous (Distributed) | Asynchronous (Distributed) |
|---|---|---|---|
| GDmax [25] | $\mathcal{O}(\frac{1}{\varepsilon^6})^{1,3}$ | NA[5] | NA[5] |
| GDA [31] | $\mathcal{O}(\frac{1}{\varepsilon^6})^1$ | NA[5] | NA[5] |
| COVER [38] | $\mathcal{O}(\frac{1}{\varepsilon^3})^2$ | NA[5] | NA[5] |
| GCIVR [22] | $\mathcal{O}(\min\{\frac{\sqrt{N}}{\varepsilon^2}, \frac{1}{\varepsilon^3}\})^2$ | $\mathcal{O}(\min\{\frac{\sqrt{N}}{p\varepsilon^2}+\frac{\sqrt{N}}{\varepsilon^2}, \frac{1}{p\varepsilon^3}+\frac{1}{\varepsilon^3}\})^{2,4}$ | NA[5] |
| **ASPIRE-EASE** | $\mathcal{O}(\frac{1}{\varepsilon^4})$ | NA[5] | $\mathcal{O}(\frac{1}{\varepsilon^6})$ |

[1] This complexity is to find an $\varepsilon$-stationary point of $\Phi(\cdot) = \max_y f(\cdot, y)$, that is $||\Phi(\cdot)||^2 \leq \varepsilon^2$.
[2] This complexity is to find an $\varepsilon$-stationary point such that $||\mathcal{G}_\eta(\cdot)||^2 \leq \varepsilon^2$, where $\mathcal{G}_\eta$ is a proximal gradient measure.
[3] This complexity corresponds to the number of iterations to solve the inner subproblem. It does not consider the complexity of solving the inner subproblem.
[4] $p$ is the number of workers, in this problem $p = N$.
[5] NA represents not applicable.

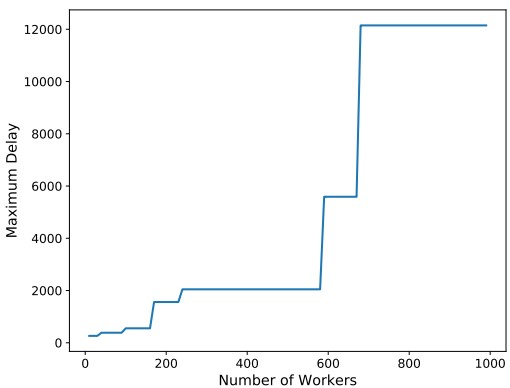

Figure E1: With the increase of the number of workers in the distributed system, the maximum delay would increase dramatically. The delay follows log-normal distribution LN(1, 0.4) in the experiment.

The proposed algorithm differs significantly from the aforementioned methods because it is designed for solving the PD-DRO problem in Eq. (4) in an *asynchronous distributed manner*. The asynchronous distributed algorithm does not suffer from the straggler problem [24] and therefore is critical for large scale distributed optimization in practice. On the contrary, synchronous distributed algorithm suffers from the straggler problem, *i.e.*, its speed is limited by the worker with maximum delay [10] and may not scale well with the size of a distributed system. For instance, we assume that the delays of workers follow a heavy-tailed distribution as given in [12]. With the increase of the number of workers in the distributed system, the maximum delay may increase dramatically as shown in Figure E1. Hence, the synchronous algorithm may incur huge delays and become practically infeasible for a large-scale distributed systems with tens of thousands of workers. Moreover, if a few workers fail to respond, which is very common in real-world large-scale data centers, the synchronous algorithm will come to an immediate halt [58]. Therefore, the asynchronous algorithm is strongly preferred in practice.

The asynchronous setting is considered when we design the distributed algorithm. Compared with centralized algorithm, the asynchronous distributed algorithm is more complicated, which pose the major challenge against the theoretical analysis. In the future work, how to improve the iteration complexity will be taken into consideration. And we summarize the convergence results of different methods in Table E1.

## F   Explanation about Assumption

The gradient Lipschitz (or smoothness) is a common assumption that has been widely used [16, 38, 31]. In some other works [47, 59], if the function $L_p$ is $\widetilde{L}$-smooth, it has to satisfy,

$$
\begin{aligned}
&||\nabla_\theta L_p(\{\boldsymbol{w}_j\}, \boldsymbol{z}, h, \{\lambda_l\}, \{\boldsymbol{\phi}_j\}) - \nabla_\theta L_p(\{\hat{\boldsymbol{w}}_j\}, \hat{\boldsymbol{z}}, \hat{h}, \{\hat{\lambda}_l\}, \{\hat{\boldsymbol{\phi}}_j\})|| \\
&\leq \widetilde{L}(\sum_{j=1}^{N} ||\boldsymbol{w}_j - \hat{\boldsymbol{w}}_j|| + ||\boldsymbol{z} - \hat{\boldsymbol{z}}|| + ||h - \hat{h}|| + \sum_{l=1}^{M} ||\lambda_l - \hat{\lambda}_l|| + \sum_{j=1}^{N} ||\boldsymbol{\phi}_j - \hat{\boldsymbol{\phi}}_j||),
\end{aligned} \tag{F.160}
$$

where $\theta \in \{\{\boldsymbol{w}_j\}, \boldsymbol{z}, h, \{\lambda_l\}, \{\boldsymbol{\phi}_j\}\}$ and we demonstrate $L_p$ that satisfies Eq. (F. 160) is also satisfied with our Assumption 1.

From Eq. (F. 160) and according to Cauchy-Schwarz inequality, we can obtain,

$$
\begin{aligned}
&||\nabla_\theta L_p(\{\boldsymbol{w}_j\}, \boldsymbol{z}, h, \{\lambda_l\}, \{\boldsymbol{\phi}_j\}) - \nabla_\theta L_p(\{\hat{\boldsymbol{w}}_j\}, \hat{\boldsymbol{z}}, \hat{h}, \{\hat{\lambda}_l\}, \{\hat{\boldsymbol{\phi}}_j\})||^2 \\
&\leq (2N + M + 2)\widetilde{L}^2(\sum_{j=1}^{N} ||\boldsymbol{w}_j - \hat{\boldsymbol{w}}_j||^2 + ||\boldsymbol{z} - \hat{\boldsymbol{z}}||^2 + ||h - \hat{h}||^2 + \sum_{l=1}^{M} ||\lambda_l - \hat{\lambda}_l||^2 + \sum_{j=1}^{N} ||\boldsymbol{\phi}_j - \hat{\boldsymbol{\phi}}_j||^2).
\end{aligned} \tag{F.161}
$$

Let $L = \sqrt{(2N + M + 2)}\widetilde{L}$, we can obtain,

$$
\begin{aligned}
&||\nabla_\theta L_p(\{\boldsymbol{w}_j\}, \boldsymbol{z}, h, \{\lambda_l\}, \{\boldsymbol{\phi}_j\}) - \nabla_\theta L_p(\{\hat{\boldsymbol{w}}_j\}, \hat{\boldsymbol{z}}, \hat{h}, \{\hat{\lambda}_l\}, \{\hat{\boldsymbol{\phi}}_j\})|| \\
&\leq L||[\boldsymbol{w}_{\text{cat}} - \hat{\boldsymbol{w}}_{\text{cat}}; \boldsymbol{z} - \hat{\boldsymbol{z}}; h - \hat{h}; \boldsymbol{\lambda}_{\text{cat}} - \hat{\boldsymbol{\lambda}}_{\text{cat}}; \boldsymbol{\phi}_{\text{cat}} - \hat{\boldsymbol{\phi}}_{\text{cat}}]||.
\end{aligned} \tag{F.162}
$$

## G   Discussion about $CD$-norm Uncertainty Set

In this paper, we utilize the $CD$-norm uncertainty set in our framework. Compared with ellipsoid and KL-divergence uncertainty sets, whose cutting plane generation subproblems are respectively a second-order cone optimization (SOCP) problem and a relative entropy programming (REP) problem, the cutting plane generation subproblem (Eq. (17)) is an LP-type problem when utilizing $CD$-norm uncertainty set. Please note that the LP-type problem in Eq. (17) can be efficiently solved by merge sort. Therefore, the cutting plane generation subproblem with $CD$-norm uncertainty set is much easier to solve than those with the ellipsoid and KL-divergence uncertainty sets.