# OpenReview forum: "Distributed Distributionally Robust Optimization with Non-Convex Objectives"
_NeurIPS.cc/2022/Conference — NeurIPS 2022 Accept_

### Official Review · Reviewer_fDdQ · 2022-07-09

**Rating:** 5
**Confidence:** 3
**Soundness:** 2 fair
**Presentation:** 2 fair
**Contribution:** 2 fair

**Summary:**

This paper proposed a distributed distributionally robust optimization method. Both theoretical convergence rate and empirical results are provided.

However, the convergence rate is much worse than existing methods.

**Questions:**

- In recent years, many optimization algorithms for nonconvex problems have been proposed and they can achieve a much faster convergence rate than this method, e.g. [1] [2].  However, the authors failed to discuss and compare them.
- It would be good if the authors could use a table to compare the convergence rate of this method and existing methods, including the centralized ones.
- As for asynchronous optimization, a critical hyperparameter is the communication delay, which heavily affects the convergence rate. But in your convergence rate, it is not clear how the communication delay affects it.
- It would be good if the authors could discuss the challenges in the theoretical analysis compared with the centralized method.
- What benefits to introducing $h$? Is it always nonnegative? In Table 1, what loss values did you report? $h$ or $\sum p_j f(w_j)$? Similarly, in figure 2, if it is $\sum p_j f(w_j)$, which is a min-max problem, why is it always decreasing?
- In table 1, FedAvg and DRFA-Prox perform periodic communication. But your method performs communication in each iteration. Hence, the comparison is not fair.



[1] An online method for a class of distributionally robust optimization with non-convex objectives

[2] Learning Distributionally Robust Models at Scale via Composite Optimization

**Strengths And Weaknesses:**

Pros:
- This paper is well organized.
- Provide extensive experimental results.

Cons:
- In recent years, many optimization algorithms for nonconvex problems have been proposed and they can achieve a much faster convergence rate than this method, e.g. [1] [2].  However, the authors failed to discuss and compare them.
- It would be good if the authors could use a table to compare the convergence rate of this method and existing methods, including the centralized ones.
- As for asynchronous optimization, a critical hyperparameter is the communication delay, which heavily affects the convergence rate. But in your convergence rate, it is not clear how the communication delay affects it.
- It would be good if the authors could discuss the challenges in the theoretical analysis compared with the centralized method.
- What benefits to introducing $h$? Is it always nonnegative? In Table 1, what loss values did you report? $h$ or $\sum p_j f(w_j)$? Similarly, in figure 2, if it is $\sum p_j f(w_j)$, which is a min-max problem, why is it always decreasing?
- In table 1, FedAvg and DRFA-Prox perform periodic communication. But your method performs communication in each iteration. Hence, the comparison is not fair.



[1] An online method for a class of distributionally robust optimization with non-convex objectives

[2] Learning Distributionally Robust Models at Scale via Composite Optimization

---

> ### Author Response · Authors · 2022-08-02
> **Author Response (Part 3)**
>
> (${\textbf{Q6}}$) In table 1, FedAvg and DRFA-Prox perform periodic communication. But your method performs communication in each iteration. Hence, the comparison is not fair.
>
> ${\textbf{Reply:}}$  Thanks for your insightful comments. We have added the performance of ASPIRE-EASE with periodic communication in Table 1 on Page 8 (For your convenience, the performance of ASPIRE-EASE with periodic communication is shown  in the Table 2 below). And we have added discussions in the 282$^{th}$ line on Page 8 as follows. ``And it is seen that ASPIRE-EASE can perform periodic communication, since ASPIRE-EASE$_{\rm{per}}$, which represents ASPIRE-EASE with periodic communication, also has excellent performance.'' Furthermore, ASPIRE-EASE with periodic communication can effectively reduce the number of communication rounds, which can be seen in Table 3.
>
> Table 2: The performance of ASPIRE-EASE with periodic communication.
>
> Dataset   $\qquad  \qquad  \qquad$  ${\bf{Acc}}_{w}$  $\qquad    \qquad$  ${\bf{Loss}}_w$  $\qquad   \qquad$  ${\bf{Std}}$
>
> Fashion MNIST  $\qquad \quad$   78.73±0.06  $\qquad   $  0.698±0.006   $\qquad    \\;  \\;$  1.09
>
> SC-MA $\qquad \qquad  \qquad  \\; $  56.71±0.16    $\qquad  $ 1.119±0.028  $\qquad  \\;  \\; $ 3.48
>
> Table 3: The number of communication rounds (between individual worker with master), $b$ represents the number of local updates in each worker before communication, $b=1$ represents that ASPIRE-EASE (without periodic communication).
>
> Dataset   $\qquad  \qquad  \qquad$  $b=1$  $\qquad    \qquad$  $b=3$  $\qquad   \qquad$  $b=5$
>
> Fashion MNIST  $\qquad \quad$   13500  $\qquad    \qquad$  5500   $\qquad    \qquad \\;  \\;$  3700
>
> SC-MA $\qquad \qquad  \qquad  \\; $  35200    $\qquad    \qquad$18600  $\qquad    \qquad \\; $ 10200
>
> ${\textbf{Reference}}$
>
> [1] An online method for a class of distributionally
> robust optimization with non-convex objectives.
>
> [2] Learning distributionally robust models at scale via composite optimization.
>
> [3] Asynchronous distributed ADMM for large-scale optimization—Part I: Algorithm and convergence analysis.
>
> [4] Asynchronous stochastic optimization robust to arbitrary delays.
>
> [5] Asynchronous distributed ADMM for consensus optimization.
>
> [6] What is local optimality in nonconvex-nonconcave minimax optimization?
>
> [7] On gradient descent ascent for nonconvex-concave minimax problems.
>
> [8] On the convergence rate of distributed gradient methods for finite-sum optimization under communication delays.
>
> [9] Robust optimization.
>
> [10] A survey of adjustable robust optimization.
>
> [11] Asynchronous distributed ADMM for consensus optimization.
>
> [12] Asynchronous online ADMM for consensus problems.
>
> [13] Asynchronous decentralized online learning.
>
> [14] Robust Optimization over Multiple Domains.

---

> ### Author Response · Authors · 2022-08-02
> **Author Response (Part 2)**
>
> (${\textbf{Q3}}$) As for asynchronous optimization, a critical hyperparameter is the communication delay, which heavily affects the convergence rate. But in your convergence rate, it is not clear how the communication delay affects it.
>
> ${\textbf{Reply:}}$ Thank you for your constructive comments. We fully agree with you that the communication delay is a critical hyperparameter for the convergence rate (time) [8] of a distributed algorithms. In section 5 (Convergence Analysis) of this paper, the iteration complexity ${T}(\varepsilon )$ is analyzed, which represents how many iterations the proposed algorithm needs to converge and is independent of the communication delay [11][12]. However, the convergence rate (time) will be heavily affected by the communication delay. Let ${{\mathcal{T}}}$ and ${T}(\varepsilon )$ denote respectively the convergence time and iterations of the proposed asynchronous algorithm. Suppose that there are $N$ workers in a distributed system and the number of active workers is $S$. For brevity, we assume the communication delay for each work remains the same during the iteration. Let $[{\hat{d}_1},{\hat{d}_2}, \cdots {\hat{d}_N}] \in {\mathbb{R}^N}$  denote a vector of communication delays for all $N$ workers. For the $j$-th worker, it has communicated with the master $\frac{{{\mathcal{T}}}}{{{d_j}}}$ times during time ${{\mathcal{T}}}$. Thus, for time ${{\mathcal{T}}  }$, it satisfies that,
>
> \begin{equation}
> \frac{{{\mathcal{T}}}}{{{\hat{d}_1}}} + \frac{{{\mathcal{T}}}}{{{\hat{d}_2}}} +  \cdots  + \frac{{{\mathcal{T}}}}{{{\hat{d}_N}}} = {T}(\varepsilon ) \times S.
> \end{equation}
>
> It is seen that the convergence time (i.e., ${{\mathcal{T}}}=\frac{{T}(\varepsilon ) \times S}{\frac{1}{{\hat{d}_1}}+\frac{1}{{\hat{d}_2}}+\cdots+\frac{1}{{\hat{d}_N}}}$) of the proposed asynchronous algorithm will be affected by the communication delay $[{\hat{d}_1},{\hat{d}_2}, \cdots {\hat{d}_N}] $.
>
> (${\textbf{Q4}}$) It would be good if the authors could discuss the challenges in the theoretical analysis compared with the centralized method.
>
> ${\textbf{Reply:}}$ Thanks for your constructive suggestion. We have added the convergence analysis about solving PD-DRO in a centralized manner in Appendix D, Page 33-41. And we have analyzed the challenge in the theoretical analysis of the proposed asynchronous algorithm compared with centralized method in the 874$^{th}$ line on Page 42 as ``The asynchronous setting is considered when we design the distributed algorithm. Compared with centralized algorithm, the asynchronous distributed algorithm is more complicated, e.g., it needs to additionally consider the update of equality constraints dual variables ${\boldsymbol{\phi}_j}, \forall j$, which pose the major challenge against the theoretical analysis.''
>
> (${\textbf{Q5}}$) What benefits to introducing $h$? Is it always nonnegative? In Table 1, what loss values did you report? $h$ or $\sum\nolimits_j p_j {f_j}(w_j)$? Similarly, in figure 2, if it is $\sum\nolimits_j p_j {f_j}(w_j)$, which is a min-max problem, why is it always decreasing?
>
> ${\textbf{Reply:}}$ Introducing the additional variable $h$ is a common reformulation technique employed in [9, 10]. This reformulation could also be viewed as an epigraph reformulation which can facilitate the design of the asynchronous algorithm. $h$ is always nonnegative in the experiments because $h\ge\mathop {\max }\limits_{{\bf{p}}\in \boldsymbol{\mathcal{P}}} \sum\nolimits_j p_j {f_j}(w_j)$, in which ${f_j}(w_j)$ is the cross entropy. Based on your suggestion, we have added the motivation of introducing additional variable $h$ in the 109$^{th}$ line on Page 3, as " Introducing additional variable $h$ is an epigraph reformulation [9, 10] ." The loss value we reported is not $h$ or$\sum\nolimits_j p_j {f_j}(w_j)$, but the worst case training loss [14] (the training loss on the worker with worst performance), we have added this description in the 256$^{th}$ line on Page 7, as ``Specifically, we use ${\bf{Acc}}_{w}$ and ${\bf{Loss}}_w$ to represent the worst case test accuracy and training loss (i.e., the test accuracy and training loss on the worker with worst performance), respectively.''

---

> ### Author Response · Authors · 2022-08-02
> **Author Response (Part 1)**
>
> Thank you for your insightful comments. We have provided a point by point reply to your questions as follows.
>
> (${\textbf{Q1}}$) In recent years, many optimization algorithms for nonconvex problems have been proposed and they can achieve a much faster convergence rate than this method, e.g. [1] [2]. However, the authors failed to discuss and compare them. [1] An online method for a class of distributionally robust optimization with non-convex objectives. [2] Learning Distributionally Robust Models at Scale via Composite Optimization.
>
> ${\textbf{Reply:}}$ We truly appreciate your suggestion and are very grateful to you for providing two important references, i.e., [1][2] to us. Per your suggestion, we have cited [1][2] and added comparisons regarding the convergence results of the proposed method against the existing methods in the literature (including the methods presented in [1][2]) in Appendix E, Page 41.
>
> As opposed to [1][2], we emphasize on developing the asynchronous distributed algorithm for DRO, as shown in Appendix E, Page 41. For your convenience, we have also enclosed the explanations below.
>
> ``The proposed algorithm differs significantly from the aforementioned methods because it is designed for solving the PD-DRO problem in Eq. (4) in an ${asynchronous}$ ${distributed}$ ${manner}$. The asynchronous distributed algorithm does not suffer from the straggler problem [13] and therefore is critical for large scale distributed optimization in practice. On the contrary, synchronous distributed algorithm suffers from the straggler problem, i.e., its speed is limited by the worker with maximum delay [3] and may not scale well with the size of a distributed system. For instance, we assume that the delays of workers follow a heavy-tailed distribution as given in [4]. With the increase of the number of workers in the distributed system, the maximum delay may increase dramatically as shown in Figure E1. Hence, the synchronous algorithm may incur huge delays and become practically infeasible for a large-scale distributed systems with tens of thousands of workers. Moreover, if a few workers fail to respond, which is very common in real-world large-scale data centers, the synchronous algorithm will come to an immediate halt [5]. Therefore, the asynchronous algorithm is strongly preferred in practice.''
>
> (${\textbf{Q2}}$)  It would be good if the authors could use a table to compare the convergence rate of this method and existing methods, including the centralized ones.
>
> ${\textbf{Reply:}}$ Thanks for your insightful comments. We have added a table (Table E1, Appendix E) which compares the convergence results of the proposed method and existing methods (including the centralized ones). In addition, the iteration complexity of the proposed method with asynchronous setting (i.e., $\mathcal{O}(\frac{1}{\varepsilon ^6})$) to obtain a $\varepsilon$-stationary point (i.e., $||\nabla G^t||^2  \le \varepsilon ^2 $) has been updated. And a new proof about that the iteration complexity of the proposed method to solve the PD-DRO in centralized manner is upper bounded by $\mathcal{O}(\frac{1}{\varepsilon ^4})$ has been also been added into Appendix D, Page 33-41.
>
> Table E1: Convergence results (with centralized and distributed setting).
>
> Method $\qquad \qquad \\;$  Centralized    $\qquad \qquad$   Synchronous (Distributed)   $\qquad \qquad$  Asynchronous (Distributed)
>
> GDmax[6] $\qquad \qquad$ $\mathcal{O}(\frac{1}{{{\varepsilon ^6}}})^{1,3}$   $\qquad  \qquad \qquad \qquad \quad$   NA$^5$ $\qquad \qquad  \qquad \qquad \qquad \qquad$  NA$^5$
>
> GDA[7]  $\qquad \qquad \quad$  $\mathcal{O}(\frac{1}{{{\varepsilon ^6}}})^{1}$     $\qquad  \qquad \qquad \qquad \quad  \\; \\; \\;$   NA$^5$ $\qquad \qquad  \qquad \qquad \qquad \qquad$  NA$^5$
>
> COVER[1] $\qquad \qquad$  $\mathcal{O}(\frac{1}{{{\varepsilon ^3}}})^{2}$     $\qquad  \qquad \qquad \qquad \quad \\; \\;  \\;$   NA$^5$ $\qquad \qquad  \qquad \qquad \qquad \qquad$ NA$^5$
>
> GCIVR[2] $\qquad $ $\mathcal{O}(\min( \frac{\sqrt N }{{{\varepsilon ^2}}}, \frac{1}{{{\varepsilon ^3}}}))^{2}$  $\qquad$  $\mathcal{O}(\min(\frac{\sqrt N }{{{p\varepsilon ^2}}}+\frac{\sqrt N }{{{\varepsilon ^2}}}, \frac{1}{{{p\varepsilon ^3}}}+\frac{1}{{{\varepsilon ^3}}}))^{2, 4}$  $\qquad \qquad \\;  \\; \\;$ NA$^5$
>
> ASPIRE-EASE $\qquad \quad $   $\mathcal{O}(\frac{1}{{\varepsilon ^4}})$ $\qquad  \qquad \qquad \qquad \quad \\; \\;  \\;$ NA$^5$   $\qquad \qquad  \qquad   \qquad  \qquad \qquad$$\mathcal{O}(\frac{1}{{\varepsilon ^6}})$
>
> $^1$ This complexity is to find an $\varepsilon$-stationary point  of $\Phi ( \cdot ) = {\max _y}f( \cdot ,y)$, i.e., $||\Phi ( \cdot )|{|^2} \le {\varepsilon ^2}$.
>
> $^2$ This complexity is to find an $\varepsilon$-stationary point  such that  $||g_{\eta} ( \cdot )|{|^2} \le {\varepsilon ^2}$, where $g_{\eta}$ is a proximal gradient measure.
>
> $^3$ Nested loop algorithm.
>
> $^4$ $p$ is the number of workers, in our problem $p=N$.
>
> $^5$ NA represents not applicable.

---

> > ### Comment · Reviewer_fDdQ · 2022-08-04
> > **More questions**
> >
> > Thanks for your answers. Here, I have more questions on the convergence rate.
> > - In the initial submission, the iteration complexity is $O(1/\epsilon^8)$. But it becomes $O(1/\epsilon^6)$ in the new version. What did you revise? I think you need to highlight what you revised.
> > - It seems the centralized variant in Appendix D uses the full gradient. So the iteration complexity is not optimal. Again, as existing algorithms have faster convergence speed, what's the advantage of the proposed algorithm?

---

> > > ### Author Response · Authors · 2022-08-05
> > > **Author Response**
> > >
> > > Thanks for your constructive comments again.
> > >
> > > (${\textbf{Q1}}$) In the initial submission, the iterations complexity is $\mathcal{O}(\frac{1}{{\varepsilon ^8}})$. But it becomes $\mathcal{O}(\frac{1}{{\varepsilon ^6}})$ in the new version. What did you revise? I think you need to highlight what you revised.
> > >
> > > ${\textbf{Reply:}}$ We truly appreciate your insightful comments. In the initial submission, we derive a loose upper bound (i.e., $\mathcal{O}(\frac{1}{{\varepsilon ^8}})$) to the iteration complexity of the proposed method. In the revision, we found out this upper bound can be further tightened to $\mathcal{O}(\frac{1}{{\varepsilon ^6}})$. Per your suggestion, we have highlighted the revisions in blue in the latest submission (223$^{th}$ line, 228$^{th}$ line, 874$^{th}$ line, Eq. (A.70), Eq. (A.71), Eq. (A.80), Eq. (A.81), Eq. (A.82), Eq. (A.83), Eq. (A.91), Eq. (A.92),  Eq. (A.93), Eq. (A.94), Eq. (A.95), Eq. (A.96),  Eq. (A.97), Eq. (A.98), Eq. (A.100)).
> > >
> > > (${\textbf{Q1}}$) It seems the centralized variant in Appendix D uses the full gradient. So the iteration complexity is not optimal. Again, as existing algorithms have faster convergence speed, what's the advantage of the proposed algorithm?
> > >
> > > ${\textbf{Reply:}}$ Thank you for your constructive comments. Similar to utilizing the mini-batch loss in ASPIRE-EASE, the centralized variant in Appendix D also uses the mini-batch loss ${\hat{f}_j}({\boldsymbol{w}})$ (i.e., ${\hat{f}_j}({\boldsymbol{w}}) = \sum\limits_\{i = 1\}^m {\frac{1}{m}{{\cal L}_j}({\bf{x}}_j^i,y_j^i;{\boldsymbol{w}})}$, $m$ is the mini-batch size and $\mathbb{E}[{\hat{f}_j}({\boldsymbol{w}})]={f_j}({\boldsymbol{w}})$). Thus, the centralized variant does not use the full gradient. According to your suggestions, we have explained this point in 790$^{th}$ line on Page 34 as ``To avoid enumerating the whole dataset, the mini-batch loss  ${\hat{f}_j}({\boldsymbol{w}})  =  \sum\limits_\{i = 1\}^m {\frac{1}{m}{{\cal L}_j}({\bf{x}}_j^i,y_j^i;{\boldsymbol{w}})}$ is used, where $m$ is the mini-batch size. It is evident that $\mathbb{E}[{\hat{f}_j}({\boldsymbol{w}})]={f_j}({\boldsymbol{w}})$ and $\mathbb{E}[\nabla {\hat{f}_j}({\boldsymbol{w}})]=\nabla{f_j}({\boldsymbol{w}})$.''
> > >
> > > The main advantage of ASPIRE-EASE is that it is an asynchronous algorithm while existing methods are synchronous algorithms. Synchronous distributed algorithms suffer from the stagger problem, i.e., its speed is limited by the worker with maximum delay [1] and may not scale well with the size of a distributed system. On the contrary, the asynchronous distributed algorithm does not suffer from the straggler problem [2]. Moreover, if a few workers fail to respond, which is very common in real-world large-scale data centers, the synchronous algorithm will come to an immediate halt [3]. Therefore, the asynchronous algorithm is strongly preferred in practice (902$^{th}$ line to 914$^{th}$ line).  As pointed out by the other reviewer, the asynchronous aspect of ASPIRE-EASE is novel.
> > >
> > > ${\textbf{Reference}}$
> > >
> > > [1] Asynchronous distributed ADMM for large-scale optimization—Part I: Algorithm and convergence analysis.
> > >
> > > [2] Asynchronous decentralized online learning.
> > >
> > > [3] Asynchronous distributed ADMM for consensus optimization.

---

> > > > ### Comment · Reviewer_fDdQ · 2022-08-05
> > > > **Thanks for your answers**
> > > >
> > > > As for Line 825, why $\tilde{L}_p=L_p$? It looks not correct. Why does it lead to D.122?

---

> > > > > ### Author Response · Authors · 2022-08-05
> > > > > **Author Response**
> > > > >
> > > > > Thank you for pointing this out, this is a typo, the correct one should be ${\nabla _{\boldsymbol{w}}}\widetilde{L}_p({\boldsymbol{w}}^{t},h^{t},\{ {\lambda _l^{t}}\}) ={\nabla _{\boldsymbol{w}}} {L_p}({\boldsymbol{w}}^{t},h^{t},\{ {\lambda _l^{t}}\})$, we have updated it in the current version.

---

### Official Review · Reviewer_CFN7 · 2022-07-09

**Rating:** 5
**Confidence:** 3
**Soundness:** 2 fair
**Presentation:** 3 good
**Contribution:** 2 fair

**Summary:**

The authors propose three challenges that existing DRO techniques face. To this end, they propose an asynchronous distributed algorithm to tackle the distributed distributionally robust optimization problem. Moreover, they propose a new uncertainty set which is developed to effectively leverage the prior distribution and flexibly control the degree of robustness. They provide theoretical guatantees for convergence of the algorithm. Extensive empirical studies corroborate their theoretical results.

**Questions:**

1. Can the authors elaborate more on the assumptions in the theoretical analysis?
2. Can the authors explain to what aspects the proposed algorithm address the challenge stated in line 6?

**Ethics Review Area:**

["I don’t know"]

**Limitations:**

1. The authors did not explain whether the assumptions made in theoretical analysis is practical.
2. To my knowledge, I do not understand why the proposed algorithm address the challenge stated in line 6. Can the authors explain more clearly?

**Strengths And Weaknesses:**

Strength:
1. They propose an algorithm with theoretical guarantees.
2. They conduct experiments to verify their statement.

Weakness:
They make  numerous assumptions in the theoretical analysis, but did not explain whether these assumptions are practical.

---

> ### Author Response · Authors · 2022-08-02
> **Author Response**
>
> Thanks for your comments. Below please find the reply to your questions.
>
> (${\textbf{Q1}}$) Can the authors elaborate more on the assumptions in the theoretical analysis?
>
> ${\textbf{Reply:}}$ We appreciate your suggestion, the gradient Lipschitz (or smoothness) is the main assumption in our analysis, we have added discussions about this assumption in Appendix F, Page 43. For your convenience, we give the explanations as follows:
>
> The gradient Lipschitz (or smoothness) is a common assumption that has been widely used [1][2][3]. In some other works [4][5], if the function $L_p$ is $\widetilde{L}$-smooth, it has to satisfy,
>
> \begin{equation}
>     \begin{array}{l}
> ||{\nabla  _\theta }{L_p}( \\{ {{\boldsymbol{w}}_j}\\},{\boldsymbol{z}},h,  \\{ {\lambda _l}\\}, \\{ {{\boldsymbol{\phi}}_j}\\} ) -  {\nabla  _\theta }{L_p}( \\{ {\hat{\boldsymbol{w}}_j}\\},\hat{\boldsymbol{z}},\hat{h},  \\{ {\hat{\lambda }_l}\\}, \\{ {\hat{\boldsymbol{\phi}}_j}\\}  )|| \\
>  \le \widetilde{L}( {\sum\nolimits_j||{{\boldsymbol{w}}_j} - {\hat{\boldsymbol{w}}_j}||}  + ||{\boldsymbol{z}} - \hat{\boldsymbol{z}}|| + ||h - \hat{h}|| +  \sum\nolimits_l{||{\lambda _l} - {\hat{\lambda}_l}||}  + {\sum\nolimits_j||{{\boldsymbol{\phi}}_j} - {\hat{\boldsymbol{\phi}}_j}||)} ,
> \end{array}
> \end{equation}
>
> where $\theta  \in  \\{ \\{ {{\boldsymbol{w}}_j}\\} ,{\boldsymbol{z}},h,\\{ {\lambda _l}\\} ,\\{ {{\boldsymbol{\phi}}_j}\\} \\} $  and we demonstrate $L_p$ that satisfies Eq.(1) is also satisfied with our Assumption 1.
>
> According to Cauchy-Schwarz inequality, we can obtain,
> \begin{equation}
>     \begin{array}{l}
> ||{\nabla  _\theta }{L_p}( \\{ {{\boldsymbol{w}}_j}\\},{\boldsymbol{z}},h,  \\{ {\lambda _l}\\}, \\{ {{\boldsymbol{\phi}}_j}\\} ) -  {\nabla  _\theta }{L_p}( \\{ {\hat{\boldsymbol{w}}_j}\\},\hat{\boldsymbol{z}},\hat{h},  \\{ {\hat{\lambda }_l}\\}, \\{ {\hat{\boldsymbol{\phi}}_j}\\}  )||^2\\
>  \le (2N  +  M   +   2)\widetilde{L}{^2}(\sum\nolimits_j {||{{\boldsymbol{w}}_j} - {\hat{\boldsymbol{w}}_j}|{|^2}}  + ||{\boldsymbol{z}} - \hat{\boldsymbol{z}}|{|^2} + ||h - \hat{h}|{|^2} +\sum\nolimits_l  {||{\lambda _l} - {\hat{\lambda}_l}|{|^2}}  +\sum\nolimits_j  {||{{\boldsymbol{\phi}}_j} - {\hat{\boldsymbol{\phi}}_j}|{|^2}} ),
> \end{array}
> \end{equation}
>
> Let $L = \sqrt {(2N + M + 2)} \widetilde{L}$, we can obtain,
>
> \begin{equation}
>  ||{\nabla  _\theta }{L_p}( \\{ {{\boldsymbol{w}}_j}\\},{\boldsymbol{z}},h,  \\{ {\lambda _l}\\}, \\{ {{\boldsymbol{\phi}}_j}\\} )  -  {\nabla  _\theta }{L_p}( \\{ {\hat{\boldsymbol{w}}_j}\\},\hat{\boldsymbol{z}},\hat{h},  \\{ {\hat{\lambda }_l}\\}, \\{ {\hat{\boldsymbol{\phi}}_j}\\}  )||\le L|| [ {\boldsymbol{w} _{cat}} - {{\hat{\boldsymbol{w}} _{cat}}}; \boldsymbol{z}- \hat{\boldsymbol{z}}; h- \hat{h}; {\boldsymbol{\lambda} _{cat}} - {{\hat{\boldsymbol{\lambda}} _{cat}}}; {\boldsymbol{\phi} _{cat}}- {{\hat{\boldsymbol{\phi}} _{cat}}}]||
> \end{equation}
>
> (${\textbf{Q2}}$) Can the authors explain to what aspects the proposed algorithm address the challenge stated in line 6?
>
> ${\textbf{Reply:}}$ Thank you for your insightful suggestion. We explain “how to leverage the prior distribution effectively” as follows. ${\bf{q}}=[q_1, \dots, q_N]  \in  {\mathbb{R}^N}$ in Eq. (16) represents the prior distribution, the uncertainty set $\boldsymbol{\mathcal{P}}$ (Eq.(16)) represents a convex set we construct, which encloses the prior distribution vector ${\bf{q}}$. And we utilize ${\bf{p}} \in \boldsymbol{\mathcal{P}}$ as the constraint in our PD-DRO problem in Eq. (4). Different from utilizing the prior distribution as a regularizer [1], utilizing ${\bf{p}} \in \boldsymbol{\mathcal{P}}$ as the constraint in PD-DRO can allow us to adjust the level of robustness in terms of the outage probability i.e., probabilistic bounds of constraint violations using a single parameter $\Gamma$. It has been shown that the outage probability decreases exponentially with the increase of $\Gamma$ (in Eq. (16)) [6].
>
> (${\textbf{Limitation}}$) 1) The authors did not explain whether the assumptions made in theoretical analysis is practical.
> 2) To my knowledge, I do not understand why the proposed algorithm address the challenge stated in line 6. Can the authors explain more clearly?
>
> ${\textbf{Reply:}}$ Thanks for your suggestion. Per your suggestion, we have added more discussions about our assumption into the revised version of the paper. Secondly, we have explained “how to leverage the prior distribution effectively” in the reply to your second question.
>
> ${\textbf{Reference}}$
>
> [1] Distributionally robust federated averaging.
>
> [2] On gradient descent ascent for nonconvex-concave minimax problems.
>
> [3] An online method for a class of distributionally robust optimization with non-convex objectives.
>
> [4] Efficient algorithms for smooth minimax optimization.
>
> [5] On the fenchel duality between strong convexity and lipschitz continuous gradient.
>
> [6] Distributed robust optimization (DRO), part I: Framework and example.

---

> ### Author Response · Authors · 2022-08-06
> **We are happy to address any further concerns**
>
> We sincerely thank you for raising the concerns in the initial reviews. We have tried our best efforts to clarify those concerns in the responses. Given the limited time for discussion, we would really appreciate it if you could let us know in case there is any additional concern.

---

### Official Review · Reviewer_X3TP · 2022-07-11

**Rating:** 6
**Confidence:** 4
**Soundness:** 3 good
**Presentation:** 3 good
**Contribution:** 3 good

**Summary:**

In this paper the authors introduce an algorithm to solve the Distributionally Robust Optimization problem in an asynchronous and distributed fashion where the data has been spread across multiple workers.
Specifically, they focus on a DRO problem where the distribution has finite support (number of workers) and formulate it as a single master problem with finite number of constraints. They then use an augmented lagrangian algorithm to solve the problem where each worker updates the components of the augmented lagrangian using the data available to it.
The provide a convergence analysis for their method and illustrate their results through numerical experiments.

**Questions:**

1. What is the novel about the CD-norm uncertainty set. It seems essentially the same as the regular budget uncertainty set. The probability constraints sum p_i = 1, don't seem to have any impact on the analysis.
2. I think the authors to better emphasize the asynchronous aspect of their algorithm. I believe this is hte novel aspect of their method but is not emphasized especially in the numerical methods.

**Limitations:**

I think the authors can better address the limitations of their work especially the limitations imposed by the  the specific budget uncertainty set that they use. It would also be interesting to see if there are any differences in the type of solution obtained by there method as compared to other methods.

**Strengths And Weaknesses:**

Strengths
1. The provide a method for solving DRO that can leverage the parallelism present in modern computer systems.
2. Their assumptions and numerical experiments allow their method to be applicable to a wide variety of settings including popular machine learning methods.
3. They provide a convergence analysis which shows that their method will converge to a stationary point.

Weaknesses
1. The uncertainty set used is quite common and the analysis using it is pretty standard.

---

> ### Author Response · Authors · 2022-08-02
> **Author Response (Part 2)**
>
> (${\textbf{Limitation}}$) I think the authors can better address the limitations of their work especially the limitations imposed by the the specific budget uncertainty set that they use. It would also be interesting to see if there are any differences in the type of solution obtained by this method as compared to other methods.
>
> ${\textbf{Reply:}}$ Thank you for your suggestion. We have added more discussions about CD-norm uncertainty set in Appendix G, Page 43 as follows. ``Compared with ellipsoid and KL-divergence uncertainty sets, whose cutting plane generation subproblems are respectively a second-order cone optimization (SOCP) problem and a relative entropy programming (REP) problem, the cutting plane generation subproblem (Eq. (17)) is a LP-type problem when utilizing CD-norm uncertainty set. The arithmetic complexity of interior point method to find the $\varepsilon'$-solution for SOCP and REP is respectively  $\mathcal{O}((m+1)^{1/2}n({n^{2}+m+\sum\limits_{i = 1}^{m} {k_i^2} })\log (\frac{1}{{\varepsilon '}}))$ [7] and $\mathcal{O}({(n)^{7/2}}|\log (\varepsilon ')|)$ [8], where $m$ and $n$ are respectively the number of inequality constraints and variables, and $k_i$ represents that the $i$-th inequality constraint is a $k_i+1$ dimension second-order cone. Please note that the LP-type problem in Eq. (17) can be efficiently solved by merge sort with arithmetic complexity $\mathcal{O}(n\log (n))$. Therefore, the cutting plane generation subproblem with CD-norm uncertainty set is much easier to solve than those with the ellipsoid and KL-divergence uncertainty sets. In addition, compared with box and polyhedron uncertainty sets, the level of robustness can be flexibly adjusted by tuning a single parameter $\Gamma$ when utilizing the CD-norm uncertainty set. Moreover, under mild assumptions, it has been proved that the outage probability decreases exponentially with the increase of $\Gamma$ [9].''
>
> ${\textbf{Reference}}$
>
> [1] Agnostic federated learning.
>
> [2] Distributionally robust federated averaging.
>
> [3] Robust optimization over multiple domains.
>
> [4] Asynchronous distributed ADMM for large-scale optimization—Part I: Algorithm and convergence analysis.
>
> [5] Asynchronous stochastic optimization robust to arbitrary delays.
>
> [6] Asynchronous distributed ADMM for consensus optimization.
>
> [7] Lectures on modern convex optimization (2012).
>
> [8] A quadratically convergent polynomial algorithm for solving entropy optimization problems.
>
> [9] Distributed robust optimization (DRO), part I: Framework and example.
>
> [10] Asynchronous decentralized online learning.

---

> ### Author Response · Authors · 2022-08-02
> **Author Response (Part 1)**
>
> Thank you for your constructive suggestions. Below please find our point by point reply to the questions you have raised.
>
> (${\textbf{Q1}}$)  What is the novel about the CD-norm uncertainty set. It seems essentially the same as the regular budget uncertainty set. The probability constraints ${\bf{1}^ \top }{\bf{p}} = 1$, dont seem to have any impact on the analysis.
>
> ${\textbf{Reply:}}$ We truly appreciate your insightful comments. ${\bf{1}^ \top }{\bf{p}} = 1$ is a probability vector so all the entries of this vector are non-negative and add up to exactly one. It is an indispensable constraint for the adversarial distribution [1][2][3]. And the constraint ${\bf{1}^ \top }{\bf{p}} = 1$ influences the generation of cutting plane, which can be seen from Eq. (17). We have added the detailed description about the CD-norm, starting from the 184$^{th}$ line on Page 5. ``We call Eq. (16) CD-norm uncertainty set since ${\bf{p}}$ is a probability vector so all the entries of this vector are non-negative and add up to exactly one, i.e., ${\bf{1}^ \top }{\bf{p}} = 1$. Due to the special structure of CD-norm, the cutting plane generation subproblem is easy to solve and the level of robustness in terms of the outage probability, i.e., probabilistic bounds of the violations of constraints can be flexibly adjusted via a single parameter $\Gamma$.''
>
> (${\textbf{Q2}}$)  I think the authors to better emphasize the asynchronous aspect of their algorithm. I believe this is the novel aspect of their method but is not emphasized especially in the numerical methods.
>
> ${\textbf{Reply:}}$ Thank you for your constructive suggestions. We fully agree with you that the asynchronous aspect of the proposed algorithm is novel. Per your suggestions, more discussions regarding the asynchronous algorithm have been added to the 903$^{th}$ line on Page 41 as follows.
>
> ``The asynchronous distributed algorithm does not suffer from the straggler problem [10] and therefore is critical for large scale distributed optimization in practice. On the contrary, synchronous distributed algorithm suffers from the stagger problem, i.e., its speed is limited by the worker with maximum delay [4] and may not scale well with the size of a distributed system. For instance, we assume that the delays of workers follow a heavy-tailed distribution as given in [5]. With the increase of the number of workers in the distributed system, the maximum delay increases dramatically as shown in Figure E1. Hence, the synchronous algorithm may incur huge delays and become practically infeasible for a large-scale distributed systems with tens of thousands of workers. Moreover, if a few workers fail to respond, which is very common in real-world large-scale data centers, the synchronous algorithm will come to an immediate halt [6]. Therefore, the asynchronous algorithm is strongly preferred in practice.''
>
> Moreover, the advantage of the asynchronous setting of our method can also be seen in Figure 2, i.e., ASPIRE-EASE VS ASPIRE-EASE(-), where ASPIRE-EASE(-) represents ASPIRE-EASE without asynchronous setting.

---

> ### Author Response · Authors · 2022-08-06
> **We are happy to address any further concerns**
>
> We sincerely thank you for raising the concerns in the initial reviews. We have tried our best efforts to clarify those concerns in the responses. Given the limited time for discussion, we would really appreciate it if you could let us know in case there is any additional concern.

---

### Author Response · Authors · 2022-08-09
**Summary of concerns and our responses.**

Thanks for the constructive suggestions from all reviewers. Since the author-reviewer discussion will end very soon, we summarize the main concerns from reviewers and our corresponding replies as follows to facilitate further discussions.

$\textbf{(Reviewer X3TP)}$

$\textbf{(Q)}$ Impact of ${\bf{1}^ \top }{\bf{p}} = 1$ on the analysis, novelty about CD-norm uncertainty set.

$\textbf{Reply:}$ ${\bf{1}^ \top }{\bf{p}} = 1$ is a probability vector so all the entries of this vector are non-negative and add up to exactly one. It is an indispensable constraint for the adversarial distribution. It influences the generation of cutting plane, which can be seen from Eq. (17). Detailed description about the CD-norm uncertainty set have been added in 184$^{th}$ line on Page 5. More discussions about CD-norm uncertainty set are added  in Appendix G.

$\textbf{(Q)}$ Emphasize the asynchronous aspect of the proposed algorithm.

$\textbf{Reply:}$ More discussions regarding the asynchronous algorithm have been added in Appendix E. Specifically, synchronous distributed algorithms suffer from the stagger problem, i.e., its speed is limited by the worker with maximum delay and may not scale well with the size of a distributed system. On the contrary, the asynchronous distributed algorithm does not suffer from the straggler problem. Moreover, if a few workers fail to respond, which is very common in real-world large-scale data centers, the synchronous algorithm will come to an immediate halt  while the asynchronous algorithm  allows the system to proceed.. Therefore, the asynchronous algorithm is strongly preferred in practice.

$\textbf{(Reviewer CFN7)}$

$\textbf{(Q)}$ How to leverage the prior distribution effectively?

$\textbf{Reply:}$ ${\bf{q}}=[q_1, \dots, q_N]  \in  {\mathbb{R}^N}$ in Eq. (16) represents the prior distribution, the uncertainty set $\boldsymbol{\mathcal{P}}$ (Eq.(16)) represents a convex set we construct, which encloses the prior distribution vector ${\bf{q}}$. We utilize ${\bf{p}} \in \boldsymbol{\mathcal{P}}$ as the constraint in our PD-DRO problem in Eq. (4). Different from utilizing the prior distribution as a regularizer, utilizing ${\bf{p}} \in \boldsymbol{\mathcal{P}}$ as the constraint in PD-DRO can allow us to adjust the level of robustness in terms of the outage probability i.e., probabilistic bounds of constraint violations using a single parameter $\Gamma$.

$\textbf{(Q)}$ Discussion about the assumption.

$\textbf{Reply:}$ More detailed discussions about the assumptions have been added in Appendix F.


$\textbf{(Reviewer fDdQ)}$

$\textbf{(Q)}$ Comparison about convergence rate.

$\textbf{Reply:}$ We have added comparisons regarding the convergence results of the proposed method against the existing methods in the literature in Appendix E. Table E1 is also added to summarize the convergence results in Appendix E.


$\textbf{(Q)}$ Challenges in the theoretical analysis compared with centralized method.

$\textbf{Reply:}$ We have analyzed the challenges in the theoretical analysis of the proposed asynchronous algorithm compared with centralized one in Appendix E. Specifically, the asynchronous algorithm is more complicated, e.g., it needs to additionally consider the update of equality constraints dual variables ${\boldsymbol{\phi}_j}, \forall j$, which pose the major challenge against the theoretical analysis.

$\textbf{(Q)}$ Periodic communication.

$\textbf{Reply:}$ We have added the performance of ASPIRE-EASE with periodic communication in Table 1.

$\textbf{(Q)}$ Advantage of the proposed algorithm.

$\textbf{Reply:}$ The main advantage of ASPIRE-EASE is that it is an asynchronous algorithm while existing methods are synchronous algorithms (for PD-DRO problem). The asynchronous algorithm is strongly preferred in practice, since it does not suffer from the straggler problem and therefore is critical for large scale distributed optimization. Nevertheless, the synchronous algorithm suffers from the straggler problem (i.e., its speed is limited by the worker with maximum delay) and will come to an immediate halt  if a few workers fail to respond.

---

### Meta-Review · Area_Chair_9k6k · 2022-08-21

**Recommendation:** Accept
**Confidence:** Less certain

**Metareview:**

Reviewers generally recommend (weak) acceptance however there are still quite a few issues that should be addressed. I concur.

**Award:**

No

---

### Decision · Program_Chairs · 2022-09-14

Accept